# A Single-Timescale Analysis for Stochastic Approximation with Multiple Coupled Sequences

**Han Shen**
Rensselaer Polytechnic Institute
Troy, NY 12180, United States
shenh5@rpi.edu

**Tianyi Chen**
Rensselaer Polytechnic Institute
Troy, NY 12180, United States
chentianyi19@gmail.com

## Abstract

Stochastic approximation (SA) with multiple coupled sequences has found broad applications in machine learning such as bilevel learning and reinforcement learning (RL). In this paper, we study the finite-time convergence of nonlinear SA with multiple coupled sequences. Different from existing multi-timescale analysis, we seek for scenarios where a fine-grained analysis can provide the tight performance guarantee for single-timescale multi-sequence SA (STSA). At the heart of our analysis is the smoothness property of the fixed points in multi-sequence SA that holds in many applications. When all sequences have strongly monotone increments, we establish the iteration complexity of $\mathcal{O}(\epsilon^{-1})$ to achieve $\epsilon$-accuracy, which improves the existing $\mathcal{O}(\epsilon^{-1.5})$ complexity for two coupled sequences. When the main sequence does not have strongly monotone increment, we establish the iteration complexity of $\mathcal{O}(\epsilon^{-2})$. We showcase the power of our result by applying it to stochastic bilevel and compositional optimization problems, as well as RL problems, all of which lead to improvements over their existing guarantees.

## 1 Introduction

Stochastic approximation (SA) is an iterative procedure used to find the zero of a function when only the noisy estimate of the function is observed. Specifically, with the mapping $v : \mathbb{R}^d \mapsto \mathbb{R}^d$, the single-sequence SA seeks to solve for $v(x) = 0$ with the following iterative update:

$$x_{k+1} = x_k + \alpha_k(v(x_k) + \xi_k), \tag{1}$$

where $\alpha_k$ is the step size and $\xi_k$ is a random variable. Since its introduction in [46], single-sequence SA has received great interests because of its broad range of applications to areas including stochastic optimization and reinforcement learning (RL) [6, 53]. The asymptotic convergence of single-sequence SA can be established by the ordinary differential equation method; see e.g., [4]. To gain more insights into the performance difference of various stochastic optimization algorithms, the finite-time convergence of SA has been widely studied in recent years; see e.g., [43, 42, 30, 50, 54, 52, 41, 13].

While most of the SA studies focus on the single-sequence case, the double-sequence SA was introduced in [3], which has been extensively applied to the RL methods involving a double-sequence stochastic update structure [53, 32, 10]. With mappings $v : \mathbb{R}^{d_0} \times \mathbb{R}^{d_1} \mapsto \mathbb{R}^{d_0}$ and $h : \mathbb{R}^{d_0} \times \mathbb{R}^{d_1} \mapsto \mathbb{R}^{d_1}$, the double-sequence SA seeks to solve $v(x, y) = h(x, y) = 0$ with the following update:

$$x_{k+1} = x_k + \alpha_k(v(x_k, y_k) + \xi_k), \tag{2a}$$
$$y_{k+1} = y_k + \beta_k(h(x_k, y_k) + \psi_k), \tag{2b}$$

where $\alpha_k, \beta_k$ are the step sizes, and $\xi_k, \psi_k$ are random variables. In (2), the update of $x_k$ and that of $y_k$ depend on each other and thus the sequences are *coupled*. To analyze (2), one natural thought is to stack $(x_k, y_k)$ into a larger variable and then resort to the celebrated single-sequence

36th Conference on Neural Information Processing Systems (NeurIPS 2022).

SA analysis [46]. However, it can be seen later that the assumptions required for convergence of this *virtual* single-sequence SA will be violated in many applications and the two sequences may be updated in an alternating rather than simultaneous fashion, both of which motivate a new analysis for double-sequence SA. Due to the coupling, the double-sequence SA is more challenging to analyze than its single-sequence counterpart.

**Prior art on double-sequence SA.** Many recent analyses of the double-sequence SA focus on the linear case where $v(x,y)$ and $h(x,y)$ are linear mappings; see e.g., [34, 11, 25, 29]. The key idea here is to use the so-called two-time-scale (TTS) step sizes: *One sequence is updated in the faster time scale while the other is updated in the slower time scale; that is* $\lim_{k\to\infty} \alpha_k/\beta_k = 0$. By doing so, the two sequences are shown to decouple asymptotically, which allows us to leverage the analysis of the single-sequence SA. In particular, [29] proves an iteration complexity of $\mathcal{O}(\epsilon^{-1})$ to achieve $\epsilon$-accuracy for the TTS linear SA, which is shown to be tight. With similar choice of the step sizes, the TTS nonlinear SA was analyzed in [39, 12]. In [39], the finite-time convergence rate of TTS nonlinear SA was established under an assumption that the two sequences converge asymptotically. Later, this assumption has been relaxed in [12] which shows that TTS nonlinear SA achieves an iteration complexity of $\mathcal{O}(\epsilon^{-1.5})$. However, this iteration complexity is larger than $\mathcal{O}(\epsilon^{-1})$ of the TTS linear SA. The single time-scale step sizes have also been explored in [37, 44], but the enabling factor in those works is the vanishing variance by incorporating variance-reduction update or increasing batch size. While this work and those in Table 1 focus on the case where the variance is non-decreasing.

The gap between the complexities of nonlinear and linear SA motivates an interesting question:

### Q1: Is it possible to prove a faster rate for the nonlinear SA with two coupled sequences?

We first conduct an experiment to examine the possibility.

**Experiment.** Figure 1 shows the performance of using the double-sequence SA (2) to solve the following problem

$$\max_{x\in\mathbb{R}} \quad -\frac{1}{2}\left(x^2 + \frac{1}{1+e^{-y^*(x)}}\right)$$

$$\text{s.t. } y^*(x) = \arg\min_{y\in\mathbb{R}} \frac{1}{2}(y-x)^2. \quad (3)$$

We use the double-sequence SA (2) to solve (3), where

$$v(x,y) = -x - \frac{e^{-y}}{(1+e^{-y})^2}, \quad h(x,y) = x-y \quad (4)$$

and $\zeta_k$, $\xi_k$ are independent Gaussian random variables with zero mean and standard deviations of $0.15$. It is easy to check that (4) satisfies the assumptions in the existing TTS-SA analysis [12]. Therefore, we can use the two time-scale step sizes and achieve the iteration complexity of $\mathcal{O}(\epsilon^{-1.5})$. However, as suggested by Figure 1, the iterates still converge with step sizes in a single-time-scale

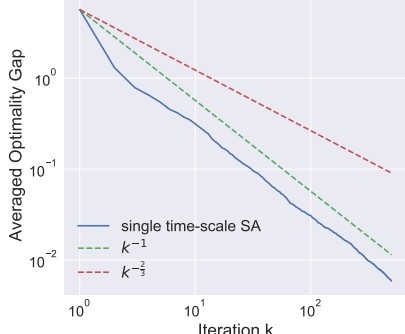

Figure 1: Solving (3) with double-sequence nonlinear SA (2). The single time-scale nonlinear SA converges with a rate of $\mathcal{O}(k^{-1})$, which is faster than the theoretical $\mathcal{O}(k^{-\frac{2}{3}})$ rate in [12].

$(\alpha_k = \Theta(\frac{1}{k}), \beta_k = \Theta(\frac{1}{k}))$. In this case, the iteration complexity is $\mathcal{O}(\epsilon^{-1})$, which is the same as that of double-sequence linear SA [29]. This suggests that existing analysis of double-sequence SA might not be tight, at least for the class of updates similar to (4). Indeed, as we will show later, the iterates generated by (4) will converge with the iteration complexity of $\mathcal{O}(\epsilon^{-1})$.

Furthermore, existing works on TTS SA mainly focus on the double-sequence case. While in cases such as the multi-level stochastic optimization; see e.g., [61], more than two sequences are involved. This necessitates the use of the multi-sequence SA. Specifically, with mappings $v : \mathbb{R}^{d_0} \times \mathbb{R}^{d_1} \cdots \times \mathbb{R}^{d_N} \mapsto \mathbb{R}^{d_0}, h^n : \mathbb{R}^{d_{n-1}} \times \mathbb{R}^{d_n} \mapsto \mathbb{R}^{d_n}$, we consider

$$\textbf{(STSA)} \quad y_{k+1}^n = y_k^n + \beta_{k,n}\left(h^n(y_k^{n-1}, y_k^n) + \psi_k^n\right), \quad n = 1, 2, ..., N \quad (5a)$$

$$x_{k+1} = x_k + \alpha_k\left(v(x_k, y_k^1, y_k^2, \ldots, y_k^N) + \xi_k\right) \quad (5b)$$

| | General result | | Application to SBO | | | | Application to multi-level SCO | | |
|---|---|---|---|---|---|---|---|---|---|
| | Ours | TTS SA | Ours | TTSA | ALSET | ALSET-AC | Ours | $\alpha$-TSCGD | SG-MRL |
| SM | $\mathcal{O}(\epsilon^{-1})$ | $\mathcal{O}(\epsilon^{-1.5})$ | $\tilde{\mathcal{O}}(\epsilon^{-1})$ | $\tilde{\mathcal{O}}(\epsilon^{-1.5})$ | ~ | ~ | $\mathcal{O}(\epsilon^{-1})$ | $\mathcal{O}(\epsilon^{-\frac{N+5}{4}})$ | ~ |
| N-SM | $\mathcal{O}(\epsilon^{-2})$ | ~ | $\tilde{\mathcal{O}}(\epsilon^{-2})$ | $\tilde{\mathcal{O}}(\epsilon^{-2.5})$ | $\tilde{\mathcal{O}}(\epsilon^{-2})$ | $\mathcal{O}(\epsilon^{-2})$ | $\mathcal{O}(\epsilon^{-2})$ | $\mathcal{O}(\epsilon^{-\frac{N+8}{4}})$ | $\mathcal{O}(\epsilon^{-4})$ |
| Merit | ~ | Rate ↑ | ~ | Rate ↑ | Relax | Relax | ~ | Rate ↑ | Rate ↑ |

Table 1: Comparisons with TTS SA [12], TTSA [26], ALSET and ALSET-AC [8], $\alpha$-TSCGD [61] and SG-MRL [14]. Strongly-monotone (SM) and non-strongly-monotone (N-SM) respectively represents the case where the main sequence has strongly-monotone and non-strongly-monotone increments. Rows of SM/N-SM are for the complexity and the row of Merit is for the improvements of this work over the existing work ("Rate ↑" stands for faster rate; "Relax" for relaxed assumptions).

where $\alpha_k, \beta_{k,1}, ..., \beta_{k,N}$ are the step sizes, and $\xi_k, \psi_k^1, \ldots, \psi_k^N$ are random variables. For conciseness, we have used $y_k^0 := x_k$ here. Our goal is to find the unique fixed-points $x^*, y^{1,*}, \ldots, y^{N,*}$ such that

$$v(x^*, y^{1,*}, \ldots, y^{N,*}) = 0, \ h^1(x^*, y^{1,*}) = 0, \ldots, \ h^N(y^{N-1,*}, y^{N,*}) = 0. \tag{6}$$

Observing that in (5), for every $n$, the sequence of $y_k^n$ is coupled with that of $y_k^{n-1}$ and is ultimately coupled with the main sequence $x_k$. Meanwhile the update of $x_k$ also depends on $\{y_k^n\}_{n=1}^N$. Since all sequences in (5) are coupled, (5) is more challenging to analyze than the double-sequence SA.

**Prior art related to multi-sequence SA.** The multilevel stochastic optimization problem [49] and the multilevel SCO problem [1, 58, 63, 47, 65] are closely related to the multi-sequence SA. To tackle the multi-level structure, these recent methods have modified the vanilla multi-sequence SA update to achieve the state-of-the-art complexity and thus their updates are no longer in the form of (5). In contrast, we focus on the multi-sequence SA update in (5). To the best of our knowledge, the only analysis for (5) is [61] where the TTS technique is generalized to multi-time-scale. In [61], the iteration complexity will get worse as the number of sequences $N$ increases.

This gives rise to another interesting question:

**Q2: Is it possible to establish convergence rate independent of the number of sequences?**

In this work, we give affirmative answers to both Questions **Q1** and **Q2**.

**Our contributions.** Specially, by exploiting the *smooth assumption* that can be satisfied in many applications, we show that the vanilla nonlinear SA can run in a *single time scale*! We further prove that the order of the convergence rate is *independent of* the number of sequences $N$! Intuitively, this is possible because when the fixed point $y^{n,*}$ is smooth in $x$, the $y_k^n$-update *converges fast enough* such that its fixed-point residual after one-step update is at the same order as the drift of $y^{n,*}$.

In the context of prior art, our contributions can be summarized as follows (see Table 1).

**C1) Single-timescale analysis for multi-sequence SA.** Different from existing two-timescale analysis [39, 5], we establish a unifying Single-Timescale analysis for SA with multiple coupled sequences that we term **STSA**. When all the sequences have strongly-monotone increments, we improve the $\mathcal{O}(\epsilon^{-1.5})$ iteration complexity for multi-sequence TTS-SA in [12] to $\mathcal{O}(\epsilon^{-1})$. When the main sequence does not have the strongly-monotone increment, we provide the $\mathcal{O}(\epsilon^{-2})$ iteration complexity.

**C2) STSA for stochastic bilevel optimization (SBO).** When applying our generic results to the SBO problem with double-sequence SA, for strongly-concave objective functions, we improve the best-known sample complexity $\tilde{\mathcal{O}}(\epsilon^{-1.5})$ of TTSA in [26] to $\tilde{\mathcal{O}}(\epsilon^{-1})$. For the non-concave objective function, we achieve the same sample complexity $\mathcal{O}(\epsilon^{-2})$ of ALSET while relaxing the bounded upper-level gradient assumption made in [8].

**C3) STSA for stochastic compositional optimization (SCO).** When applying our results to the multi-level SCO problems, we improve the level-dependent sample complexities $\mathcal{O}(\epsilon^{-\frac{N+5}{4}})$ and $\mathcal{O}(\epsilon^{-\frac{N+8}{4}})$ of multi-sequence SA based $\alpha$-TSCGD method in [61] to the level-independent complexities $\tilde{\mathcal{O}}(\epsilon^{-1})$ and $\mathcal{O}(\epsilon^{-2})$, under the strongly-concave and non-concave objective functions, respectively.

**C4) STSA for policy optimization in RL problems.** Moreover, applying our results to the actor-critic method achieves the same $\mathcal{O}(\epsilon^{-2})$ sample complexity of ALSET-AC in [8] while relaxing the unverifiable assumption on the stationary distribution of Markov chains; applying our results to the meta policy gradient improves the $\mathcal{O}(\epsilon^{-4})$ sample complexity of SG-MRL in [14] to $\mathcal{O}(\epsilon^{-2})$.

## 2 Main Results: Convergence of Single-timescale Multi-sequence SA

Before introducing the main results, we will first make some standard assumptions. Throughout the discussion, we define $[N] := \{1, 2, ..., N\}$, $[K] := \{1, 2, ..., K\}$ and $y^0 := x$ for conciseness.

**Assumption 1 (Smoothness of the fixed points)** *For any $n \in [N]$ and $y^{n-1} \in \mathbb{R}^{d_{n-1}}$, there exists a unique $y^{n,*}(y^{n-1}) \in \mathbb{R}^{d_n}$ such that $h^n(y^{n-1}, y^{n,*}(y^{n-1})) = 0$. Moreover, there exist constants $L_{y,n}$ and $L_{y',n}$ such that for any $y^{n-1}, \bar{y}^{n-1} \in \mathbb{R}^{d_{n-1}}$, the following inequalities hold*

$$\|y^{n,*}(y^{n-1}) - y^{n,*}(\bar{y}^{n-1})\| \le L_{y,n}\|y^{n-1} - \bar{y}^{n-1}\|, \tag{7a}$$

$$\|\nabla y^{n,*}(y^{n-1}) - \nabla y^{n,*}(\bar{y}^{n-1})\| \le L_{y',n}\|y^{n-1} - \bar{y}^{n-1}\|. \tag{7b}$$

Due to the change of $y_k^{n-1}$ at each iteration, the solution of $h^n(y_k^{n-1}, y^n) = 0$ with respect to (w.r.t.) $y^n$, that is $y^{n,*}(y_k^{n-1})$, is drifting over consecutive iterations. Given $y_k^{n-1}$, since only one-step of $y_k^n$ update is performed at each iteration, one can only hope to establish convergence of $y_k^n$ if the drift of its optimal solution is controlled in some sense. Assumption 1 ensures both the zeroth-order and first-order drifts are controlled in the same scale of the change of $y_k^{n-1}$. This assumption is satisfied in linear SA [29] and other applications which will be shown later.

Define $v(x) := v\big(x, y^{1,*}(x), y^{2,*}(y^{1,*}(x)), \ldots, y^{N,*}(\ldots y^{2,*}(y^{1,*}(x))\ldots)\big)$. With $y^{1:N}$ as a concise notation for $(y^1, ..., y^N)$, we make the following assumption.

**Assumption 2 (Lipschitz continuity of increments)** *For any $n \in [N]$, $x, \bar{x} \in \mathbb{R}^{d_0}$ and $y^n, \bar{y}^n \in \mathbb{R}^{d_n}$, there exist constants $L_v$, $L_{v,y}$ and $L_{h,n}$ such that the following inequalities hold*

$$\|v(x) - v(\bar{x})\| \le L_v\|x - \bar{x}\|, \quad \|v(x, y^{1:N}) - v(x, \bar{y}^{1:N})\| \le L_{v,y}\sum_{n=1}^{N}\|y^n - \bar{y}^n\|, \tag{8a}$$

$$\|h^n(y^{n-1}, y^n) - h^n(y^{n-1}, \bar{y}^n)\| \le L_{h,n}\|y^n - \bar{y}^n\|. \tag{8b}$$

Define $\mathcal{F}_k$ as the $\sigma$-algebra generated by the random variables in $\{x_i, y_i^{1:N}\}_{i=1}^k$ and $\mathcal{F}_k^n$ as the $\sigma$-algebra generated by $\{x_i, y_i^{1:N}\}_{i=1}^k \cup \{y_{k+1}^n\}$. We make the following assumption on the noises.

**Assumption 3 (Bias and variance)** *Define $\mathcal{F}_k^{N+1} := \mathcal{F}_k$. There exist $\{c_n, \sigma_n\}_{n=0}^N$ such that $\forall k, n$, $\|\mathbb{E}[\xi_k|\mathcal{F}_k^1]\|^2 \le c_0^2\alpha_k$, $\|\mathbb{E}[\psi_k^n|\mathcal{F}_k^{n+1}]\|^2 \le c_n^2\beta_{k,n}$, $\mathbb{E}[\|\xi_k\|^2|\mathcal{F}_k^1] \le \sigma_0^2$ and $\mathbb{E}[\|\psi_k^n\||\mathcal{F}_k^{n+1}] \le \sigma_n^2$.*

Assumption 3 is a generalized version of the bias and variance assumption in stochastic programming [19] or the noise assumption in single-sequence SA [30] to multi-sequence case. Similar assumption has also been made in the existing analysis of double-sequence SA [26]. As will be shown later, when applying STSA to the stochastic optimization problems, the conditional independence between samples of different levels along with the bias and variance conditions will lead to this assumption.

**Assumption 4 (Monotonicity of $h$)** *For $n \in [N]$, $h^n(y^{n-1}, y^n)$ is one-point strongly monotone on $y^{n,*}(y^{n-1})$ given any $y^{n-1}$; that is, there exists constant $\lambda_n > 0$ such that (cf. $h^n(y^{n-1}, y^{n,*}) = 0$)*

$$\langle y^n - y^{n,*}(y^{n-1}), h^n(y^{n-1}, y^n)\rangle \le -\lambda_n\|y^n - y^{n,*}(y^{n-1})\|^2, \ \forall y^n \in \mathbb{R}^{d_n}. \tag{9}$$

Assumption 4 is implied by the standard regularity assumptions in the previous works on TTS linear SA [34, 29], and has also been exploited in the TTS nonlinear SA works; see e.g. [39, 12].

### 2.1 The strongly-monotone case

We first consider the case when the main sequence $x_k$ has strongly-monotone increment.

**Assumption 5 (Monotonicity of $v$)** *Suppose $v(x)$ is one-point strongly monotone on $x^*$; that is, there exists a positive constant $\lambda_0$ such that (cf. $v(x^*) = 0$)*

$$\langle x - x^*, v(x)\rangle \le -\lambda_0\|x - x^*\|^2, \ \forall x \in \mathbb{R}^{d_0}. \tag{10}$$

Same as Assumption 4, Assumption 5 is standard in the previous works on TTS SA [39, 12]. This assumption is a regularity assumption in the case of TTS linear SA; see e.g., [34, Assumption 2.3]. Or in the case of bilevel optimization which will be discussed later, this assumption is satisfied when the objective function is strongly-concave.

Due to space limitation, we directly present the result below and defer the proof to Appendix B.

**Theorem 1** *Consider the sequences generated by* (5). *Suppose Assumptions 1–5 hold. Select step sizes* $\alpha_k = \Theta(\frac{1}{k})$ *and* $\beta_{k,n} = \Theta(\frac{1}{k})$. *It holds for any* $k$ *that*

$$\mathbb{E}\|x_k - x^*\|^2 + \sum_{n=1}^{N} \mathbb{E}\|y_k^n - y^{n,*}(y_k^{n-1})\|^2 = \mathcal{O}\Big(\frac{1}{k}\Big) \tag{11}$$

*where* $\mathcal{O}(\cdot)$ *hides constants in the polynomial of* $N$, *and we have used* $y_k^0 = x_k$ *for convenience. Moreover, for any* $n \in [N]$ *we have*

$$\lim_{k \to \infty} \|x_k - x^*\|^2 = 0 \quad \text{almost surely (a.s.),} \quad \lim_{k \to \infty} \|y_k^n - y^{n,*}(y_k^{n-1})\|^2 = 0 \quad a.s. \tag{12}$$

It is worth noting that with (7a), Theorem 1 also implies the same convergence result for the error metric $\|x_k - x^*\|^2 + \sum_{n=1}^{N} \|y_k^n - y^{n,*}\|^2$, the formal justification of which is deferred to the proof of Theorem 1. It is worth noting that the order of convergence in Theorem 1 is independent of $N$, which is in contrast to the convergence rate that gets worse as $N$ increases [12, 61].

**Remark 1 (Comparison with prior art in multi-sequence SA)** Theorem 1 bridges the gap between the convergence rates of double-sequence linear and nonlinear SA by improving over the $\mathcal{O}(k^{-\frac{2}{3}})$ rate shown in [12] with the additional assumption (7b). As will be shown later, this assumption is satisfied in various applications. Theorem 1 also generalizes the $\mathcal{O}(\frac{1}{k})$ convergence rate in the double-sequence linear SA analysis (e.g., [29]) to the multi-sequence nonlinear SA case.

## 2.2 The non-strongly-monotone case

Some applications of multi-sequence nonlinear SA such as the actor-critic method [32], Assumption 5 does not hold. This motivates us to consider a more general setting in this subsection where $v(x)$ is non-strongly-monotone. Throughout this subsection, we make the following assumption.

**Assumption 6** *Suppose there exists a mapping* $F : \mathbb{R}^{d_0} \mapsto \mathbb{R}$ *such that* $\nabla F(x) = v(x)$. *The sequence of* $\{x_k\}$ *is contained in an open set over which* $F(x)$ *is upper bounded; e.g.* $F(x) \leq C_F$.

As will be shown later, $F(x)$ can be chosen as the objective function when applying SA to maximization problems. Then assumption 6 is standard to ensure the convergence of $x_k$; see e.g. [6].

The following theorem gives the general finite-time convergence result of the nonlinear SA when the main sequence has the non-strongly-monotone increment. The proof is deferred to Appendix C.

**Theorem 2** *Consider the sequences generated by* (5) *for k=[K]. Suppose Assumptions 1–4 & 6 hold. Select* $\alpha_k = \Theta(\frac{1}{\sqrt{K}})$, $\beta_{k,n} = \Theta(\frac{1}{\sqrt{K}})$ *with properly chosen initial step sizes, then it holds that*

$$\frac{1}{K} \sum_{k=1}^{K} \Big( \mathbb{E}\|\nabla F(x_k)\|^2 + \sum_{n=1}^{N} \mathbb{E}\|y_k^n - y^{n,*}(y_k^{n-1})\|^2 \Big) = \mathcal{O}\Big(\frac{1}{\sqrt{K}}\Big), \tag{13}$$

*where* $\mathcal{O}(\cdot)$ *hides problem dependent constants of a polynomial of* $N$, *and we have used* $y_k^0 = x_k$.

Theorem 2 implies a finite-time convergence rate of $\mathcal{O}(K^{-\frac{1}{2}})$, which is independent of the number of sequences $N$. The error metric $\|\nabla F(x_k)\|$ used in Theorem 2 is of interest since it is a general measure of the convergence of $x_k$ widely adopted in many applications of SA, especially when the increment of $x_k$ is not strongly-monotone. Moreover, although we have assumed the existence and uniqueness of $x^*$ in (6), the proof of Theorem 2 does not utilize this fact and thus the theorem applies to the more general case where $x^*$ is not unique or even does not exist.

**Remark 2 (Comparison with stacking all the variables)** *One naive way to establish the convergence of a multi-sequence update is to stack all the variables and view it as one sequence. However, the stacked sequence requires stronger assumptions that are violated in the applications to converge. For one, we would need $v, h^1, ..., h^N$ to be jointly lipschitz continuous w.r.t. the stacked variable $(x, y^1, ..., y^N)$. This condition is violated in, e.g., the application of actor-critic (Section 3.2). The upper-bounded function $F$ can also be difficult to find. As it can be seen later in, e.g., Section 3 that such a $F$ only exists for $x$ and might not exist for the stacked variable $(x, y)$.*

Next we will showcase how the results can be applied to optimization and RL problems.

## 3 Applications to Stochastic Bilevel Optimization

With mappings $f : \mathbb{R}^{d_0} \times \mathbb{R}^{d_1} \mapsto \mathbb{R}$ and $g : \mathbb{R}^{d_0} \times \mathbb{R}^{d_1} \mapsto \mathbb{R}$, consider the following formulation of the bilevel optimization problem:

$$\max_{x \in \mathbb{R}^{d_0}} F(x) := f(x, y^*(x)) := \mathbb{E}_\zeta \big[ f(x, y^*(x); \zeta) \big]$$
$$\text{s.t.} \quad y^*(x) := \arg \min_{y \in \mathbb{R}^{d_1}} g(x, y) := \mathbb{E}_\varphi \big[ g(x, y; \varphi) \big] \tag{14}$$

where $\zeta$ and $\varphi$ are two random variables.

### 3.1 Reduction from the generic STSA results

A popular approach to solving (14) is the gradient-based method [21, 26, 27, 8]. Under some conditions that will be specified later, the gradient of $F(x)$ takes the following form [21]:

$$\nabla F(x) = \nabla_x f(x, y^*(x)) - \nabla_{xy}^2 g(x, y^*(x)) [\nabla_{yy}^2 g(x, y^*(x))]^{-1} \nabla_y f(x, y^*(x)). \tag{15}$$

Computing (15) requires $y^*(x)$, which is often unknown in practice. Instead, one can iteratively update $y_k$ to approach $y^*(x_k)$ while using $y_k$ in place of $y^*(x_k)$ during the computation of (15) [26, 8]. This leads to an update same as that in (5) with $N = 1$, where the mappings are defined as

$$h(x, y) = -\nabla_y g(x, y), \quad \psi_k = -h(x_k, y_k) - \nabla_y g(x_k, y_k; \varphi_k), \tag{16a}$$

$$v(x, y) = \nabla_x f(x, y) - \nabla_{xy}^2 g(x, y)[\nabla_{yy} g(x, y)]^{-1} \nabla_y f(x, y), \tag{16b}$$

$$\xi_k = -v(x_k, y_k) + \nabla_x f(x_k, y_k; \zeta_k) - \nabla_{xy}^2 g(x_k, y_k; \varphi_k') H_k^{yy} \nabla_y f(x_k, y_k; \zeta_k). \tag{16c}$$

Since we only have two sequences, that is $N = 1$, we omit the index $n$ to simplify notations. In (16), $\zeta_k$ is a random variable with the same distribution as that of $\zeta$, and $\varphi_k, \varphi_k'$ have the same distribution as that of $\varphi$. Here $H_k^{yy}$ is a stochastic approximation of the Hessian inverse $[\nabla_{yy} g(x_k, y_k)]^{-1}$. Given $x_k$, when $y_k$ reaches the optimal solution $y^*(x_k)$, it follows from (15) that $v(x_k, y^*(x_k)) = \nabla F(x_k)$.

As being discussed below Assumption 1, the lower-level optimal solution $y^*(x_k)$ is drifting at each iteration. Under the Lipschitz continuity assumption of $y^*(x)$, the drifting $\|y^*(x_{k+1}) - y^*(x_k)\|$ scales with $\|x_{k+1} - x_k\|$ which ultimately scales with $\|\nabla F(x_k)\|$. To control the drift scale, former analysis heavily relies on the condition that $\|\nabla F(x_k)\|$ can be bounded for any $k$. In SBO, this means to either make a strong assumption on the Lipschitz continuity of $f(x, y)$ w.r.t. $(x, y)$, which leads to the Lipschitz continuity of $F(x)$ and the boundedness of $\|\nabla F(x_k)\|$ [8]; or to introduce projection in (16) to forcibly confine $x_k$ in a compact set [26], all of which greatly narrow the range of application. We will show that neither of these the conditions is needed by applying our generic results to SBO.

**Lemma 1 (Verifying assumptions of STSA)** *Consider the following conditions*

- *(a) For any $x \in \mathbb{R}^{d_1}$, $g(x, y)$ is strongly convex w.r.t. $y$ with modulus $\lambda_1 > 0$.*

- *(b) There exist constants $L_{xy}, l_{xy}, l_{yy}$ such that $\nabla_y g(x, y)$ is $L_{xy}$-Lipschitz continuous w.r.t. $x$; $\nabla_y g(x, y)$ is $L_h$-Lipschitz continuous w.r.t. $y$. $\nabla_{xy} g(x, y)$, $\nabla_{yy} g(x, y)$ are respectively $l_{xy}$-Lipschitz and $l_{yy}$-Lipschitz continuous w.r.t. $(x, y)$.*

- *(c) There exist constants $l_{fx}, l_{fy}, l'_{fy}, l_y$ such that $\nabla_x f(x, y)$ and $\nabla_y f(x, y)$ are respectively $l_{fx}$ and $l_{fy}$ Lipschitz continuous w.r.t. $y$; $\nabla_y f(x, y)$ is $l'_{fy}$-Lipschitz continuous w.r.t. $x$; $f(x, y)$ is $l_y$-Lipschitz continuous w.r.t. $y$.*

*(d) $F(x)$ satisfies the restricted secant inequality: There exists a constant $\lambda_0 > 0$ such that $\langle \nabla F(x), x - x^* \rangle \leq -\lambda_0 \|x - x^*\|^2$, where $x^* := \arg\max_{x \in \mathbb{R}^{d_1}} F(x)$.*

*(e) For any $k$, there exist constants $c_0, c_1$ such that $\|\mathbb{E}[\xi_k | \mathcal{F}_k^1]\|^2 \leq c_0^2 \alpha_k$ and $\|\mathbb{E}[\psi_k | \mathcal{F}_k]\|^2 \leq c_1^2 \beta_k$; there exist constants $\sigma_0, \sigma_1$ such that $\mathbb{E}[\|\xi_k\|^2 | \mathcal{F}_k^1] \leq \sigma_0^2$ and $\mathbb{E}[\|\psi_k\|^2 | \mathcal{F}_k] \leq \sigma_1^2$.*

*(f) There exists a constant $C_F$ such that $F(x) \leq C_F$.*

*We use $a \Rightarrow b$ to indicate that $a$ is a sufficient condition of $b$. Then we have*

$$(a)\&(b) \Rightarrow \text{ Assumption 1}; \quad (a)-(c) \Rightarrow \text{ Assumption 2}; \quad (e) \Rightarrow \text{ Assumption 3};$$
$$(a) \Rightarrow \text{ Assumption 4}; \quad (d) \Rightarrow \text{ Assumption 5}; \quad (f) \Rightarrow \text{ Assumption 6}.$$

The conditions listed above are commonly adopted in the literature [21, 26, 8]. It is worth noting that Lemma 1 does not need the $L_{xy}$-Lipschitz continuity condition of $f(x, y)$ w.r.t. $(x, y)$. This Lipschitz condition along with the $L_y$-Lipschitz continuity of $y^*(x)$, which is implied by the standard conditions in Lemma 1, further leads to the Lipschitz continuity of $F(x)$:

$$|F(x) - F(x')| \leq L_{xy}(\|x - x'\| + \|y^*(x) - y^*(x')\|) \leq L_{xy}(L_y + 1)\|x - x'\|. \quad (17)$$

Although it is rather restrictive, this condition has been used in the previous work when $F(x)$ is not strongly-concave. While our analysis does not need this condition. Lastly, condition $(e)$ is guaranteed by using independent samples in the upper- and lower-level along with [21, Algorithm 3] to obtain a good $H_k^{yy}$, which takes $\Omega(-\log \alpha_k)$ samples per iteration. With Lemma 1, we have the following corollary regarding the convergence of (16).

**Corollary 1 (STSA for SBO)** *Consider the STSA sequences with the update in (16). Under Conditions (a)–(e), Theorem 1 holds; that is, with $\alpha_k = \Theta(\frac{1}{k})$ and $\beta_k = \Theta(\frac{1}{k})$ we have*

$$\mathbb{E}\|x_k - x^*\|^2 + \mathbb{E}\|y_k - y^*(x_k)\|^2 = \mathcal{O}\left(\frac{1}{k}\right), \quad (18a)$$

$$\lim_{k \to \infty} \|x_k - x^*\|^2 = 0 \quad \text{and} \quad \lim_{k \to \infty} \|y_k - y^*(x_k)\|^2 = 0 \text{ a.s.} \quad (18b)$$

*Under Conditions (a)–(c), (e) and (f), Theorem 2 holds; i.e., with $\alpha_k = \Theta(\frac{1}{\sqrt{K}})$, $\beta_k = \Theta(\frac{1}{\sqrt{K}})$, we have*

$$\frac{1}{K} \sum_{k=1}^{K} \left( \mathbb{E}\|\nabla F(x_k)\|^2 + \mathbb{E}\|y_k - y^*(x_k)\|^2 \right) = \mathcal{O}\left(\frac{1}{\sqrt{K}}\right). \quad (19)$$

**Remark 3 (Comparison with prior art in SBO)** When $F(x)$ is strongly concave, Corollary 1 implies the sample complexity of $\mathcal{O}(\epsilon^{-1}\log \epsilon^{-1})$, which improves over the best-known sample complexity $\mathcal{O}(\epsilon^{-1.5}\log \epsilon^{-1})$ in [26]. Different from [26], we do not need the projection of $x_k$ to a compact set. When $F(x)$ is non-concave, corollary 1 suggests a sample complexity of $\mathcal{O}(\epsilon^{-2}\log \epsilon^{-1})$, which is the same as the state-of-art complexity established in [8]. Corollary 1 improves the result in [8] in two major aspects: 1) it relaxes the Lipschitz continuity assumption on $f(x, y)$; and, 2) an alternating update is adopted in [8] to ensure stability, while some applications of SBO only allow simultaneous updates. Corollary 1 applies to those cases and thus has a broader range of application.

## 3.2 Application to advantage actor-critic

RL problems are often modeled as a MDP described by $\mathcal{M} = \{\mathcal{S}, \mathcal{A}, \mathcal{P}, r, \gamma\}$, where $\mathcal{S}$ is the state space, $\mathcal{A}$ is the action space; $\mathcal{P}(s'|s, a)$ is the probability of transitioning to $s' \in \mathcal{S}$ given $(s, a) \in \mathcal{S} \times \mathcal{A}$; $r(s, a) \in [0, 1]$ is the reward associated with $(s, a)$; and $\gamma \in (0, 1)$ is a discount factor. A policy $\pi$ maps $\mathcal{S}$ to a distribution over $\mathcal{A}$, and we use $\pi(a|s)$ to denote the probability of choosing $a$ under $s$. Given a policy $\pi$, we define the value functions as $V_\pi(s) := \mathbb{E}_\pi\left[ \sum_{t=0}^{\infty} \gamma^t r(s_t, a_t) \mid s_0 = s \right]$, where $\mathbb{E}_\pi$ is taken over the trajectory $(s_0, a_0, s_1, a_1, \dots)$ generated under policy $\pi$ and transition kernel $\mathcal{P}$. With $\rho$ denoting the initial state distribution, the discounted visitation distribution induced by policy $\pi$ is defined via $d_\pi(s, a) = (1 - \gamma) \sum_{t=0}^{\infty} \gamma^t \mathbf{Pr}_\pi(s_t = s \mid s_0 \sim \rho) \pi(a|s)$. To overcome the difficulty of learning a function, we parameterize the policy with $x \in \mathbb{R}^{d_0}$, and solve

$$\max_{x \in \mathbb{R}^{d_0}} F(x) := (1 - \gamma) \mathbb{E}_{s \sim \rho}[V_{\pi_x}(s)]. \quad (20)$$

To solve for (20), a popular method is the actor-critic (AC) method [32]. The actor-critic algorithm with linear critic function is a special case of (5). Specifically, the critic variable $y$ is updated with

$$h(x,y) = \mathbb{E}_{s \sim \mu_{\pi_x}, a \sim \pi_x, s' \sim \mathcal{P}}[\phi(s)(\gamma\phi(s') - \phi(s))^\top]y + \mathbb{E}_{s \sim \mu_{\pi_x}, a \sim \pi_x}[r(s,a)\phi(s)],$$
$$\psi_k = -h(x_k, y_k) + \phi(s_k)(\gamma\phi(s'_k) - \phi(s_k))^\top y + r(s_k, a_k)\phi(s_k), \tag{21}$$

where $\mu_{\pi_x}$ is the stationary distribution of the Markov chain induced by $\pi_x$, $\phi(s) \in \mathbb{R}^{d_1}$ is the feature vector encoding state $s$ and the sample $(s_k, a_k, s'_k)$ is returned by some sampling protocol. Under some regularity conditions, it is known that there exists a unique $y^*(x)$ such that $h(x, y^*(x)) = 0$ [2]. The actor variable $x$ is then updated with

$$v(x,y) = \mathbb{E}_{s,a \sim d_{\pi_x}, s' \sim \mathcal{P}}[(r(s,a) + (\gamma\phi(s') - \phi(s))^\top y)\nabla \log \pi_x(a|s)],$$
$$\xi_k = -v(x_k, y_k) + r(\bar{s}_k, \bar{a}_k) + \gamma(\phi(\bar{s}'_k) - \phi(\bar{s}_k))^\top y_k \nabla \log \pi_{x_k}(\bar{a}_k|\bar{s}_k). \tag{22}$$

For the AC update in (21) and (22), Assumption 2–4 and 6 or their sufficient conditions have been explored in the RL context by previous works [57]. However, the smoothness of $y^*(x)$ in Assumption 1, which is the key condition leading to a faster convergence rate, has yet been verified. With the same conditions as those adopted in [57], we prove that $y^*(x)$ is indeed smooth.

**Lemma 2** *Consider the AC update in* (21)-(22). *Under the standard conditions specified in Appendix E, $y^*(x)$ is differentiable and there exists $L_{y'} > 0$ such that $\|\nabla y^*(x) - \nabla y^*(x')\| \le L_{y'}\|x - x'\|$.*

As a comparison, the above condition was directly assumed in [8], while we provide a formal justification for Lemma 2 in this work. With detailed verification of all assumptions deferred to Appendix E, we then directly present the theorem regarding the convergence of AC.

**Theorem 3 (Complexity of AC)** *Consider the AC update* (21)-(22). *Under the standard conditions specified in Appendix E, Theorem 2 holds; that is, with $\alpha_k = \Theta(\frac{1}{\sqrt{K}})$ and $\beta_k = \Theta(\frac{1}{\sqrt{K}})$, we have*

$$\frac{1}{K}\sum_{k=1}^{K}\left(\mathbb{E}\|\nabla F(x_k)\|^2 + \mathbb{E}\|y_k - y^*(x_k)\|^2\right) = \mathcal{O}\left(\frac{1}{\sqrt{K}}\right). \tag{23}$$

In [8, 57], the projection step is adopted in the $y_k$ update to ensure that $\|y_k\| < \infty, \forall k$. Since the projection radius is unknown in practice, adopting the projection is essentially assuming that $\|y_k\|$ can be bounded for any $k$, which is quite strong. Theorem 3 holds without this projection.

## 4 Applications to Stochastic Compositional Optimization

Define mappings $f^n : \mathbb{R}^{d_n} \mapsto \mathbb{R}^{d_{n+1}}$ for $n = 0, 1, ..., N$ with $d_{N+1} = 1$. The multi-level stochastic compositional problem can be formulated as

$$\max_{x \in \mathbb{R}^{d_0}} F(x) := f^N(f^{N-1}(\dots f^0(x)\dots) \quad \text{with} \quad f^n(x) := \mathbb{E}_{\zeta^n}[f^n(x; \zeta^n)], n = 0, 1, ..., N \tag{24}$$

where $\zeta^0, \zeta^1, \dots, \zeta^N$ are random variables. Here we slightly overload the notation and use $f^n(x; \zeta^n)$ to represent the stochastic version of the mapping.

### 4.1 Reduction from the generic STSA results

To solve the problem in (24), a natural scheme is to use the stochastic gradient descent method with the gradient given by

$$\nabla F(x) = \nabla f^0(x)\nabla f^1(f^0(x))...\nabla f^N(f^{N-1}(...f^0(x)...)) \tag{25}$$

where we use $\nabla f^n(f^{n-1}(\dots f^0(x)\dots)) = \nabla f^n(x)|_{x = f^{n-1}(...f^0(x)...)}$. To obtain a stochastic estimator of $\nabla F(x)$, we will need to obtain the stochastic estimators for $\nabla f^n(f^{n-1}(...f^0(x)...))$ for each $n$. For example, when $n = 1$, one will need the estimator of $\nabla f^1(\mathbb{E}_{\zeta^0}[f^0(x; \zeta^0)])$. However, due to the possible non-linearity of $\nabla f^1(\cdot)$, the natural candidate $\nabla f^1(f^0(x; \zeta^0))$ is not an unbiased

estimator of $\nabla f^1(\mathbb{E}_{\zeta^0}[f^0(x;\zeta^0)])$. To tackle this issue, a popular method is to directly approximate the mean $\mathbb{E}_{\zeta^n}[f^n(\cdot;\zeta^n)]$ with a tracking variable $y^n \in \mathbb{R}^{d_n}$ for $n = 0, 1, ..., N$, see e.g., [61].

The update of $y^n$ is then a special case of the SA update in (5) with the generic mapping defined as

$$h^n(y^{n-1}, y^n) = f^{n-1}(y^{n-1}) - y^n, \quad \psi_k^n = -h^n(y_k^{n-1}, y_k^n) + f^{n-1}(y_k^{n-1}; \zeta_k^{n-1}) - y_k^n \quad (26)$$

where $\zeta_k^0, \ldots, \zeta_k^N$ have the same distributions as that of $\zeta^0, \ldots, \zeta^N$ respectively. It is then clear that each $y_k^n$ has a unique fixed-point $y_k^{n,*} = f^{n-1}(y_k^{n-1})$, and thus $y_k^n$ can be viewed as an approximation of $f^n(y_k^{n-1})$. With these approximations, variable $x$ is updated in the form of (5) by defining

$$v(x, y^1, \ldots, y^N) = \nabla f^0(x) \nabla f^1(y^1)...\nabla f^N(y^N),$$
$$\xi_k = -v(x_k, y_k^1, \ldots, y_k^N) + \nabla f^0(x_k; \hat{\zeta}_k^0) \cdots \nabla f^N(y_k^N; \zeta_k^N) \quad (27)$$

where $\hat{\zeta}_k^0$ has the same distribution as that of $\zeta^0$. It is clear that when every $y_k^n$ reaches its fixed-point $y_k^{n,*}$, it follows from (25) that $v(x_k, y_k^{1,*}, ..., y_k^{N,*}) = \nabla F(x_k)$, which indicates that the expected update direction of $x_k$ in (27) is $\nabla F(x_k)$.

Next we provide a lemma that summarizes the sufficient conditions of Assumption 1–6. The listed conditions are standard in the stochastic compositional optimization literature [61, 7].

**Lemma 3 (Verifying assumptions of STSA)**  *Consider the following conditions*

- *(g) Given any $n \in [N] \cup \{0\}$, there exist positive constants $L_{y,n}$ and $L_{y',n}$ such that the mapping $f^n(\cdot)$ is $L_{y,n}$-Lipschitz continuous and $L_{y',n}$-smooth.*

- *(h) Given $\mathcal{F}_k$, for any $n \in [N]$: $f^n(y_k^{n-1}; \zeta_k^n)$ and $\nabla f^n(y_k^{n-1}; \zeta_k^n)$ are respectively the unbiased estimators of $f^n(y_k^{n-1})$ and $\nabla f^n(y_k^{n-1})$ with bounded variance; $f^0(x_k; \hat{\zeta}_k^0)$ and $\nabla f^0(x_k; \hat{\zeta}_k^0)$ are respectively the unbiased estimators of $f^0(x_k)$ and $\nabla f^0(x_k)$ with bounded variance.*

- *(i) Per iteration $k$, $\hat{\zeta}_k^0, \zeta_k^0, \zeta_k^1, \ldots, \zeta_k^N$ are conditionally independent of each other given $\mathcal{F}_k$.*

- *(j) Function $F(x)$ satisfies the restricted secant inequality: There exists a constant $\lambda_0 > 0$ such that $\langle \nabla F(x), x - x^* \rangle \leq -\lambda_0 \|x - x^*\|^2$, where $x^* := \arg\max_{x \in \mathbb{R}^{d_1}} F(x)$.*

- *(k) There exists a constant $C_F$ such that $F(x) \leq C_F$.*

*We use $a \Rightarrow b$ to indicate that $a$ is a sufficient condition of $b$. Then we have*

$$(g) \Rightarrow \text{Assumption 1 and 2}; \quad (h) \text{ and } (i) \Rightarrow \text{Assumption 3}; \quad (j) \Rightarrow \text{Assumption 5};$$
$$(k) \Rightarrow \text{Assumption 6}; \quad \text{Assumption 4 holds for (26)}.$$

With Lemma 3, we can directly arrive at the following corollary on the convergence of the stochastic compositional optimization method.

**Corollary 2 (STSA for multi-level SCO)**  *Consider the STSA sequences generated by (26)-(27). Under Conditions (g)–(j), Theorem 1 holds. Under Conditions (g)–(i) and (k), Theorem 2 holds.*

**Remark 4 (Comparison with prior art in SCO)**  *Corollary 2 establishes the sample complexity of $\mathcal{O}(\epsilon^{-1})$ for the strongly monotone case and the complexity of $\mathcal{O}(\epsilon^{-2})$ for the non-monotone case, which are both independent of $N$. This improves over the $\mathcal{O}(\epsilon^{-\frac{N+5}{4}})$ complexity for the strongly concave case and the $\mathcal{O}(\epsilon^{-\frac{N+8}{4}})$ complexity for the non-concave case shown in [61]. There are other works that establish the same complexity as that in Corollary 2, but they require modification to the basic SA update (26) and (27) to achieve acceleration; see e.g., [7, 1, 47].*

### 4.2  Application to model-agnostic meta policy gradient

Consider a set of MDPs $\{\mathcal{M}_i\}_{i=1}^M$ with $\mathcal{M}_i = \{\mathcal{S}, \mathcal{A}, \mathcal{P}_i, r_i, \gamma\}$. The MDPs model a set of RL tasks that share the same state-action space while having different transition kernels $\mathcal{P}_i$ and reward functions $r_i$. To better compare with the previous work [14], we consider the finite-horizon objective function with the policy $\pi$ parametrized by $x \in \mathbb{R}^{d_0}$: $F_i(x) := \mathbb{E}_{\zeta \sim \pi_x}\left[\sum_{t=0}^H \gamma^t r_i(s_t, a_t)|\rho_i, \mathcal{P}_i\right],$

where $H \in \mathbb{N}^+$ is the horizon, and $\mathbb{E}_{\zeta \sim \pi_x}$ is taken over the trajectory $\zeta := (s_0, a_0, s_1, a_1, ..., s_H, a_H)$ generated under policy $\pi_x$, initial distribution $\rho_i$ and transition kernels $\mathcal{P}_i$.

The goal of MAMPG is to find an initial policy $\pi_x$ that can achieve good performance in new tasks by performing a few policy gradient steps [15, 14]. In the case where $N$ steps of gradient update are performed, the problem of finding an initial policy parameter $x$ can be formulated as

$$\max_{x \in \mathbb{R}^{d_0}} F(x) := \frac{1}{M} \sum_{i=1}^{M} F_i(\tilde{x}_i^N(x)) \ \text{ with } \tilde{x}_i^{n+1} = \tilde{x}_i^n + \eta \nabla F_i(\tilde{x}_i^n), \ n = 0, 1, ..., N-1, \quad (28)$$

where $x$ is the shared initial policy parameter, i.e., $\tilde{x}_i^0 = x$ for any task $i$ and $\tilde{x}_i^N(x)$ is the parameter after running $N$ steps of gradient ascent with respect to $F_i$ starting from $x$.

**Solving** (28) **with SCO method.** The MAMPG problem in (28) can be solved by the stochastic compositional optimization method introduced before. In order to get $\nabla F(x)$, one will need $\nabla F_i(\tilde{x}_i^N(x))$ for each task $i$. Observe that $F_i(\tilde{x}_i^N(x))$ can be written as a compositional function:

$$F_i(\tilde{x}_i^N(x)) = f_i^N(f_i^{N-1}(\ldots f_i^0(x) \ldots)) \ \text{ with } \ f_i^n(x) := x + \eta \nabla F_i(x), \ n = 0, ..., N-1, \quad (29)$$

where $f_i^N(x) = F_i(x)$. In order to approximate $\nabla F_i(\tilde{x}_i^N(x))$, we can follow the discussion in Section 4 and introduce tracking variables $y_i^n \in \mathbb{R}^{d_0}$ for $n \in [N]$ which are updated as follows

$$y_{k+1,i}^n = y_{k,i}^n - \beta_{k,n}(y_{k,i}^n - f_i^{n-1}(y_k^{n-1}; \zeta_{k,i}^{n-1})), \ n = 0, 1, ..., N-1 \quad (30)$$

where we define $f_i^n(\cdot; \zeta)$ as a stochastic approximation of $f_i^n(\cdot)$ with random trajectory $\zeta$. Then we estimate $\nabla F_i(\tilde{x}_i^N(x))$ by $\hat{\nabla} F_{i,k}$ defined as

$$\hat{\nabla} F_{i,k} := \nabla f_i^0(x; \hat{\zeta}_{k,i}^0) \nabla f_i^1(y_{k,i}^1; \zeta_{k,i}^1) \cdots \nabla f_i^N(y_{k,i}^N; \zeta_{k,i}^N). \quad (31)$$

To obtain an estimation of $\nabla F(x)$, we need $\hat{\nabla} F_{i,k}$ for each $i \in \{1, 2, ..., M\}$. Thus we do (30) for each $i$. With $\{\hat{\nabla} F_{i,k}\}_{i=1}^M$, the initial policy is updated as $x_{k+1} = x_k + \alpha_k \frac{1}{M} \sum_{i=1}^{M} \hat{\nabla} F_{i,k}$.

**Reduction from the generic results.** Let $y^n \in \mathbb{R}^{d_n M}$ be a concatenation of $y_i^n$ for $i \in \{1, 2, ..., M\}$. With $\zeta_k^n := \{\zeta_{k,i}^n\}_{i=1}^M$, let $f^n(y_k^n; \zeta_k^n)$ be a concatenation of $f_i^n(y_{k,i}^n; \zeta_{k,i}^n)$ for $i \in \{1, 2, ..., M\}$. Then we can write the tracking variable update of all tasks jointly in the form of (5a), that is

$$h^n(y^{n-1}, y^n) = f^{n-1}(y^{n-1}) - y^n, \ \ \psi_k^n = -h^n(y_k^{n-1}, y_k^n) + f^{n-1}(y_k^{n-1}; \zeta_k^{n-1}) - y_k^n. \quad (32)$$

The initial policy update is a special case of (5b), that is

$$v(x, y^1, ..., y^N) = \frac{1}{M} \sum_{i=1}^{M} \nabla f_i^0(x)...\nabla f_i^N(y_i^N), \ \ \xi_k = -v(x_k, y_k^1, ..., y_k^N) + \frac{1}{M} \sum_{i=1}^{M} \hat{\nabla} F_{i,k}. \quad (33)$$

Due to space limitation, we directly give the result below and defer the proof to Appendix G.

**Theorem 4 (Complexity of MAMPG)** *Consider the STSA sequences generated by the MAMPG update in* (32) *and* (33)*. Under some standard conditions specified in Appendix G, Theorem 2 holds.*

Theorem 4 implies a sample complexity of $\mathcal{O}(\epsilon^{-2})$ to achieve the $\epsilon$-stationary initial policy, which improves over the $\mathcal{O}(\epsilon^{-4})$ sample complexity in [14]. Moreover, Theorem 4 holds for any $N \geq 1$ in (28), while the method in [14] only applies to the case $N = 1$.

## 5   Conclusions

In this work, we consider the general nonlinear SA with multiple coupled sequences, and study its non-asymptotic performance. Different from the dominating two-timescale SA analysis, we are particularly interested in under which conditions, single-timescale analysis can be applied to nonlinear SA with multiple coupled sequences. When all the sequences have strongly monotone increments, we establish the iteration complexity of $\mathcal{O}(\epsilon^{-1})$. When the main sequence is not strongly-monotone, we establish the iteration complexity of $\mathcal{O}(\epsilon^{-2})$. We then apply our generic SA analysis to stochastic bilevel and compositional optimization and improve their existing results. Specifically, we improve the state-of-the-art convergence rate of: 1) the SBO method and its application to the AC method; and, 2) the multi-level SCO method and its application to the MAMPG method.

## Acknowledgments

This work was partially supported by National Science Foundation MoDL-SCALE Grant 2134168, Amazon Research Awards, and the Rensselaer-IBM AI Research Collaboration (`http://airc.rpi.edu`), part of the IBM AI Horizons Network (`http://ibm.biz/AIHorizons`). We thank anonymous reviewers for their valuable feedback on improving the current paper.

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
