# Table of Contents

# A   Additional related works

In this section, we review the prior art on the applications of multi-sequence SA.

**Gradient-based bilevel optimization.** The bilevel optimization was first introduced in [51]. Recently, the gradient-based bilevel optimization methods have gained growing popularity [48, 16, 22, 38]. The finite-time convergence of the double-loop bilevel optimization methods has been studied in some previous works; see e.g., [21, 27]. Later, [26] proved the finite-time convergence rate for the single-loop two time-scale bilevel optimization method, which was then improved by [8] to the optimal rate with additional assumptions and a more refined analysis. There are also other works that incorporate momentum to accelerate the convergence; see e.g., [31, 24, 60]. After our initial conference submission, we have also noticed some concurrent works that are relevant to this work [9, 23, 36]. Specifically, [9] proposed a SBO method with the variance-reduction technique and achieved optimal rate. And [23] proposed a SBO method that achieves the optimal rate without warm-start. The algorithms in [9, 23] are not a case of the SA update discussed in this work and thus its analysis is not applicable to our problem. Lastly, [36] proposed a single-loop SBO method without Hessian inverse, but it required the bounded-gradient assumption which is not needed in this work.

**Actor-critic method.** After its frist introduction in [33], the finite-sample guarantee for the AC algorithm has been established in [62, 35, 17] with i.i.d. sampling. In [45], the finite-time convergence rate has been established for the nested-loop AC under the Markovian setting, which was later improved improved by [59]. On the other hand, the finite-time convergence of two-timescale AC has been studied in [57] under Markovian sampling and [26, 8] under i.i.d. sampling.

**Gradient-based stochastic compositional optimization.** The two time-scale stochastic compositional optimization method was proposed in [55, 56]. Due to the two time-scale step sizes choice, the convergence rate of [55, 56] is slower than that of the SGD. In order to achieve acceleration, [20, 7, 47, 1] have modified the basic update in [55, 61] and successfully established the convergence rate same as that of SGD. Concurrent to this work, [28] proposed a variance-reduced SCO method

that achieved the optimal rate under variance-reduction. While this work focuses on establishing an optimal rate for the SA update without having diminishing variance. Due to the difference in update scheme, their analysis is not directly applicable to our case.

## B Proof of Theorem 1

### B.1 Analysis of the lower-level sequences

For brevity, we define the shorthand notations $y_k^{n,*} := y^{n,*}(y_k^{n-1})$ with $y_k^{1,*} := y^{1,*}(x_k)$. Also, we write $\mathbb{E}[\cdot|\mathcal{F}_k]$ as $\mathbb{E}_k[\cdot]$ for brevity.

**One-step contraction of lower-level sequences.** With $y_k^0 = x_k$, it holds for any $n \in [N]$ that

$$\mathbb{E}_k\|y_{k+1}^n - y_k^{n,*}\|^2 = \|y_k^n - y_k^{n,*}\|^2 + 2\beta_{k,n}\mathbb{E}_k\langle y_k^n - y_k^{n,*}, h^n(y_k^{n-1}, y_k^n) + \psi_k^n\rangle + \mathbb{E}_k\|y_{k+1}^n - y_k^n\|^2. \tag{34}$$

The second term in (34) can be bounded as

$$\mathbb{E}_k\langle y_k^n - y_k^{n,*}, h^n(y_k^{n-1}, y_k^n) + \psi_k^n\rangle = \langle y_k^n - y_k^{n,*}, h^n(y_k^{n-1}, y_k^n))\rangle + \langle y_k^n - y_k^{n,*}, \mathbb{E}_k[\psi_k^n]\rangle$$

$$\leq -\lambda_n\|y_k^n - y_k^{n,*}\|^2 + \|y_k^n - y_k^{n,*}\|\|\mathbb{E}_k[\psi_k^n]\|$$

$$\leq -\lambda_n\|y_k^n - y_k^{n,*}\|^2 + \frac{\lambda_n}{4}\|y_k^n - y_k^{n,*}\|^2 + \frac{1}{\lambda_n}\|\mathbb{E}_k[\psi_k^n]\|^2$$

$$\leq -\frac{3\lambda_n}{4}\|y_k^n - y_k^{n,*}\|^2 + \frac{c_n^2}{\lambda_n}\beta_{k,n}, \tag{35}$$

where the first inequality follows from the strong monotonicity of $h(y^{n-1}, y^n)$ in Assumption 4, the second inequality follows from the Young's inequality, and the last inequality follows from the bias of the increment $\psi_k^n$ in Assumption 3.

The third term in (34) can be bounded as

$$\mathbb{E}_k\|y_{k+1}^n - y_k^n\|^2 \leq 2\beta_{k,n}^2\big(\|h^n(y_k^{n-1}, y_k^n)\|^2 + \sigma_n^2\big) \leq 2L_{h,n}^2\beta_{k,n}^2\|y_k^n - y_k^{n,*}\| + 2\sigma_n^2\beta_{k,n}^2 \tag{36}$$

where the last inequality follows from Assumption 2 which gives

$$\|h^n(y_k^{n-1}, y_k^n)\| = \|h^n(y_k^{n-1}, y_k^n) - \underbrace{h^n(y_k^{n-1}, y_k^{n,*}(y_k^{n-1}))}_{=0}\| \leq L_{h,n}\|y_k^n - y_k^{n,*}(y_k^{n-1})\|. \tag{37}$$

Collecting the upper bounds in (35) and (36) yields

$$\mathbb{E}_k\|y_{k+1}^n - y_k^{n,*}\|^2 \leq (1 - \frac{3}{2}\lambda_n\beta_{k,n} + 2L_{h,n}^2\beta_{k,n}^2)\|y_k^n - y_k^{n,*}\|^2 + 2(\sigma_n^2 + c_n^2\lambda_n^{-1})\beta_{k,n}^2$$

$$\leq (1 - \lambda_n\beta_{k,n})\|y_k^n - y_k^{n,*}\|^2 + 2(\sigma_n^2 + c_n^2\lambda_n^{-1})\beta_{k,n}^2, \tag{38}$$

where the last inequality is due to the choice of step size that satisfies $2L_{h,n}^2\beta_{k,n}^2 \leq \frac{\lambda_n}{2}\beta_{k,n}$.

**Bounding the drifting optimality gap.** For any $n \geq 1$, we have

$$\|y_{k+1}^n - y_{k+1}^{n,*}\|^2 = \|y_{k+1}^n - y_k^{n,*}\|^2 + 2\langle y_k^{n,*} - y_{k+1}^n, y_{k+1}^{n,*} - y_k^{n,*}\rangle + \|y_k^{n,*} - y_{k+1}^{n,*}\|^2. \tag{39}$$

**(1) When $n \geq 2$.** By the mean-value theorem, for some $\hat{y}_{k+1}^{n-1} = ay_k^{n-1} + (1-a)y_{k+1}^{n-1}, a \in [0,1]$, the second term in (39) can be rewritten as

$$\langle y_k^{n,*} - y_{k+1}^n, y_{k+1}^{n,*} - y_k^{n,*}\rangle = \langle y_k^{n,*} - y_{k+1}^n, \nabla y^{n,*}(\hat{y}_{k+1}^{n-1})^\top(y_{k+1}^{n-1} - y_k^{n-1})\rangle$$

$$= \langle y_k^{n,*} - y_{k+1}^n, \beta_{k,n-1}\nabla y^{n,*}(\hat{y}_{k+1}^{n-1})^\top h^{n-1}(y_k^{n-2}, y_k^{n-1})\rangle$$

$$+ \langle y_k^{n,*} - y_{k+1}^n, \beta_{k,n-1}\nabla y^{n,*}(\hat{y}_{k+1}^{n-1})^\top \psi_k^{n-1}\rangle. \tag{40}$$

The first term in the right-hand side (RHS) of (40) can be bounded as

$$\langle y_k^{n,*} - y_{k+1}^n, \beta_{k,n-1}\nabla y^{n,*}(\hat{y}_{k+1}^{n-1})^\top h^{n-1}(y_k^{n-2}, y_k^{n-1})\rangle$$

$$\leq L_{y,n}\beta_{k,n-1}\|y_k^{n,*} - y_{k+1}^n\|\|h^{n-1}(y_k^{n-2}, y_k^{n-1})\|$$

$$\leq L_{y,n}L_{h,n-1}\beta_{k,n-1}\|y_k^{n,*} - y_{k+1}^n\|\|y_k^{n-1} - y_k^{n-1,*}\|$$

$$\leq \frac{2L_{y,n}^2L_{h,n-1}^2}{\lambda_{n-1}}\beta_{k,n-1}\|y_k^{n,*} - y_{k+1}^n\|^2 + \frac{\lambda_{n-1}}{8}\beta_{k,n-1}\|y_k^{n-1} - y_k^{n-1,*}\|^2 \tag{41}$$

where the second inequality follows from

$$\|h^{n-1}(y_k^{n-2}, y_k^{n-1})\| = \|h^{n-1}(y_k^{n-2}, y_k^{n-1}) - h^{n-1}(y_k^{n-2}, y_k^{n-1,*})\|$$
$$\leq L_{h,n-1} \|y_k^{n-1} - y_k^{n-1,*}\|. \tag{42}$$

The second term in the RHS of (40) can be further decomposed into

$$\langle y_k^{n,*} - y_{k+1}^n, \beta_{k,n-1} \nabla y^{n,*}(\hat{y}_{k+1}^{n-1})^\top \psi_k^{n-1} \rangle$$
$$= \langle y_k^{n,*} - y_{k+1}^n, \beta_{k,n-1}(\nabla y^{n,*}(\hat{y}_{k+1}^{n-1}) - \nabla y^{n,*}(y_k^{n-1}))^\top \psi_k^{n-1} \rangle$$
$$+ \langle y_k^{n,*} - y_{k+1}^n, \beta_{k,n-1} \nabla y^{n,*}(y_k^{n-1})^\top \psi_k^{n-1} \rangle. \tag{43}$$

Taking expectation on the first term in the RHS of (43) leads to

$$\mathbb{E}_k \langle y_k^{n,*} - y_{k+1}^n, \beta_{k,n-1}(\nabla y^{n,*}(\hat{y}_{k+1}^{n-1}) - \nabla y^{n,*}(y_k^{n-1}))^\top \psi_k^{n-1} \rangle$$
$$\leq L_{y',n}\beta_{k,n-1}\mathbb{E}_k\big[\|y_k^{n,*} - y_{k+1}^n\|\|\hat{y}_{k+1}^{n-1} - y_k^{n-1}\|\|\psi_k^{n-1}\|\big]$$
$$\overset{(a)}{\leq} L_{y',n}\beta_{k,n-1}\mathbb{E}_k\big[\|y_k^{n,*} - y_{k+1}^n\|\|y_{k+1}^{n-1} - y_k^{n-1}\|\|\psi_k^{n-1}\|\big]$$
$$\overset{(b)}{\leq} L_{y',n}\beta_{k,n-1}^2\Big(\mathbb{E}_k\big[\|y_k^{n,*} - y_{k+1}^n\|\|h^{n-1}(y_k^{n-2}, y_k^{n-1})\|\|\psi_k^{n-1}\|\big] + \mathbb{E}_k\big[\|y_k^{n,*} - y_{k+1}^n\|\|\psi_k^{n-1}\|^2\big]\Big)$$
$$= L_{y',n}\beta_{k,n-1}^2\Big(\mathbb{E}_k\big[\|y_k^{n,*} - y_{k+1}^n\|\|h^{n-1}(y_k^{n-2}, y_k^{n-1})\|\mathbb{E}[\|\psi_k^{n-1}\|\,|\,\mathcal{F}_k^n]\big] + \mathbb{E}_k\big[\|y_k^{n,*} - y_{k+1}^n\|\|\psi_k^{n-1}\|^2\big]\Big)$$
$$\overset{(c)}{\leq} L_{y',n}\beta_{k,n-1}^2\Big(\sigma_{n-1}\mathbb{E}_k\big[\|y_k^{n,*} - y_{k+1}^n\|\|h^{n-1}(y_k^{n-2}, y_k^{n-1})\|\big] + \sigma_{n-1}^2\mathbb{E}_k\|y_k^{n,*} - y_{k+1}^n\|\Big)$$
$$\leq L_{y',n}\beta_{k,n-1}^2\Big(\sigma_{n-1}\mathbb{E}_k\big[\|y_k^{n,*} - y_{k+1}^n\|\|h^{n-1}(y_k^{n-2}, y_k^{n-1})\|\big] + \frac{\sigma_{n-1}^2}{2}\mathbb{E}_k\|y_k^{n,*} - y_{k+1}^n\|^2 + \frac{\sigma_{n-1}^2}{2}\Big)$$
$$\overset{(d)}{\leq} L_{y',n}\sigma_{n-1}\beta_{k,n-1}^2\Big(\frac{L_{h,n-1}+\sigma_{n-1}}{2}\mathbb{E}_k\|y_{k+1}^n - y_k^{n,*}\|^2 + \frac{L_{h,n-1}}{2}\|y_k^{n-1} - y_k^{n-1,*}\|^2 + \frac{\sigma_{n-1}}{2}\Big), \tag{44}$$

where (a) is due to

$$\|\hat{y}_{k+1}^{n-1} - y_k^{n-1}\| = (1-a)\|y_k^{n-1} - y_{k+1}^{n-1}\| \leq \|y_k^{n-1} - y_{k+1}^{n-1}\|, \tag{45}$$

then (b) is due to

$$\|y_{k+1}^{n-1} - y_k^{n-1}\| \leq \beta_k^{n-1}\big(\|h^{n-1}(y_k^{n-2}, y_k^{n-1})\| + \|\psi_k^{n-1}\|\big) \tag{46}$$

and (c) follows from Assumption 3 and Jensen's inequality:

$$\mathbb{E}[\|\psi_k^n\|] = \mathbb{E}[\sqrt{\|\psi_k^n\|^2}] \leq \sqrt{\mathbb{E}\|\psi_k^n\|^2} \leq \sigma_n, \tag{47}$$

the (d) follows from (42) and one-step Young's inequality:

$$\|y_k^{n,*} - y_{k+1}^n\|\|h^{n-1}(y_k^{n-2}, y_k^{n-1})\| \overset{(42)}{\leq} L_{h,n-1}\|y_k^{n,*} - y_{k+1}^n\|\|y_k^{n-1} - y_k^{n-1,*}\|$$
$$\leq \frac{L_{h,n-1}}{2}\|y_k^{n,*} - y_{k+1}^n\|^2 + \frac{L_{h,n-1}}{2}\|y_k^{n-1} - y_k^{n-1,*}\|^2. \tag{48}$$

The second term in (43) can be bounded as

$$\mathbb{E}_k \langle y_k^{n,*} - y_{k+1}^n, \beta_{k-1,n} \nabla y^{n,*}(y_k^{n-1})^\top \psi_k^{n-1} \rangle = \mathbb{E}_k\big[\langle y_k^{n,*} - y_{k+1}^n, \beta_{k,n-1}\nabla y^{n,*}(x_k)^\top \mathbb{E}[\psi_k^{n-1}|\mathcal{F}_k^n]\rangle\big]$$
$$\leq L_{y,n}\beta_{k,n-1}\mathbb{E}_k\big[\|y_k^{n,*} - y_{k+1}^n\|\|\mathbb{E}[\psi_k^{n-1}|\mathcal{F}_k^n]\|\big]$$
$$\overset{(a)}{\leq} \frac{L_{y,n}c_{n-1}}{2}\beta_{k,n-1}\big(\mathbb{E}_k\|y_k^{n,*} - y_{k+1}^n\|^2 + \beta_{k,n-1}\big) \tag{49}$$

where (a) follows from Assumption 3.

Collecting and substituting the upper bounds in (41), (44) and (49) into (40) yields

$$\mathbb{E}_k \langle y_k^{n,*} - y_{k+1}^n, y_{k+1}^{n,*} - y_k^{n,*} \rangle$$

$$\leq \Big( \Big( \frac{L_{y,n} c_{n-1}}{2} + \frac{2 L_{y,n}^2 L_{h,n-1}^2}{\lambda_{n-1}} \Big) \beta_{k,n-1} + L_{y',n} \sigma_{n-1} \frac{L_{h,n-1} + \sigma_{n-1}}{2} \beta_{k,n-1}^2 \Big) \mathbb{E}_k \| y_{k+1}^n - y_k^{n,*} \|^2$$

$$+ \Big( \frac{\lambda_{n-1}}{8} \beta_{k,n-1} + \frac{L_{y',n} \sigma_{n-1} L_{h,n-1}}{2} \beta_{k,n-1}^2 \Big) \| y_k^{n-1} - y_k^{n-1,*} \|^2 + \frac{L_{y',n} \sigma_{n-1}^2 + L_{y,n} c_{n-1}}{2} \beta_{k,n-1}^2.$$

$$(50)$$

The last term in (39) can be bounded as

$$\mathbb{E}_k \| y_k^{n,*} - y_{k+1}^{n,*} \|^2 \leq L_{y,n}^2 \beta_{k,n-1}^2 \mathbb{E}_k \| h^{n-1}(y_k^{n-2}, y_k^{n-1}) + \psi_k^{n-1} \|^2$$

$$\leq 2 L_{y,n}^2 \beta_{k,n-1}^2 \| h^{n-1}(y_k^{n-2}, y_k^{n-1}) \|^2 + 2 L_{y,n}^2 \sigma_{n-1}^2 \beta_{k,n-1}^2$$

$$\overset{(42)}{\leq} 2 L_{y,n}^2 L_{h,n-1}^2 \beta_{k,n-1}^2 \| y_k^{n-1} - y_k^{n-1,*} \|^2 + 2 L_{y,n}^2 \sigma_{n-1}^2 \beta_{k,n-1}^2. \quad (51)$$

Substituting the upper bounds in (50) and (51) into (39) yields (for $2 \leq n \leq N$)

$$\mathbb{E}_k \| y_{k+1}^n - y_{k+1}^{n,*} \|^2$$

$$\leq \Big( 1 + \big( L_{y,n} c_{n-1} + \frac{4 L_{y,n}^2 L_{h,n-1}^2}{\lambda_{n-1}} \big) \beta_{k,n-1} + L_{y',n} \sigma_{n-1} (L_{h,n-1} + \sigma_{n-1}) \beta_{k,n-1}^2 \Big) \mathbb{E}_k \| y_{k+1}^n - y_k^{n,*} \|^2$$

$$+ \frac{\lambda_{n-1}}{2} \beta_{k,n-1} \| y_k^{n-1} - y_k^{n-1,*} \|^2 + \big( L_{y',n} \sigma_{n-1}^2 + L_{y,n} c_{n-1} + 2 L_{y,n}^2 \sigma_{n-1}^2 \big) \beta_{k,n-1}^2 \quad (52)$$

where we have used the following condition of the step size to simplify the inequality:

$$(L_{y',n} \sigma_{n-1} L_{h,n-1} + 2 L_{y,n}^2 L_{h,n-1}^2) \beta_{k,n-1}^2 \leq \frac{\lambda_{n-1}}{4} \beta_{k,n-1}, \quad 2 \leq n \leq N. \quad (53)$$

**(2) When $n = 1$.** The update of $y_k^1$ is correlated with its upper level variable $x_k$ instead of $y_k^{n-1}$ when $n \geq 2$. And since the update of $x_k$ depends on all variables while the update of $y_k^{n-1}$ ($n \geq 2$) only depends on $y_k^{n-2}$, the analysis of $y_k^1$ is different from that of $y_k^n$ ($n > 2$). The difference therefore lies in analyzing (39), which captures the dependence of lower level variable to its upper level variable.

By the mean-value theorem, for some $\hat{x}_{k+1} = a x_k + (1 - a) x_{k+1}, a \in [0, 1]$, the second term in (39) can be rewritten as

$$\langle y_k^{1,*} - y_{k+1}^1, y_{k+1}^{1,*} - y_k^{1,*} \rangle = \langle y_k^{1,*} - y_{k+1}^1, \nabla y^{1,*}(\hat{x}_{k+1})^\top (x_{k+1} - x_k) \rangle$$

$$= \langle y_k^{1,*} - y_{k+1}^1, \alpha_k \nabla y^{1,*}(\hat{x}_{k+1})^\top v(x_k, y_k^{1:N}) \rangle$$

$$+ \langle y_k^{1,*} - y_{k+1}^1, \alpha_k \nabla y^{1,*}(\hat{x}_{k+1})^\top \xi_k \rangle. \quad (54)$$

The first term in the RHS of (54) can be bounded as

$$\langle y_k^{1,*} - y_{k+1}^1, \alpha_k \nabla y^{1,*}(\hat{x}_{k+1})^\top v(x_k, y_k^{1:N}) \rangle$$

$$\leq L_{y,1} \alpha_k \| y_k^{1,*} - y_{k+1}^1 \| \| v(x_k, y_k^{1:N}) \| \quad (55)$$

$$\overset{(a)}{\leq} L_{y,1} \alpha_k \Big( L_{v,y} \| y_k^{1,*} - y_{k+1}^1 \| \sum_{n=1}^N L_y(n) \| y_k^n - y_k^{n,*} \| + L_v \| y_k^{1,*} - y_{k+1}^1 \| \| x_k - x^* \| \Big)$$

$$\overset{(b)}{\leq} L_{y,1} \alpha_k \Big( \frac{L_{v,y}}{2} \| y_k^{1,*} - y_{k+1}^1 \|^2 + \frac{L_{v,y} N}{2} \sum_{n=1}^N L_y^2(n) \| y_k^n - y_k^{n,*} \|^2 + \frac{L_{y,1} L_v^2}{\lambda_0} \| y_k^{1,*} - y_{k+1}^1 \|^2 + \frac{\lambda_0}{4 L_{y,1}} \| x_k - x^* \|^2 \Big)$$

$$= \Big( \frac{L_{y,1} L_{v,y}}{2} + \frac{L_{y,1}^2 L_v^2}{\lambda_0} \Big) \alpha_k \| y_{k+1}^1 - y_k^{1,*} \|^2 + \frac{L_{y,1} L_{v,y} N}{2} \alpha_k \sum_{n=1}^N L_y^2(n) \| y_k^n - y_k^{n,*} \|^2 + \frac{\lambda_0}{4} \alpha_k \| x_k - x^* \|^2$$

$$(56)$$

and (a) follows from

$$\|v(x_k, y_k^{1:N})\| = \|v(x_k, y_k^{1:N}) - v(x_k) + v(x_k) - \underbrace{v(x^*)}_{=0}\|$$

$$\leq \|v(x_k, y_k^{1:N}) - v(x_k)\| + L_v\|x_k - x^*\|$$

$$\leq L_{v,y} \sum_{n=1}^{N} L_y(n)\|y_k^n - y_k^{n,*}\| + L_v\|x_k - x^*\|, \tag{57}$$

where the first inequality follows from Assumption 2 and the last inequality follows from Lemma 10; and (b) follows from Young's inequality:

$$\|y_k^{1,*} - y_{k+1}^1\| \sum_{n=1}^{N} L_y(n)\|y_k^n - y_k^{n,*}\| \leq \frac{1}{2}\|y_k^{1,*} - y_{k+1}^1\|^2 + \frac{1}{2}\left(\sum_{n=1}^{N} L_y(n)\|y_k^n - y_k^{n,*}\|\right)^2$$

$$\leq \frac{1}{2}\|y_k^{1,*} - y_{k+1}^1\|^2 + \sum_{n=1}^{N} \frac{N}{2} L_y^2(n)\|y_k^n - y_k^{n,*}\|^2, \quad \text{(58a)}$$

and

$$L_v\|y_k^{1,*} - y_{k+1}^1\|\|x_k - x^*\| \leq \frac{L_{y,1} L_v^2}{\lambda_0}\|y_k^{1,*} - y_{k+1}^1\|^2 + \frac{\lambda_0}{4L_{y,1}}\|x_k - x^*\|^2. \tag{58b}$$

The second term in the RHS of (54) can be further decomposed as

$$\mathbb{E}_k\langle y_k^{1,*} - y_{k+1}^1, \alpha_k \nabla y^{1,*}(\hat{x}_{k+1})^\top \xi_k\rangle$$
$$= \mathbb{E}_k\langle y_k^{1,*} - y_{k+1}^1, \alpha_k\left(\nabla y^{1,*}(\hat{x}_{k+1}) - \nabla y^{1,*}(x_k)\right)^\top \xi_k\rangle + \mathbb{E}_k\langle y_k^{1,*} - y_{k+1}^1, \alpha_k \nabla y^{1,*}(x_k)^\top \xi_k\rangle. \tag{59}$$

The first term in the RHS of (59) can be bounded similarly to (44), with the upper level update term $\|x_{k+1} - x_k\|$ in place of $\|y_{k+1}^{n-1} - y_k^{n-1}\|$ $(n>2)$, that is

$$\mathbb{E}_k\langle y_k^{1,*} - y_{k+1}^1, \alpha_k\left(\nabla y^{1,*}(\hat{x}_{k+1}) - \nabla y^{1,*}(x_k)\right)^\top \xi_k\rangle$$
$$= \mathbb{E}_k\langle y_k^{1,*} - y_{k+1}^1, \alpha_k\left(\nabla y^{1,*}(\hat{x}_{k+1}) - \nabla y^{1,*}(x_k)\right)^\top \xi_k\rangle$$
$$\leq L_{y',1}\alpha_k \mathbb{E}_k\left[\|y_k^{1,*} - y_{k+1}^1\|\|\hat{x}_{k+1} - x_k\|\|\xi_k\|\right]$$
$$\leq L_{y',1}\alpha_k \mathbb{E}_k\left[\|y_k^{1,*} - y_{k+1}^1\|\|x_{k+1} - x_k\|\|\xi_k\|\right]$$
$$\leq L_{y',1}\alpha_k^2\left(\mathbb{E}_k\left[\|y_k^{1,*} - y_{k+1}^1\|\|v(x_k, y_k^{1:N})\|\|\xi_k\|\right] + \mathbb{E}_k\left[\|y_k^{1,*} - y_{k+1}^1\|\|\xi_k\|^2\right]\right)$$
$$= L_{y',1}\alpha_k^2\left(\mathbb{E}_k\left[\|y_k^{1,*} - y_{k+1}^1\|\|v(x_k, y_k^{1:N})\|\mathbb{E}[\|\xi_k\||\mathcal{F}_k^1]\right] + \mathbb{E}_k\left[\|y_k^{1,*} - y_{k+1}^1\|\mathbb{E}[\|\xi_k\|^2|\mathcal{F}_k^1]\right]\right)$$
$$\leq L_{y',1}\alpha_k^2\left(\sigma_0 \mathbb{E}_k\left[\|y_k^{1,*} - y_{k+1}^1\|\|v(x_k, y_k^{1:N})\|\right] + \sigma_0^2 \mathbb{E}_k\|y_k^{1,*} - y_{k+1}^1\|\right)$$
$$\leq L_{y',1}\alpha_k^2\left(\sigma_0 \mathbb{E}_k\left[\|y_k^{1,*} - y_{k+1}^1\|\|v(x_k, y_k^{1:N})\|\right] + \frac{\sigma_0^2}{2}\mathbb{E}_k\|y_k^{1,*} - y_{k+1}^1\|^2 + \frac{\sigma_0^2}{2}\right) \tag{60}$$
$$\leq L_{y',1}\sigma_0\alpha_k^2\left(\frac{\sigma_0 + L_{v,y} + L_v}{2}\mathbb{E}_k\|y_{k+1}^1 - y_k^{1,*}\|^2 + \frac{L_{v,y}N}{2}\sum_{n=1}^{N}\|y_k^n - y_k^{n,*}\|^2 + \frac{L_v}{2}\|x_k - x^*\|^2 + \frac{\sigma_0}{2}\right), \tag{61}$$

where the fourth inequality follows from Assumption 3 and the last inequality follows from similar derivations of the upper bound of $\|y_k^{1,*} - y_{k+1}^1\|\|v(x_k, y_k^{1:N})\|$ shown in (55)–(56).

The second term in the RHS of (59) can be bounded as

$$\mathbb{E}_k\langle y_k^{1,*} - y_{k+1}^1, \alpha_k \nabla y^{1,*}(x_k)^\top \xi_k\rangle = \mathbb{E}_k\left[\langle y_k^{1,*} - y_{k+1}^1, \alpha_k \nabla y^{1,*}(x_k)^\top \mathbb{E}[\xi_k|\mathcal{F}_k^1]\rangle\right]$$

$$\leq L_{y,1}\alpha_k \mathbb{E}_k\left[\|y_k^{1,*} - y_{k+1}^1\|\|\mathbb{E}[\xi_k|\mathcal{F}_k^1]\|\right]$$

$$\leq \frac{L_{y,1} c_0}{2}\alpha_k\left(\mathbb{E}_k\|y_k^{1,*} - y_{k+1}^1\|^2 + \alpha_k\right). \tag{62}$$

Substituting the upper bounds in (56), (61) and (62) into (54) yields

$$
\mathbb{E}_k \langle y_k^{1,*} - y_{k+1}^1, y_{k+1}^1 - y_k^{1,*} \rangle
$$
$$
\leq \Big( \big( \big( \frac{L_{y,1} L_{v,y}}{2} + \frac{L_{y,1}^2 L_v^2}{\lambda_0} + \frac{L_{y,1} c_0}{2} \big) \alpha_k + L_{y',1} \frac{\sigma_0^2 + (L_{v,y} + L_v)\sigma_0}{2} \alpha_k^2 \big) \mathbb{E}_k \| y_{k+1}^1 - y_k^{1,*} \|^2
$$
$$
+ \big( \frac{L_{y,1} L_{v,y} N}{2} \alpha_k + \frac{L_{y',1} \sigma_0 L_{v,y} N}{2} \alpha_k^2 \big) N \sum_{n=1}^N L_y^2(n) \| y_k^n - y_k^{n,*} \|^2
$$
$$
+ \big( \frac{\lambda_0}{4} \alpha_k + \frac{L_{y',1} L_v \sigma_0}{2} \alpha_k^2 \big) \| x_k - x^* \|^2 + \frac{L_{y',1} \sigma_0^2 + L_{y,1} c_0}{2} \alpha_k^2. \tag{63}
$$

The last term in (39) can be bounded as

$$
\mathbb{E}_k \| y_k^{1,*} - y_{k+1}^1 \|^2
$$
$$
\leq L_{y,1}^2 \alpha_k^2 \mathbb{E}_k \| v(x_k, y_k^{1:N}) + \xi_k \|^2
$$
$$
\leq 2 L_{y,1}^2 \alpha_k^2 \| v(x_k, y_k^{1:N}) \|^2 + 2 L_{y,1}^2 \sigma_0^2 \alpha_k^2
$$
$$
\overset{(57)}{\leq} 4 L_{y,1}^2 \alpha_k^2 \big( L_{v,y}^2 N \sum_{n=1}^N L_y(n)^2 \| y_k^n - y_k^{n,*} \|^2 + L_v^2 \| x_k - x^* \|^2 \big) + 2 L_{y,1}^2 \sigma_0^2 \alpha_k^2. \tag{64}
$$

Substituting the upper bounds in (63) and (64) into (39) yields

$$
\mathbb{E}_k \| y_{k+1}^1 - y_{k+1}^{1,*} \|^2
$$
$$
\leq \big( 1 + L_{y,1} \big( L_{v,y} + 2 L_{y,1} L_v^2 \lambda_0^{-1} + c_0 \big) \alpha_k + L_{y',1} \sigma_0 (L_{v,y} + L_v + \sigma_0) \alpha_k^2 \big) \mathbb{E}_k \| y_{k+1}^1 - y_k^{1,*} \|^2
$$
$$
+ \big( L_{y,1} L_{v,y} N \alpha_k + (L_{y',1} \sigma_0 L_{v,y} + 4 L_{y,1}^2 L_{v,y}^2) N \alpha_k^2 \big) \sum_{n=1}^N L_y^2(n) \| y_k^n - y_k^{n,*} \|^2
$$
$$
+ \big( \frac{\lambda_0}{2} \alpha_k + (L_{y',1} L_v \sigma_0 + 4 L_{y,1}^2 L_v^2) \alpha_k^2 \big) \| x_k - x^* \|^2 + (L_{y',1} \sigma_0^2 + L_{y,1} c_0 + 2 L_{y,1}^2 \sigma_0^2) \alpha_k^2. \tag{65}
$$

This completes the analysis of lower-level sequences.

## B.2 Analysis of the main sequence

Recall that we defined the shorthand notations $y_k^{n,*} = y^{n,*}(y_k^{n-1})$ with $y_k^{1,*} = y^{1,*}(x_k)$; $y_k^{1:N} = (y_k^1, y_k^2, \ldots, y_k^N)$. For convenience, we write $\mathbb{E}[\cdot | \mathcal{F}_k]$ as $\mathbb{E}_k[\cdot]$. In this section, we will analyze the main sequence and then establish the convergence rate.

First we have

$$
\mathbb{E}_k \| x_{k+1} - x^* \|^2
$$
$$
= \| x_k - x^* \|^2 + 2 \alpha_k \mathbb{E}_k \langle x_k - x^*, v(x_k, y_k^{1:N}) + \xi_k \rangle + \mathbb{E}_k \| x_{k+1} - x_k \|^2
$$
$$
= \| x_k - x^* \|^2 + 2 \alpha_k \mathbb{E}_k \langle x_k - x^*, v(x_k, y_k^{1:N}) - v(x_k) \rangle + 2 \alpha_k \langle x_k - x^*, v(x_k) \rangle
$$
$$
+ 2 \alpha_k \langle x_k - x^*, \mathbb{E}_k[\xi_k] \rangle + \alpha_k^2 \mathbb{E}_k \| v(x_k, y_k^{1:N}) + \xi_k \|^2. \tag{66}
$$

By Lemma 10, the second term in (66) can be bounded as

$$
\langle x_k - x^*, v(x_k, y_k^{1:N}) - v(x_k) \rangle \leq L_{v,y} \| x_k - x^* \| \sum_{n=1}^N L_y(n) \| y_k^n - y_k^{n,*} \|
$$
$$
\leq \frac{\lambda_0}{8} \| x_k - x^* \|^2 + \frac{2 L_{v,y}^2 N}{\lambda_0} \sum_{n=1}^N L_y(n)^2 \| y_k^n - y_k^{n,*} \|^2. \tag{67}
$$

By the strong monotonicity of $v(x, y^*(x))$ in Assumption 5, the third term in (66) can be bounded as

$$
\langle x_k - x^*, v(x_k) \rangle \leq -\lambda_0 \| x_k - x^* \|^2. \tag{68}
$$

Using Assumption 3, the fourth term in (66) can be bounded as

$$\langle x_k - x^*, \mathbb{E}_k[\xi_k]\rangle \leq \frac{\lambda_0}{8}\|x_k - x^*\|^2 + \frac{2c_0^2}{\lambda_0}\alpha_k. \tag{69}$$

The last term in (66) can be bounded as

$$\mathbb{E}_k\|v(x_k, y_k^{1:N}) + \xi_k\|^2 \leq 2\|v(x_k, y_k^{1:N})\|^2 + 2\sigma_0^2$$

$$\overset{(57)}{\leq} 4L_v^2\|x_k - x^*\|^2 + 4NL_{v,y}^2\sum_{n=1}^{N}L_y(n)^2\|y_k^n - y_k^{n,*}\|^2 + 2\sigma_0^2. \tag{70}$$

Substituting the upper bounds in (67)–(70) into (66) yields

$$\mathbb{E}_k\|x_{k+1} - x^*\|^2 \leq \big(1 - \frac{3}{2}\lambda_0\alpha_k + 4L_v^2\alpha_k^2\big)\|x_k - x^*\|^2 + 4\big(\frac{L_{v,y}^2}{\lambda_0}\alpha_k + L_{v,y}^2\alpha_k^2\big)N\sum_{n=1}^{N}L_y^2(n)\|y_k^n - y_k^{n,*}\|^2$$

$$+ 2\big(\sigma_0^2 + \frac{2c_0^2}{\lambda_0}\big)\alpha_k^2. \tag{71}$$

**Establishing convergence.** For brevity, we fist define the following series

$$C_0(1) := L_{y,1}\big(L_{v,y} + 2L_{y,1}L_v^2\lambda_0^{-1} + c_0\big), \ C_1(1) := L_{y',1}\sigma_0(L_v + L_{v,y} + \sigma_0);$$

$$C_0(n) := L_{y,n}c_{n-1} + \frac{4L_{y,n}^2L_{h,n-1}^2}{\lambda_{n-1}}, \ C_1(n) := L_{y',n}\sigma_{n-1}(L_{h,n-1} + \sigma_{n-1}), 2 \leq n \leq N;$$

$$C_2(n) := (4\frac{L_{v,y}^2}{\lambda_0} + \frac{L_{y,1}L_{v,y}}{2})NL_y^2(n), \ C_3(n) := (L_{v,y}^2 + \frac{L_{y',1}\sigma_0 L_{v,y}}{2})NL_y^2(n), \forall n. \tag{72}$$

Define a Lyapunov function $\mathcal{J}_k := \|x_k - x^*\|^2 + \sum_{n=1}^{N}\|y_k^n - y_k^{n,*}\|^2$. Then we have

$$\mathbb{E}_k[\mathcal{J}_{k+1}] - \mathcal{J}_k = \mathbb{E}_k\|x_{k+1} - x^*\|^2 - \|x_k - x^*\|^2 + \sum_{n=1}^{N}\|y_{k+1}^n - y_{k+1}^{n,*}\|^2 - \|y_k^n - y_k^{n,*}\|^2. \tag{73}$$

Substituting (52), (65) and (71) into (73), and then applying (38) yields

$$\mathbb{E}_k[\mathcal{J}_{k+1}] - \mathcal{J}_k$$
$$\leq \big(-\lambda_0\alpha_k + \big(L_{y',1}L_v\sigma_0 + 4L_{y,1}^2L_v^2 + 4L_v^2\big)\alpha_k^2\big)\|x_k - x^*\|^2$$
$$+ \sum_{n=1}^{N-1}\big((1 + C_0(n)\beta_{k,n-1} + C_1(n)\beta_{k,n-1}^2)(1 - \lambda_n\beta_{k,n}) - 1 + \frac{\lambda_n}{2}\beta_{k,n} + C_2(n)\alpha_k + C_3(n)\alpha_k^2\big)\|y_k^n - y_k^{n,*}\|^2$$
$$+ \big((1 + C_0(N)\beta_{k,N-1} + C_1(N)\beta_{k,N-1}^2)(1 - \lambda_N\beta_{k,N}) - 1 + C_2(N)\alpha_k + C_3(N)\alpha_k^2\big)\|y_k^N - y_k^{N,*}\|^2$$
$$+ \Theta(\alpha_k^2) + \Theta\big(\sum_{n=1}^{N}(1 + \beta_{k,n-1} + \beta_{k,n-1}^2)\beta_{k,n}^2\big), \tag{74}$$

where we define $\beta_{k,0} := \alpha_k$ to simplify the result. As a clarification, the second term in the last inequality disappears when $N \leq 1$. Let the step sizes satisfy

$$-\lambda_0\alpha_k + \big(L_{y',1}L_v\sigma_0 + 4L_{y,1}^2L_v^2 + 4L_v^2\big)\alpha_k^2 \leq -\frac{\lambda_0}{2}\alpha_k, \tag{75}$$

$$(1 + C_0(n)\beta_{k,n-1} + C_1(n)\beta_{k,n-1}^2)(1 - \lambda_n\beta_{k,n}) - 1 + \frac{\lambda_n}{2}\beta_{k,n} + C_2(n)\alpha_k + C_3(n)\alpha_k^2 \leq -\frac{\lambda_0}{2}\alpha_k, 1 \leq n \leq N-1, \tag{76}$$

$$(1 + C_0(N)\beta_{k,n-1} + C_1(N)\beta_{k,n-1}^2)(1 - \lambda_N\beta_{k,N}) - 1 + C_2(N)\alpha_k + C_3(N)\alpha_k^2 \leq -\frac{\lambda_0}{2}\alpha_k, \tag{77}$$

Note that (75) always admits solution for small enough $\alpha_1$. Given $\beta_{k,N}$, applying Lemma 11 for $n = N, \ldots, 1$ to (77) and (76) implies that there exist solutions for $\beta_{k,n}(\forall n)$.

Then by (75)–(77), we have from (74) that

$$\mathbb{E}_k[\mathcal{J}_{k+1}] \leq \left(1 - \frac{\lambda_0}{2}\alpha_k\right)\mathcal{J}_k + \Theta(\alpha_k^2) + \Theta\left(\sum_{n=1}^{N}(1 + \beta_{k,n-1} + \beta_{k,n-1}^2)\beta_{k,n}^2\right). \tag{78}$$

Note that (78) implies a finite-time convergence rate of $\frac{1}{k}$ with the choice of step size. Applying Robbins-Siegmund's theorem stated in Lemma 12 to (78) gives $\sum_{k=1}^{\infty} \alpha_k \mathcal{J}_k < \infty$ and $\lim_{k\to\infty} \mathcal{J}_k < \infty$ almost surely, which along with the fact that $\sum_{k=1}^{\infty} \alpha_k = \infty$ implies $\lim_{k\to\infty} \mathcal{J}_k = 0$, i.e. for any $n \in [N]$

$$\lim_{k\to\infty} \|x_k - x^*\|^2 = 0, \qquad \lim_{k\to\infty} \|y_k^n - y_k^{n,*}\|^2 = 0, \ a.s. \tag{79}$$

Finally, as a direct result of Lemma 13, we can directly obtain the same convergence theorem for the alternative error metric $\|x_k - x^*\|^2 + \sum_{n=1}^{N} \|y_k^n - y^{n,*}\|^2$. This completes the proof.

## C  Proof of Theorem 2

### C.1  Analysis of the lower-level sequences

In this section, we provide a bound of the lower-level optimality gaps. Recall that we defined the shorthand notations $y_k^{n,*} = y^{n,*}(y_k^{n-1})$ with $y_k^{1,*} = y^{1,*}(x_k)$; $y_k^{1:N} = (y_k^1, y_k^2, \ldots, y_k^N)$. For convenience, we write $\mathbb{E}[\cdot|\mathcal{F}_k]$ as $\mathbb{E}_k[\cdot]$.

It follows from (38) that

$$\mathbb{E}_k\|y_{k+1}^n - y_k^{n,*}\|^2 \leq (1 - \lambda_n\beta_{k,n})\|y_k^n - y_k^{n,*}\|^2 + 2(\sigma_n^2 + c_n^2\lambda_n^{-1})\beta_{k,n}^2. \tag{80}$$

**Bounding the drifting optimality gap.** For any $n \geq 1$, we have

$$\|y_{k+1}^n - y_{k+1}^{n,*}\|^2 = \|y_{k+1}^n - y_k^{n,*}\|^2 + 2\langle y_k^{n,*} - y_{k+1}^n, y_{k+1}^{n,*} - y_k^{n,*}\rangle + \|y_k^{n,*} - y_{k+1}^{n,*}\|^2. \tag{81}$$

**(1) When $n = 1$.** By the mean-value theorem, for some $\hat{x}_{k+1} = ax_k + (1-a)x_{k+1}, a \in [0,1]$, the second term in (81) can be rewritten as

$$\begin{aligned}
\langle y_k^{1,*} - y_{k+1}^1, y_{k+1}^{1,*} - y_k^{1,*}\rangle &= \langle y_k^{1,*} - y_{k+1}^1, \nabla y^{1,*}(\hat{x}_{k+1})^\top (x_{k+1} - x_k)\rangle \\
&= \langle y_k^{1,*} - y_{k+1}^1, \alpha_k \nabla y^{1,*}(\hat{x}_{k+1})^\top v(x_k, y_k^{1:N})\rangle \\
&\quad + \langle y_k^{1,*} - y_{k+1}^1, \alpha_k \nabla y^{1,*}(\hat{x}_{k+1})^\top \xi_k\rangle.
\end{aligned} \tag{82}$$

The first term in (82) can be bounded as

$$\langle y_k^{1,*} - y_{k+1}^1, \alpha_k \nabla y^{1,*}(\hat{x}_{k+1})^\top v(x_k, y_k^{1:N})\rangle$$

$$\leq L_{y,1}\alpha_k \|y_k^{1,*} - y_{k+1}^1\| \|v(x_k, y_k^{1:N})\| \tag{83}$$

$$\leq L_{y,1}\alpha_k \left( L_{v,y}\|y_k^{1,*} - y_{k+1}^1\| \sum_{n=1}^{N} L_y(n)\|y_k^n - y_k^{n,*}\| + \|y_k^{1,*} - y_{k+1}^1\| \|v(x_k)\| \right)$$

$$\leq L_{y,1}\alpha_k \left( \frac{L_{v,y}}{2}\|y_k^{1,*} - y_{k+1}^1\|^2 + \frac{L_{v,y}N}{2}\sum_{n=1}^{N} L_y^2(n)\|y_k^n - y_k^{n,*}\|^2 \right.$$

$$\left. + 2L_{y,1}\|y_k^{1,*} - y_{k+1}^1\|^2 + \frac{1}{8L_{y,1}}\|v(x_k)\|^2 \right)$$

$$= L_{y,1}\left(\frac{L_{v,y}}{2} + 2L_{y,1}\right)\alpha_k\|y_{k+1}^1 - y_k^{1,*}\|^2 + \frac{L_{y,1}L_{v,y}N}{2}\alpha_k \sum_{n=1}^{N} L_y^2(n)\|y_k^n - y_k^{n,*}\|^2 + \frac{1}{8}\alpha_k\|v(x_k)\|^2, \tag{84}$$

where the second inequality follows from Lemma 10:

$$\|v(x_k, y_k^{1:N})\| \leq \|v(x_k, y_k^{1:N}) - v(x_k)\| + \|v(x_k)\|$$

$$\leq L_{v,y}\sum_{n=1}^{N} L_y(n)\|y_k^n - y_k^{n,*}\| + \|v(x_k)\|. \tag{85}$$

The second term in (82) can be further decomposed as

$$\mathbb{E}_k \langle y_k^{1,*} - y_{k+1}^1, \alpha_k \nabla y^{1,*}(\hat{x}_{k+1})^\top \xi_k \rangle$$

$$= \mathbb{E}_k \langle y_k^{1,*} - y_{k+1}^1, \alpha_k (\nabla y^{1,*}(\hat{x}_{k+1}) - \nabla y^{1,*}(x_k))^\top \xi_k \rangle + \mathbb{E}_k \langle y_k^{1,*} - y_{k+1}^1, \alpha_k \nabla y^{1,*}(x_k)^\top \xi_k \rangle.$$
(86)

The first term in (86) can be bounded as

$$\mathbb{E}_k \langle y_k^{1,*} - y_{k+1}^1, \alpha_k (\nabla y^{1,*}(\hat{x}_{k+1}) - \nabla y^{1,*}(x_k))^\top \xi_k \rangle \tag{87}$$

$$\overset{(60)}{\leq} L_{y',1} \sigma_0 \alpha_k^2 \Big( \mathbb{E}_k \big[ \| y_k^{1,*} - y_{k+1}^1 \| \| v(x_k, y_k^{1:N}) \| \big] + \frac{\sigma_0}{2} \mathbb{E}_k \| y_k^{1,*} - y_{k+1}^1 \|^2 + \frac{\sigma_0}{2} \Big)$$

$$\leq L_{y',1} \sigma_0 \alpha_k^2 \Big( \frac{L_{v,y} + \sigma_0 + 1}{2} \mathbb{E}_k \| y_{k+1}^1 - y_k^{1,*} \|^2 + \frac{L_{v,y} N}{2} \sum_{n=1}^N L_y^2(n) \| y_k^n - y_k^{n,*} \|^2 + \frac{1}{2} \| v(x_k) \|^2 + \frac{\sigma_0}{2} \Big)$$

where the last inequality follows from similar derivations of the upper bound of $\| y_k^{1,*} - y_{k+1}^1 \| \| v(x_k, y_k^{1:N}) \|$ shown in (83)–(84).

The second term in (86) can be bounded as

$$\mathbb{E}_k \langle y_k^{1,*} - y_{k+1}^1, \alpha_k \nabla y^{1,*}(x_k)^\top \xi_k \rangle \overset{(62)}{\leq} \frac{L_{y,1} c_0}{2} \alpha_k \big( \mathbb{E}_k \| y_k^{1,*} - y_{k+1}^1 \|^2 + \alpha_k \big). \tag{88}$$

Substituting the upper bounds in (84), (87) and (88) into (82) yields

$$\langle y_k^{1,*} - y_{k+1}^1, y_{k+1}^1 - y_k^{1,*} \rangle \leq \Big( L_{y,1} \big( \frac{L_{v,y} + c_0}{2} + 2L_{y,1} \big) \alpha_k + L_{y',1} \sigma_0 \frac{L_{v,y} + \sigma_0 + 1}{2} \alpha_k^2 \Big) \| y_{k+1}^1 - y_k^{1,*} \|^2$$

$$+ \frac{1}{2} \big( L_{y,1} L_{v,y} N \alpha_k + L_{y',1} \sigma_0 L_{v,y} N \alpha_k^2 \big) \sum_{n=1}^N L_y^2(n) \| y_k^n - y_k^{n,*} \|^2$$

$$+ \big( \frac{1}{8} \alpha_k + \frac{L_{y',1} \sigma_0}{2} \alpha_k^2 \big) \| v(x_k) \|^2 + \frac{L_{y',1} \sigma_0^2 + L_{y,1} c_0}{2} \alpha_k^2. \tag{89}$$

The last term in (81) can be bounded as

$$\mathbb{E}_k \| y_k^{1,*} - y_{k+1}^{1,*} \|^2$$

$$\leq L_{y,1}^2 \alpha_k^2 \mathbb{E}_k \| v(x_k, y_k^1) + \xi_k \|^2 \leq 2L_{y,1}^2 \alpha_k^2 \| v(x_k, y_k^1) \|^2 + 2L_{y,1}^2 \sigma_0^2 \alpha_k^2$$

$$\overset{(85)}{\leq} 4L_{v,y}^2 L_{y,1}^2 N \alpha_k^2 \sum_{n=1}^N L_y^2(n) \| y_k^n - y_k^{n,*} \|^2 + 4L_{y,1}^2 \alpha_k^2 \| v(x_k) \|^2 + 2L_{y,1}^2 \sigma_0^2 \alpha_k^2. \tag{90}$$

Substituting the upper bounds in (89) and (90) into (81) yields

$$\mathbb{E}_k \| y_{k+1}^1 - y_{k+1}^{1,*} \|^2$$

$$\leq \big( 1 + L_{y,1} (L_{v,y} + c_0 + 4L_{y,1}) \alpha_k + L_{y',1} \sigma_0 (L_{v,y} + \sigma_0 + 1) \alpha_k^2 \big) \mathbb{E}_k \| y_{k+1}^1 - y_k^{1,*} \|^2$$

$$+ \big( L_{y,1} L_{v,y} N \alpha_k + (L_{y',1} \sigma_0 L_{v,y} + 4L_{v,y}^2 L_{y,1}^2) N \alpha_k^2 \big) \sum_{n=1}^N L_y^2(n) \| y_k^n - y_k^{n,*} \|^2$$

$$+ \big( \frac{1}{4} \alpha_k + (L_{y',1} \sigma_0 + 4L_{y,1}^2) \alpha_k^2 \big) \| v(x_k) \|^2 + (L_{y',1} \sigma_0^2 + L_{y,1} c_0 + 2L_{y,1}^2 \sigma_0^2) \alpha_k^2. \tag{91}$$

**(2) When** $n \geq 2$. The update of $y_k^n$ $(n \geq 2)$ has no direct dependence on $x_k$, therefore the analysis is identical to that of Theorem 1. It directly follows from (51) that

$$\mathbb{E}_k \| y_{k+1}^n - y_{k+1}^{n,*} \|^2$$

$$\leq \big( 1 + \big( L_{y,n} c_{n-1} + \frac{4L_{y,n}^2 L_{h,n-1}^2}{\lambda_{n-1}} \big) \beta_{k,n-1} + L_{y',n} \sigma_{n-1} (L_{h,n-1} + \sigma_{n-1}) \beta_{k,n-1}^2 \big) \mathbb{E}_k \| y_{k+1}^n - y_k^{n,*} \|^2$$

$$+ \frac{\lambda_{n-1}}{2} \beta_{k,n-1} \| y_k^{n-1} - y_k^{n-1,*} \|^2 + (L_{y',n} \sigma_{n-1}^2 + L_{y,n} c_{n-1} + 2L_{y,n}^2 \sigma_{n-1}^2) \beta_{k,n-1}^2 \tag{92}$$

where we have imposed the following condition on the step size

$$(L_{y',n} \sigma_{n-1} L_{h,n-1} + 2L_{y,n}^2 L_{h,n-1}^2) \beta_{k,n-1}^2 \leq \frac{\lambda_{n-1}}{4} \beta_{k,n-1}, 2 \leq n \leq N. \tag{93}$$

This completes the analysis of the lower-level sequences.

## C.2  Analysis of the main sequence

In this section, we provide an analysis of the main sequence update, and then establish the finite-time convergence rate. Recall the shorthand notations $y_k^{n,*} = y^{n,*}(y_k^{n-1})$ with $y_k^{1,*} = y^{1,*}(x_k)$.

By the $L_v$-smoothness of $F(x)$, we have

$$
\mathbb{E}_k[F(x_{k+1})] - F(x_k)
$$
$$
\geq \mathbb{E}_k\langle v(x_k), x_{k+1} - x_k\rangle - \frac{L_v}{2}\mathbb{E}_k\|x_{k+1} - x_k\|^2
$$
$$
= \mathbb{E}_k\langle v(x_k), \alpha_k v(x_k, y_k^{1:N})\rangle + \mathbb{E}_k\langle v(x_k), \alpha_k \xi_k\rangle - \frac{L_v}{2}\mathbb{E}_k\|x_{k+1} - x_k\|^2. \tag{94}
$$

Define $L_y(n) := \sum_{i=n}^N L_{y,i-1}L_{y,i-2}\ldots L_{y,n}$ with $L_{y,n-1}L_{y,i-2}\ldots L_{y,n} := 1$. Using Lemma 10, the first term in (94) can be bounded as

$$
\langle v(x_k), \alpha_k v(x_k, y_k^{1:N})\rangle = \langle v(x_k), \alpha_k(v(x_k, y_k^{1:N}) - v(x_k))\rangle + \alpha_k\|v(x_k)\|^2
$$
$$
\geq -L_{v,y}\alpha_k\left[\|v(x_k)\|\sum_{n=1}^N L_y(n)\|y_k^n - y_k^{n,*}\|\right] + \alpha_k\|v(x_k)\|^2
$$
$$
\geq -\frac{\alpha_k}{4}\|v(x_k)\|^2 - L_{v,y}^2 N\alpha_k\sum_{n=1}^N L_y^2(n)\|y_k^n - y_k^{n,*}\|^2 + \alpha_k\|v(x_k)\|^2
$$
$$
= \frac{3\alpha_k}{4}\|v(x_k)\|^2 - L_{v,y}^2 N\alpha_k\sum_{n=1}^N L_y^2(n)\|y_k^n - y_k^{n,*}\|^2. \tag{95}
$$

The second term in (94) can be bounded as

$$
\mathbb{E}_k\langle v(x_k), \alpha_k\xi_k\rangle = \langle v(x_k), \alpha_k\mathbb{E}_k[\xi_k]\rangle
$$
$$
\geq -\frac{\alpha_k}{4}\|v(x_k)\|^2 - \alpha_k\|\mathbb{E}_k[\xi_k]\|^2
$$
$$
\geq -\frac{\alpha_k}{4}\|v(x_k)\|^2 - c_0^2\alpha_k^2. \tag{96}
$$

The last term in (94) can be bounded as

$$
\mathbb{E}_k\|x_{k+1} - x_k\|^2 \leq 2\alpha_k^2\left(\|v(x_k, y_k^{1:N})\|^2 + \mathbb{E}_k\|\xi_k\|^2\right)
$$
$$
\overset{(85)}{\leq} 4\alpha_k^2\|v(x_k)\|^2 + 4L_{v,y}^2 N\alpha_k^2\sum_{n=1}^N L_y^2(n)\|y_k^n - y_k^{n,*}\|^2 + 2\sigma_0^2\alpha_k^2. \tag{97}
$$

Substituting the bounds in (95), 96 and (97) into (94) yields

$$
\mathbb{E}_k[F(x_{k+1})] - F(x_k)
$$
$$
\geq \left(\frac{\alpha_k}{2} - 2L_v\alpha_k^2\right)\|v(x_k)\|^2 - N(L_{v,y}^2\alpha_k + 2L_v L_{v,y}^2\alpha_k^2)\sum_{n=1}^N L_y^2(n)\|y_k^n - y_k^{n,*}\|^2 - (L_v\sigma_0^2 + c_0^2)\alpha_k^2. \tag{98}
$$

**Establishing convergence.** For brevity, we fist define the following series

$$
C_4(1) := L_{y,1}(L_{v,y} + c_0 + 4L_{y,1}), \quad C_5(1) := L_{y',1}\sigma_0(L_{v,y} + \sigma_0 + 1);
$$
$$
C_4(n) := L_{y,n}c_{n-1} + 4L_{y,n}^2 L_{h,n-1}^2\lambda_{n-1}^{-1}, \quad C_5(n) := L_{y',n}\sigma_{n-1}(L_{h,n-1} + \sigma_{n-1}), 2 \leq n \leq N;
$$
$$
C_6(n) := (L_{y,1}L_{v,y} + L_{v,y}^2)NL_y^2(n), \quad C_7(n) := (L_{y',1}\sigma_0 L_{v,y} + 4L_{v,y}^2 L_{y,1}^2 + 2L_v L_{v,y}^2)NL_y^2(n), \forall n. \tag{99}
$$

Define a Lyapunov function $\mathcal{L}_k := -F(x_k) + \sum_{n=1}^N \|y_k^n - y_k^{n,*}\|^2$. Then we have

$$
\mathbb{E}_k[\mathcal{L}_{k+1}] - \mathcal{L}_k = F(x_k) - \mathbb{E}_k[F(x_{k+1})] + \sum_{n=1}^N \mathbb{E}_k\|y_{k+1}^n - y_{k+1}^{n,*}\|^2 - \|y_k^n - y_k^{n,*}\|^2. \tag{100}
$$

Substituting (91), (92) and (98) into (100), and then applying (38) yields

$$\mathbb{E}_k[\mathcal{L}_{k+1}] - \mathcal{L}_k$$

$$\leq \left(-\frac{1}{4}\alpha_k + (L_{y',1}\sigma_0 + 4L_{y,1}^2 + 2L_v)\alpha_k^2\right)\|v(x_k)\|^2$$

$$+ \sum_{n=1}^{N-1} \left((1 + C_4(n)\beta_{k,n-1} + C_5(n)\beta_{k,n-1}^2)(1 - \lambda_n\beta_{k,n}) - 1 + \frac{\lambda_n}{2}\beta_{k,n} + C_6(n)\alpha_k + C_7(n)\alpha_k^2\right)\|y_k^n - y_k^{n,*}\|^2$$

$$+ \left((1 + C_4(N)\beta_{k,N-1} + C_5(N)\beta_{k,N-1}^2)(1 - \lambda_N\beta_{k,N}) - 1 + C_6(N)\alpha_k + C_7(N)\alpha_k^2\right)\|y_k^N - y_k^{N,*}\|^2$$

$$+ \Theta(\alpha_k^2) + \Theta\left(\sum_{n=1}^{N}(1 + \beta_{k,n-1} + \beta_{k,n-1}^2)\beta_{k,n}^2\right). \tag{101}$$

As a clarification, the second term in the last inequality is $0$ when $N = 1$. We have also used $\beta_{k,0} = \alpha_k$. Consider the following choice of step sizes

$$-\frac{1}{4}\alpha_k + (L_{y',1}\sigma_0 + 4L_{y,1}^2 + 2L_v)\alpha_k^2 \leq -\frac{1}{8}\alpha_k, \tag{102}$$

$$(1 + C_1(n)\beta_{k,n-1} + C_2(n)\beta_{k,n-1}^2)(1 - \lambda_n\beta_{k,n}) - 1 + \frac{\lambda_n}{2}\beta_{k,n} + C_3(n)\alpha_k + C_4(n)\alpha_k^2 \leq -\lambda_n\alpha_k, n \leq N-1, \tag{103}$$

$$(1 + C_1(N)\beta_{k,N-1} + C_2(N)\beta_{k,N-1}^2)(1 - \lambda_N\beta_{k,N}) - 1 + C_3(N)\alpha_k + C_4(N)\alpha_k^2 \leq -\lambda_N\alpha_k. \tag{104}$$

Note that (102) always admits solution for small enough $\alpha_1$. Given $\beta_{k,N}$, applying Lemma 11 for $n = N, \ldots, 1$ to (104) and (103) tells that there exist solutions for $\beta_{k,n}(\forall n)$.

With (102)–(104), it follows from (101) that

$$\mathbb{E}_k[\mathcal{L}_{k+1}] - \mathcal{L}_k$$

$$\leq -\frac{\alpha_k}{8}\|v(x_k)\|^2 - \sum_{n=1}^{N}\lambda_n\alpha_k\|y_k^n - y_k^{n,*}\|^2 + \Theta(\alpha_k^2) + \Theta\left(\sum_{n=1}^{N}(1 + \beta_{k,n-1} + \beta_{k,n-1}^2)\beta_{k,n}^2\right). \tag{105}$$

Furthermore, taking expectation on both sides of (105) then summing over $k = 1, \ldots, K$ yields

$$\sum_{k=1}^{K}\alpha_k\mathbb{E}\left[\frac{1}{8}\|v(x_k)\|^2 + \lambda_n\|y_k^n - y_k^{n,*}\|^2\right]$$

$$\leq \mathcal{L}_1 - \mathbb{E}[\mathcal{L}_{K+1}] + \Theta\left(\sum_{k=1}^{K}\alpha_k^2\right) + \Theta\left(\sum_{k=1}^{K}\sum_{n=1}^{N}(1 + \beta_{k,n-1} + \beta_{k,n-1}^2)\beta_{k,n}^2\right)$$

$$\leq \mathcal{L}_1 + C_F + \Theta\left(\sum_{k=1}^{K}\alpha_k^2\right) + \Theta\left(\sum_{k=1}^{K}\sum_{n=1}^{N}(1 + \beta_{k,n-1} + \beta_{k,n-1}^2)\beta_{k,n}^2\right). \tag{106}$$

The inequality (106) implies a convergence rate of $\mathcal{O}(\frac{1}{\sqrt{K}})$ with step sizes $\alpha_k = \Theta(\frac{1}{\sqrt{K}})$ and $\beta_k = \Theta(\frac{1}{\sqrt{K}})$. This completes the proof.

## D  Proof of Lemma 1  and  Corollary 1

To prove the corollary, it suffices to prove Lemma 1 and then directly apply Theorem 1 and 2. We direct the readers interested in why we can relax the assumptions in [8] to the proof of Theorem 1 and 2. In particular, we provide a refined technique on bounding the drifting optimality gap in (39) and (81), which is crucial in alleviating the assumption.

**Proof.**  We start to verify the Assumptions by order.

**(1) Conditions (a) and (b)** $\Rightarrow$ **Assumption 1.** Since $g(x, y)$ is strongly-convex w.r.t. $y$, there exists a unique $y^*(x)$ such that $h(x, y^*(x)) = -\nabla_y g(x, y^*(x)) = 0$.

By [21, Lemma 2.2], we have

$$\|y^*(x) - y^*(x')\| \leq L_y \|x - x'\|, \ L_y = \frac{L_{xy}}{\lambda_1}. \tag{107}$$

By [8, Lemma 2], we have

$$\|\nabla y^*(x) - \nabla y^*(x')\| \leq L_{y'} \|x - x'\|, \ L_{y'} = \frac{l_{xy} + l_{xy} L_y}{\lambda_1} + \frac{L_{xy}(l_{yy} + l_{yy} L_y)}{\lambda_1^2}. \tag{108}$$

**(2) Conditions (a)–(c) $\Rightarrow$ Assumption 2.** By [21, Lemma 2.2], we have

$$\|v(x,y) - v(x,y')\| \leq L_{v,y} \|y - y'\|, \ L_{v,y} = l_{fx} + \frac{l_{fy} L_{xy}}{\lambda_1} + l_y\left(\frac{l_{xy}}{\lambda_1} + \frac{L_h L_{xy}}{\lambda_1^2}\right) \tag{109a}$$

$$\|v(x) - v(x')\| \leq L_v \|y - y'\|, \ L_v = \frac{L_{xy}(L_{v,y} + l'_{fy})}{\lambda_1} + l_{fx} + l_y\left(\frac{l_{xy} l_y}{\lambda_1} + \frac{l_{yy} L_{xy}}{\lambda_1^2}\right). \tag{109b}$$

Lastly, it follows from condition (b) that $\|h(x,y) - h(x,y')\| \leq L_h \|y - y'\|$.

**(3) Condition (e) $\Rightarrow$ Assumption 3; (a) $\Rightarrow$ Assumption 4; (d) $\Rightarrow$ Assumption 5; (f) $\Rightarrow$ Assumption 6.** These conditions directly imply their corresponding Assumption 3 when $N = 1$. ∎

# E Proof of Theorem 3

In this section, we will provide a proof of theorem 3. We omit all the index $n$ since $N = 1$. We also write $y^*(x_k)$ in short as $y_k^*$. With $A_x := \mathbb{E}_{s \sim \mu_{\pi_x}, a \sim \pi_x, s' \sim \mathcal{P}}[\phi(s)(\gamma\phi(s') - \phi(s))^\top], b_x :=$ $\mathbb{E}_{s \sim \mu_{\pi_x}, a \sim \pi_x}[r(s,a)\phi(s)]$, we list the conditions we need as follow. These conditions are also adopted in [57].

**Lemma 4 (Verification of assumptions)** *In the context of the AC update* (21) *and* (22). *Consider the following conditions*

(l) *For any $s \in \mathcal{S}$, $\|\phi(s)\| \leq 1$. For any $x \in \mathbb{R}^{d_0}$, there exists a constant $\lambda_1 > 0$ such that $\langle y - y', A_x(y - y')\rangle \leq -\lambda_1 \|y - y'\|^2$ for any $y, y' \in \mathbb{R}^{d_1}$. The smallest singular value of $A_x$ is lower bounded by $\sigma > 0$.*

(m) *There exist constants $L_\pi, L'_\pi$ and $C_\pi$ such that for any $s \in \mathcal{S}$ and $a \in \mathcal{A}$ and $x, x' \in \mathbb{R}^{d_0}$, the following inequalities hold: i) $\|\pi_x(a|s) - \pi_{x'}(a|s)\| \leq L_\pi \|x - x'\|$. ii) $\|\nabla \log \pi_x(a|s) - \nabla \log \pi_{x'}(a|s)\| \leq L'_\pi \|x - x'\|$. iii) $\|\nabla \log \pi_x(a|s)\| \leq C_\pi$.*

(n) *For any $x \in \mathbb{R}^{d_0}$, the Markov chain induced by the policy $\pi_x$ and transition kernel $\mathcal{P}$ is ergodic. There exist positive constants $\kappa$ and $\rho < 1$ such that*

$$\|\mathbb{P}_{\pi_x}(s_t \in \cdot | s_0 = s, a_0 = a) - \mu_{\pi_x}(\cdot)\|_{TV} \leq \kappa\rho^t, \forall(s,a) \in \mathcal{S} \times \mathcal{A}, \tag{110}$$

*where $\mathbb{P}_{\pi_x}(s_t \in \cdot | s_0, a_0)$ is the probability measure of the tth state $s_t$ on the Markov chain induced by policy $\pi_x$ and transition kernel $\mathcal{P}$, given the initial state and action $s_0, a_0$.*

(o) *The sampling protocol is: $s_k, a_k \sim d_{\pi_x}, s'_k \sim \mathcal{P}(\cdot | s_k, a_k)$; $\bar{s}_k \sim \mu_x, \bar{a}_k \sim \pi_x(\cdot | \bar{s}_k)$ and $\bar{s}'_k \sim \mathcal{P}(\cdot | \bar{s}_k, \bar{a}_k)$.*

*Consider the actor critic update defined in* (21) *and* (22). *Then we have:*

$$(l)\text{–}(n) \Rightarrow \text{Assumption 1}; \quad (l)\&(m) \Rightarrow \text{Assumption 2\&4}; \quad \text{Assumption 6 holds}. \tag{111}$$

*Moreover, a slightly more generalized version of Assumption 3 holds under condition (l)&(o):*

$$\mathbb{E}[\xi_k | \mathcal{F}_k^1] = 0, \ \mathbb{E}[\psi_k | \mathcal{F}_k] = 0,$$
$$\|\xi_k\|^2 \leq \sigma_0^2 + \bar{\sigma}_0^2 \|y_k - y^*(x_k)\|^2, \|\psi_k\|^2 \leq \sigma_1^2 + \bar{\sigma}_1^2 \|y_k - y^*(x_k)\|^2, \tag{112}$$

*where $\sigma_0^2 = 8C_\pi^2(1 + 4\sigma^{-2})$, $\bar{\sigma}_0^2 = 32C_\pi^2$, $\sigma_1^2 = 32\sigma^{-2} + 8$ and $\bar{\sigma}_1^2 = 32$.*

**Proof.** We will check the assumptions by order.

**(1) Condition (l)–(n) ⇒Assumption 1.** This is shown in Lemma 5.

**(2) Condition (l)&(m) ⇒Assumption 2&4.** We first check Assumption 2. In actor critic, we have $v(x) = v(x, y^*(x)) = \nabla F(x)$. By [64, Lemma 3.2], there exists a constant $L_v := \frac{L'_\pi}{(1-\gamma)^2} + \frac{(1+\gamma)C_\pi}{(1-\gamma)^2}$ such that

$$\|\nabla F(x) - \nabla F(x')\| \le L_v \|x - x'\|. \tag{113}$$

Then we have

$$\|v(x, y) - v(x, y')\| = \|\mathbb{E}[(\gamma\phi(s') - \phi(s))^\top (y - y')\nabla \log \pi_x(a|s)]\| \le 2C_\pi \|y - y'\|,$$
$$\|h(x, y) - h(x, y')\| = \|A_x(y - y')\| \le 2\|y - y'\|. \tag{114}$$

This completes the verification of Assumption 2. Lastly, Assumption 4 is directly implied by the inequality $\langle y - y', A_x(y - y')\rangle \le -\lambda_1 \|y - y'\|^2$ in condition (l).

**(3) Assumption 6 holds.** It is clear that $|F(x)| \le \frac{1}{1-\gamma}$.

**(4) Proving** (112). It is easy to check that $\mathbb{E}[\xi_k|\mathcal{F}_k^1] = 0$, $\mathbb{E}[\psi_k|\mathcal{F}_k] = 0$. Next we have

$$\begin{aligned}
\|\xi_k\|^2 &\le 2\mathbb{E}\|(r(s,a) + (\gamma\phi(s') - \phi(s))^\top y_k)\nabla \log \pi_{x_k}(a|s)]\|^2 \\
&\quad + 2\|(r(\bar{s}_k, \bar{a}_k) + (\gamma\phi(\bar{s}_k) - \phi(\bar{s}_k))^\top y_k)\nabla \log \pi_{x_k}(\bar{a}_k|\bar{s}_k)\|^2 \\
&\le 8C_\pi^2 + 16C_\pi^2\|y_k\|^2 \\
&\le 8C_\pi^2 + 32C_\pi^2\|y_k^*\|^2 + 32C_\pi^2\|y_k - y_k^*\|^2 \\
&\le 8C_\pi^2(1 + 4\sigma^{-2}) + 32C_\pi^2\|y_k - y_k^*\|^2 := \sigma_0^2 + \bar{\sigma}_0^2\|y_k - y_k^*\|^2,
\end{aligned} \tag{115}$$

where to get the last inequality we have used $\|y_k^*\| = \|A_{x_k}^{-1}b_{x_k}\| \le \sigma^{-1}$. Similarly we have

$$\begin{aligned}
\|\psi_k\|^2 &\le 2\mathbb{E}\|\phi(s)(\gamma\phi(s') - \phi(s))^\top y_k + r(s,a)\phi(s)\|^2 \\
&\quad + 2\|\phi(s_k)(\gamma\phi(s'_k) - \phi(s_k))^\top y_k + r(s_k, a_k)\phi(s_k)\|^2 \\
&\le 16\|y_k\|^2 + 8 \\
&\le 32\|y_k - y_k^*\|^2 + 32\sigma^{-2} + 8 := \sigma_1^2 + \bar{\sigma}_1^2\|y_k - y_k^*\|^2.
\end{aligned} \tag{116}$$

This completes the proof. ∎

We restate Theorem 3 as follows.

**Theorem 5 (Restatement of Theorem 3)** *Consider the sequences generated by* (21) *and* (22) *for $k = [K]$. Under conditions (l)–(o), Theorem 2 holds; that is, with $\alpha_k = \Theta(\frac{1}{\sqrt{K}})$ and $\beta_k = \Theta(\frac{1}{\sqrt{K}})$, we have*

$$\frac{1}{K}\sum_{k=1}^K \left(\mathbb{E}\|\nabla F(x_k)\|^2 + \mathbb{E}\|y_k - y^*(x_k)\|^2\right) = \mathcal{O}\left(\frac{1}{\sqrt{K}}\right). \tag{117}$$

We have verified the necessary assumptions for Theorem 2 to hold in Lemma 4, except that Assumption 3 needs a slight adaptation in AC. Thus the proof will be similar to that of Theorem 2, and only the steps that are different due to the adaptation of Assumption 3 will be shown here.

### E.1 Analysis of the critic optimality gap

**Contraction of the critic optimality gap.** First we have

$$\mathbb{E}_k\|y_{k+1} - y_k^*\|^2 = \|y_k - y_k^*\|^2 + 2\beta_k\mathbb{E}_k\langle y_k - y_k^*, h(x_k, y_k) + \psi_k\rangle + \mathbb{E}_k\|y_{k+1} - y_k\|^2, \tag{118}$$

The second term in (118) can be bounded as

$$\begin{aligned}
\mathbb{E}_k\langle y_k - y_k^*, h(x_k, y_k) + \psi_k\rangle &= \langle y_k - y_k^*, h(x_k, y_k)\rangle + \langle y_k - y_k^*, \mathbb{E}_k[\psi_k]\rangle \\
&\le -\lambda_1\|y_k - y_k^*\|^2.
\end{aligned} \tag{119}$$

where the last inequality follows from the strong monotonicity of $h(x, y)$ and $\mathbb{E}_k[\psi_k] = 0$ verified in Lemma 4.

The third term in (118) can be bounded as

$$
\begin{aligned}
\mathbb{E}_k \|y_{k+1} - y_k\|^2 &= \beta_k^2 \mathbb{E}_k \|h(x_k, y_k) + \psi_k\|^2 \\
&= \beta_k^2 (\|h(x_k, y_k)\|^2 + \mathbb{E}_k \|\psi_k\|^2) \\
&\leq \beta_k^2 (\|h(x_k, y_k)\|^2 + \sigma_1^2 + \bar{\sigma}_1^2 \|y_k - y_k^*\|^2) \\
&\leq (L_h^2 + \bar{\sigma}_1^2) \beta_k^2 \|y_k - y_k^*\|^2 + \sigma_1^2 \beta_k^2,
\end{aligned}
\tag{120}
$$

where the second last inequality follows from (112) and the last inequality follows from Assumption 2 which gives

$$
\|h(x, y)\| = \|h(x, y) - \underbrace{h(x, y^*(x))}_{=0}\| \leq L_h \|y - y^*(x)\|.
\tag{121}
$$

Collecting the upper bounds in (119) and (120) yields

$$
\begin{aligned}
\mathbb{E}_k \|y_{k+1} - y_k^*\|^2 &\leq (1 - 2\lambda_1 \beta_k + (L_h^2 + \bar{\sigma}_1^2) \beta_k^2) \|y_k - y_k^*\|^2 + \sigma_1^2 \beta_k^2 \\
&\leq (1 - \lambda_1 \beta_k) \|y_k - y_k^*\|^2 + \sigma_1^2 \beta_k^2,
\end{aligned}
\tag{122}
$$

where the last inequality is due to the choice of step size that satisfies $(L_h^2 + \bar{\sigma}_1^2) \beta_k^2 \leq \lambda_1 \beta_k$.

**Bounding the drifting optimality gap.** Next we start to bound the second term in (82) as follows

$$
\begin{aligned}
&\mathbb{E}_k \langle y_k^* - y_{k+1}, \alpha_k \nabla y^*(\hat{x}_{k+1})^\top \xi_k \rangle \\
&= \mathbb{E}_k \langle y_k^* - y_{k+1}, \alpha_k (\nabla y^*(\hat{x}_{k+1}) - \nabla y^*(x_k))^\top \xi_k \rangle + \mathbb{E}_k \langle y_k^* - y_{k+1}, \alpha_k \nabla y^*(x_k)^\top \mathbb{E}_k[\xi_k | \mathcal{F}_k^1] \rangle \\
&= \mathbb{E}_k \langle y_k^* - y_{k+1}, \alpha_k (\nabla y^*(\hat{x}_{k+1}) - \nabla y^*(x_k))^\top \xi_k \rangle \\
&\leq \alpha_k \mathbb{E}_k [\|y_k^* - y_{k+1}\| \|\nabla y^*(\hat{x}_{k+1}) - \nabla y^*(x_k)\| \|\xi_k\|] \\
&\leq \sigma_0 \alpha_k \mathbb{E}_k [\|y_k^* - y_{k+1}\| \|\nabla y^*(\hat{x}_{k+1}) - \nabla y^*(x_k)\|] \\
&\quad + \bar{\sigma}_0 \alpha_k \mathbb{E}_k [\|y_k^* - y_{k+1}\| \|\nabla y^*(\hat{x}_{k+1}) - \nabla y^*(x_k)\| \|y_k - y_k^*\|]
\end{aligned}
\tag{123}
$$

where the second inequality follows from $\mathbb{E}_k[\xi_k | \mathcal{F}_k^1] = 0$ shown in Lemma 4, and the last inequality follows from (112).

The first term in the RHS of (123) can be bounded as

$$
\begin{aligned}
&\mathbb{E}_k [\|y_k^* - y_{k+1}\| \|\nabla y^*(\hat{x}_{k+1}) - \nabla y^*(x_k)\|] \\
&\leq L_{y'} \mathbb{E}_k [\|y_k^* - y_{k+1}\| \|x_{k+1} - x_k\|] \\
&\leq L_{y'} \alpha_k (\mathbb{E}_k [\|y_k^* - y_{k+1}\| \|v(x_k, y_k)\|] + \mathbb{E}_k [\|y_k^* - y_{k+1}\| \|\xi_k\|]) \\
&\leq \frac{1}{2} L_{y'} \alpha_k (\mathbb{E}_k \|y_k^* - y_{k+1}\|^2 + \|v(x_k, y_k)\|^2 + \mathbb{E}_k \|y_k^* - y_{k+1}\|^2 + \|\xi_k\|^2) \\
&\leq \frac{1}{2} L_{y'} \alpha_k (2\mathbb{E}_k \|y_k^* - y_{k+1}\|^2 + \|v(x_k, y_k)\|^2 + \sigma_1^2 + \bar{\sigma}_1^2 \|y_k - y_k^*\|^2)
\end{aligned}
\tag{124}
$$

where the first inequality follows from Lemma 5 and the last inequality follows from (112).

The second term in (123) can be bounded as

$$
\begin{aligned}
\mathbb{E}_k [\|y_k^* - y_{k+1}\| \|\nabla y^*(\hat{x}_{k+1}) - \nabla y^*(x_k)\| \|y_k - y_k^*\|] &\leq 2L_y \mathbb{E}_k [\|y_k^* - y_{k+1}\| \|y_k - y_k^*\|] \\
&\leq L_y \mathbb{E}_k \|y_k^* - y_{k+1}\|^2 + L_y \|y_k - y_k^*\|^2.
\end{aligned}
\tag{125}
$$

Substituting (124) and (125) into (123), then substituting (123) and (83) into (82) gives

$$
\begin{aligned}
\mathbb{E}_k \langle y_k^* - y_{k+1}, y_{k+1}^* - y_k^* \rangle &\leq \left(L_y \left(\frac{L_{v,y}}{2} + 2L_y + \bar{\sigma}_0\right) \alpha_k + L_{y'} \sigma_0 \alpha_k^2\right) \mathbb{E}_k \|y_{k+1} - y_k^*\|^2 \\
&\quad + \frac{1}{2} \left(L_y (L_{v,y} + \bar{\sigma}_0) \alpha_k + L_{y'} \sigma_0 \bar{\sigma}_1^2 \alpha_k^2\right) \|y_k - y_k^*\|^2 \\
&\quad + \left(\frac{1}{8} \alpha_k + \frac{1}{2} L_{y'} \sigma_0 \alpha_k^2\right) \|v(x_k)\|^2 + \frac{1}{2} L_{y'} \sigma_0 \sigma_1^2 \alpha_k^2.
\end{aligned}
\tag{126}
$$

The last term in (81) can be bounded as

$$\mathbb{E}_k\|y_{k+1}^* - y_k^*\|^2 \le L_y^2\alpha_k^2\mathbb{E}_k\|v(x_k, y_k) + \xi_k\|^2 = L_y^2\alpha_k^2(\|v(x_k, y_k)\|^2 + \mathbb{E}_k\|\xi_k\|^2)$$
$$\le L_y^2\alpha_k^2(\|v(x_k, y_k)\|^2 + \sigma_0^2 + \bar\sigma_0^2\|y_k - y_k^*\|^2), \tag{127}$$

where the last inequality follows from (112). Substituting (126) and (127) into (81) gives

$$\mathbb{E}_k\|y_{k+1} - y_{k+1}^*\|^2 \le \big(1 + L_y\big(L_{v,y} + 4L_y + 2\bar\sigma_0\big)\alpha_k + 2L_{y'}\sigma_0\alpha_k^2\big)\mathbb{E}_k\|y_{k+1} - y_k^*\|^2$$
$$+ \big(L_y(L_{v,y} + \bar\sigma_0)\alpha_k + (L_{y'}\sigma_0\bar\sigma_1^2 + L_y^2\bar\sigma_0^2)\alpha_k^2\big)\|y_k - y_k^*\|^2$$
$$+ \big(\frac{1}{4}\alpha_k + (L_{y'}\sigma_0 + L_y^2)\alpha_k^2\big)\|v(x_k, y_k*)\|^2 + (\sigma_0\sigma_1^2 + L_y^2\sigma_0^2)\alpha_k^2. \tag{128}$$

## E.2  Analysis of the actor sequence

**Analysis of main sequence**. The second term in (94) is instead bounded as

$$\mathbb{E}_k\langle v(x_k, y_k), \alpha_k\xi_k\rangle = \langle v(x_k, y_k), \alpha_k\mathbb{E}_k[\xi_k]\rangle = 0. \tag{129}$$

Then the last term in (94) is instead bounded as

$$\mathbb{E}_k\|x_{k+1} - x_k\|^2 = \alpha_k^2(\|v(x_k, y_k)\|^2 + \mathbb{E}_k\|\xi_k\|^2) \le \alpha_k^2(\|v(x_k, y_k)\|^2 + \sigma_0^2 + \bar\sigma_0^2\|y_k - y_k^*\|^2). \tag{130}$$

Substituting the bounds in (95), (129) and (130) into (94) yields

$$\mathbb{E}_k[F(x_{k+1})] - F(x_k) \ge \big(\frac{3\alpha_k}{4} - \frac{L_v}{2}\alpha_k^2\big)\|v(x_k, y_k^*)\|^2 - (L_{v,y}^2\alpha_k + \frac{L_{v,y}}{2}\bar\sigma_0^2\alpha_k^2)\|y_k - y_k^*\|^2 - \frac{L_v\sigma_0^2}{2}\alpha_k^2. \tag{131}$$

**Establishing convergence.** Recall that the Lyapunov function $\mathcal{L}_k = -F(x_k) + \|y_k - y_k^*\|^2$. With the bounds in (122), (128) and (131), we have

$$\mathbb{E}_k[\mathcal{L}_{k+1}] - \mathcal{L}_k \le \big(-\frac{1}{2}\alpha_k + (\frac{L_v}{2} + L_{y'}\sigma_0 L_y^2)\alpha_k^2\big)\|v(x_k, y_k^*)\|^2$$
$$+ \big((1 + C_0'\alpha_k + C_1'\alpha_k^2)(1 - \lambda_1\beta_k) - 1 + C_2'\alpha_k + C_3'\alpha_k^2\big)\|y_k - y_k^*\|^2$$
$$+ \Theta(\alpha_k^2 + (1 + \alpha_k + \alpha_k^2)\beta_k^2), \tag{132}$$

where $C_0' := L_y\big(L_{v,y} + 4L_y + 2\bar\sigma_0\big)$, $C_1' := 2L_{y'}\sigma_0$, $C_2' := L_y(L_{v,y} + \bar\sigma_0) + L_{v,y}^2$, $C_3' := L_{y'}\sigma_0\bar\sigma_1^2 + \frac{L_{v,y}}{2}\bar\sigma_0^2$. Notice that (132) takes a similar form to that of (101) ($N = 1$).

If the step sizes are chosen such that

$$-\frac{1}{2}\alpha_k + (\frac{L_v}{2} + L_{y'}\sigma_0 L_y^2)\alpha_k^2 \le -\frac{1}{4}\alpha_k,$$
$$(1 + C_0'\alpha_k + C_1'\alpha_k^2)(1 - \lambda_1\beta_k) - 1 + C_2'\alpha_k + C_3'\alpha_k^2 \le -\lambda_1\alpha_k, \tag{133}$$

then it follows from the derivation after (101) that Theorem 2 holds for AC update.

## E.3  Supporting lemmas for Theorem 3

**Lemma 5 (Complete version of Lemma 2)**  *Consider the AC update in* (21)-(22). *Under conditions (l)–(n), there exist constants $L_y, L_{y'}$ such that*

$$\|y^*(x) - y^*(x')\| \le L_y\|x - x'\|,$$
$$\|\nabla y^*(x) - \nabla y^*(x')\| \le L_{y'}\|x - x'\|. \tag{134}$$

**Proof.** Under condition (l), we have $y^*(x) = -A_x^{-1}b_x$. With Lemma 7 and $\|A_x^{-1}\| \le \sigma^{-1}, \|b_x\| \le 1$, applying Lemma 14 to $y^*(x)$ implies that it is Lipschitz continuous with modulus

$$L_y := (\sigma^{-1} + 2\sigma^{-2})L_\mu'.$$

We next verify the Lipschitz continuity of $\nabla y^*(x)$. For $x \in \mathbb{R}^d$ and $f : \mathbb{R}^d \mapsto \mathbb{R}^{d_1 \times d_2}$, we denote $[x]_i$ as the $i$th element of $x$ and we use $\nabla_i f(x) := \frac{\partial f(x)}{\partial [x]_i}$. Then we have

$$\nabla_i y^*(x) = A_x^{-1} \nabla_i A_x A_x^{-1} b_x - A_x^{-1} \nabla_i b_x = -A_x^{-1} \nabla_i A_x y^*(x) - A_x^{-1} \nabla_i b_x. \tag{135}$$

By Lemma 14, to prove $\nabla_i y^*(x)$ is Lipschitz continuous w.r.t. $x$, it suffices to prove $\nabla_i A_x$, $y^*(x)$, $\nabla_i b_x$ and $A_x^{-1}$ are bounded (in norm) and Lipschitz continuous. First we have

$$\|A_x^{-1}\| \le \sigma^{-1}, \|y^*(x)\| \le \|A_x^{-1}\| \|b_x\| \le \sigma^{-1}. \tag{136}$$

And by Lemma 7, we have

$$\|A_x^{-1} - A_{x'}^{-1}\| \le 2\sigma^{-2} L_\mu' \|x - x'\|. \tag{137}$$

Thus it suffices to prove $\nabla_i A_x$ and $\nabla_i b_x$ are bounded in norm and Lipschitz continuous.

We start by

$$\nabla_i b_x = \mathbb{E}_{s \sim \mu_{\pi_x}, a \sim \pi_x(\cdot|s)} [\nabla_i \log \pi_x(a|s) G_x(s, a)], \tag{138}$$

where $G_x(s, a) := \mathbb{E}_{\pi_x} [\sum_{t=0}^{\infty} (r(s_t, a_t) \phi(s_t) - b_x) | s_0 = s, a_0 = a]$. By letting $\hat{r}(s, a, s') = r(s, a) \phi(s)$ in Lemma 8, we have

$$\|G_x(s, a)\| \le C_G, \ \|G_x(s, a) - G_{x'}(s, a)\| \le L_G \|x - x'\|, \tag{139}$$

where $C_G := 2 + \frac{\rho\kappa}{1-\rho}$ and $L_G := L_\mu' + \frac{\rho\kappa L_\pi |\mathcal{A}|}{1-\rho} + (\frac{\kappa}{1-\rho} + 1)^2 (L_\pi |\mathcal{A}| + L_\mu) + L_\mu$. Then we have $\|\nabla_i b_x\|$ can be bounded as

$$\|\nabla_i b_x\| \le C_\pi \mathbb{E}_{s \sim \mu_x, a \sim \pi_x(\cdot|s)} [\|G_x(s, a)\|] \le C_\pi C_G, \tag{140}$$

Now we start to prove the Lipschitz continuity of $\nabla_i b_x$. First we have

$$
\begin{aligned}
&\|\nabla_i b_x - \nabla_i b_{x'}\| \\
&\le \left\| \mathbb{E}_{s \sim \mu_{\pi_x}, a \sim \pi_x} [\nabla_i \log \pi_x(a|s) G_x(s, a)] - \mathbb{E}_{s \sim \mu_{\pi_{x'}}, a \sim \pi_{x'}} [\nabla_i \log \pi_x(a|s) G_x(s, a)] \right\| \\
&\quad + \mathbb{E}_{s \sim \mu_{\pi_{x'}}, a \sim \pi_{x'}} \left\| \nabla_i \log \pi_x(a|s) G_x(s, a) - \nabla_i \log \pi_{x'}(a|s) G_{x'}(s, a) \right\| \\
&\le \|\mu_{\pi_x} \cdot \pi_x - \mu_{\pi_{x'}} \cdot \pi_{x'}\|_{TV} \sup \|\nabla \log \pi_x(a|s) G_x(s, a)\| \\
&\quad + \mathbb{E}_{s \sim \mu_{\pi_{x'}}, a \sim \pi_{x'}} \left\| \nabla_i \log \pi_x(a|s) G_x(s, a) - \nabla_i \log \pi_{x'}(a|s) G_{x'}(s, a) \right\| \\
&\le C_G L_\mu' \|x - x'\| + (C_\pi L_G + L_\pi C_G) \|x - x'\| := L_b' \|x - x'\|, \tag{141}
\end{aligned}
$$

where the $\mu_{\pi_x} \cdot \pi_x$ denotes the probability measure specified by the probability function $(\mu_{\pi_x} \cdot \pi_x)(s, a) = \mu_{\pi_x}(s) \pi_x(a|s)$. In the second inequality, we apply Lemma 6 to the first term; and for the second term, we apply Lemma 14 along with (139) and condition (m).

For $\nabla_i A_x$, we have

$$\nabla_i A_x = \mathbb{E}_{s \sim \mu_{\pi_x}, a \sim \pi_x} [\nabla_i \log \pi_x(a|s) G_x(s, a)], \tag{142}$$

where we slightly abuse the notation and define $G_x(s, a) := \mathbb{E}_{\pi_x} [\sum_{t=0}^{\infty} (\phi(s_t)(\gamma\phi(s_{t+1}) - \phi(s_t))^\top - A_x) | s_0 = s, a_0 = a]$. Observing that $\nabla_i A_x$ has similar structure as that of $\nabla_i b_x$, we can apply the same technique and obtain

$$\|\nabla_i A_x\| \le C_\pi C_G',$$
$$\|\nabla_i A_x - \nabla_i A_{x'}\| \le C_G' L_\mu' \|x - x'\| + (C_\pi L_G' + L_\pi C_G') \|x - x'\| := L_A' \|x - x'\|, \tag{143}$$

where $C_G' := 4 + \frac{2\rho\kappa}{1-\rho}$ and $L_G' := 2L_\mu' + \frac{\rho\kappa L_\pi |\mathcal{A}|}{1-\rho} + (\frac{\kappa}{1-\rho} + 1)^2 (L_\pi |\mathcal{A}| + L_\mu) + L_\mu$.

Finally, applying Lemma 14 to (135) with (136), (137), (140), (141) and (143) yields

$$\|\nabla_i y^*(x) - \nabla_i y^*(x')\| \le L_{y'} \|x - x'\|, \tag{144}$$

where $L_{y'} := 2\sigma^{-3} L_\mu' C_\pi C_G' + L_A' \sigma^{-2} + L_y C_\pi C_G' \sigma^{-1} + 2\sigma^{-2} L_\mu' C_\pi C_G + \sigma^{-1} L_b'$. This completes the proof. ∎

**Lemma 6** *[66, Lemma 3] Define $(\mu_{\pi_x} \cdot \pi_x)(s,a) := \mu_{\pi_x}(s)\pi_x(a|s)$. Under conditions (n) and (m), it holds that*

$$\|\mu_{\pi_x} - \mu_{\pi_{x'}}\|_{TV} \leq L_\mu \|x - x'\|, \ \|\mu_{\pi_x} \cdot \pi_x - \mu_{\pi_{x'}} \cdot \pi_{x'}\|_{TV} \leq L'_\mu \|x - x'\| \quad (145)$$

*where $L_\mu := 2L_\pi |\mathcal{A}|(\log_\rho \kappa^{-1} + \frac{1}{1-\rho})$ and $L'_\mu := L_\mu + 2L_\pi |\mathcal{A}|$.*

**Lemma 7** *Define $\mu_{\pi_x} \cdot \pi_x(s,a) := \mu_{\pi_x}(s)\pi_x(a|s)$. Under conditions (n) and (m), the following inequalities hold*

$$\|A_x - A_{x'}\| \leq 2L'_\mu \|x - x'\|, \ \|A_x^{-1} - A_{x'}^{-1}\| \leq 2\sigma^{-2}L'_\mu \|x - x'\|, \ \|b_x - b_{x'}\| \leq L'_\mu \|x - x'\| \quad (146)$$

*where $L'_\mu = 2L_\pi |\mathcal{A}|(1 + \log_\rho \kappa^{-1} + \frac{1}{1-\rho})$.*

**Proof.** First we have

$$\|b_x - b_{x'}\| \leq \|\mu_{\pi_x} \cdot \pi_x - \mu_{\pi_{x'}} \cdot \pi_{x'}\|_{TV} \sup_{s,a} \|r(s,a)\phi(s)\| \leq L'_\mu \|x - x'\|, \quad (147)$$

where the last inequality follows from Lemma 6. And similarly, we have

$$\|A_x - A_{x'}\| \leq 2L'_\mu \|x - x'\|. \quad (148)$$

Finally, we have

$$\|A_x^{-1} - A_{x'}^{-1}\| = \|A_{x'}^{-1}(A_x - A_{x'})A_x^{-1}\| \leq \sigma^{-2}\|A_x - A_{x'}\|$$
$$\leq \sigma^{-2}\|\mu_{\pi_x} \cdot \pi_x - \mu_{\pi_{x'}} \cdot \pi_{x'}\|_{TV} \sup_{s,s'} \|\phi(s)(\gamma\phi(s') - \phi(s))\| \leq 2\sigma^{-2}L'_\mu \|x - x'\|, \quad (149)$$

where the last inequality follows from Lemma 6. This completes the proof. ∎

**Lemma 8** *Suppose conditions (l)–(n) hold. With mapping $\hat{r} : \mathcal{S} \times \mathcal{A} \times \mathcal{S} \mapsto \mathbb{R}^{d \times d'}$ such that $\|r(s,a,s')\| \leq C_r$ for any $(s,a,s')$, define*

$$G_x(s,a) := \mathbb{E}_{\substack{a_t \sim \pi_x(\cdot|s_t) \\ s_{t+1} \sim \mathcal{P}(\cdot|s_t,a_t)}} \Big[\sum_{t=0}^\infty \big(\hat{r}(s_t,a_t,s_{t+1}) - \bar{r}_x\big)\big|s_0 = s, a_0 = a\Big],$$
$$\text{with} \ \ \bar{r}_x := \mathbb{E}_{\substack{s \sim \mu_{\pi_x}, a \sim \pi_x(\cdot|s) \\ s' \sim \mathcal{P}(\cdot|s,a)}}[\hat{r}(s,a,s')]. \quad (150)$$

*Then there exists a constant $L_G$ such that for any $(s,a) \in \mathcal{S} \times \mathcal{A}$ and $x, x' \in \mathbb{R}^d$, the following inequalities hold*

$$\|G_x(s,a) - G_{x'}(s,a)\| \leq L_G \|x - x'\|,$$
$$\|G_x(s,a)\| \leq 2C_r + \frac{C_r \rho \kappa}{1 - \rho}. \quad (151)$$

**Proof.** We write $G_x(s_0, a_0)$ as:

$$G_x(s_0,a_0) = \mathbb{E}_{s_1 \sim \mathcal{P}}[\hat{r}(s_0,a_0,s_1)] - \bar{r}_x + \sum_{t=1}^\infty \Big(\sum_{(s,a) \in \mathcal{S} \times \mathcal{A}} \mathbf{Pr}_{\pi_x}(s_t = s|s_0,a_0)\pi_x(a|s)\mathbb{E}_{s' \sim \mathcal{P}}[\hat{r}(s,a,s')]$$
$$- \sum_{(s,a) \in \mathcal{S} \times \mathcal{A}} \mu_{\pi_x}(s)\pi_x(a|s)\mathbb{E}_{s' \sim \mathcal{P}}[\hat{r}(s,a,s')]\Big). \quad (152)$$

Given $(s_0, a_0)$, define the vector $p_1 := [\mathcal{P}(s^{(0)}|s_0,a_0), \mathcal{P}(s^{(1)}|s_0,a_0), ..., \mathcal{P}(s^{(|\mathcal{S}|)}|s_0,a_0)]$ where $s^{(0)}, ..., s^{(|\mathcal{S}|)}$ are states in $\mathcal{S}$. Given $\pi_x$, define the following state transition matrix

$$P_{\pi_x} := \begin{bmatrix} \mathcal{P}_{\pi_x}(s^{(0)}|s^{(0)}) & \mathcal{P}_{\pi_x}(s^{(1)}|s^{(0)}) & \dots & \mathcal{P}_{\pi_x}(s^{(|\mathcal{S}|)}|s^{(0)}) \\ \vdots & & & \\ \mathcal{P}_{\pi_x}(s^{(0)}|s^{(|\mathcal{S}|)}) & \mathcal{P}_{\pi_x}(s^{(1)}|s^{(|\mathcal{S}|)}) & \dots & \mathcal{P}_{\pi_x}(s^{(|\mathcal{S}|)}|s^{(|\mathcal{S}|)}) \end{bmatrix}, \quad (153)$$

where $\mathcal{P}_{\pi_x}(s'|s) = \sum_{a \in \mathcal{A}} \mathcal{P}(s'|s,a)\pi_x(a|s)$. Then it is clear that we can write the probability function $\mathbf{Pr}_{\pi_x}(s_t = \cdot|s_0, a_0)$ as its vector form $p_1 P_{\pi_x}^{t-1}$. We slightly abuse the notation and use $[p_1 P_{\pi_x}^t]_s = \mathbf{Pr}_{\pi_x}(s_t = s|s_0, a_0)$. Then (152) can be rewritten as

$$G_x(s_0, a_0) = \mathbb{E}_{s_1 \sim \mathcal{P}}[\hat{r}(s_0, a_0, s_1)] - \bar{r}_x + \sum_{t=0}^{\infty} \Big( \sum_{(s,a) \in \mathcal{S} \times \mathcal{A}} [p_1 P_{\pi_x}^t]_s \pi_x(a|s) \mathbb{E}_{s' \sim \mathcal{P}}[\hat{r}(s,a,s')]$$

$$- \sum_{s,a} [p_1 P_{\pi_x}^{\infty}]_s \pi_x(a|s) \mathbb{E}_{s' \sim \mathcal{P}}[\hat{r}(s,a,s')] \Big)$$

$$= \mathbb{E}_{s_1 \sim \mathcal{P}}[\hat{r}(s_0, a_0, s_1)] - \bar{r}_x + \sum_{t=0}^{\infty} \sum_{(s,a)} ([p_1 P_{\pi_x}^t]_s - [p_1 P_{\pi_x}^{\infty}]_s) \pi_x(a|s) \mathbb{E}_{s' \sim \mathcal{P}}[\hat{r}(s,a,s')]$$

$$= \mathbb{E}_{s_1 \sim \mathcal{P}}[\hat{r}(s_0, a_0, s_1)] - \bar{r}_x + \sum_{(s,a)} [p_1 Y_x]_s \pi_x(a|s) \mathbb{E}_{s' \sim \mathcal{P}}[\hat{r}(s,a,s')], \tag{154}$$

where $Y_x := \sum_{t=0}^{\infty}(P_{\pi_x}^t - P_{\pi_x}^{\infty})$. Then $\|G_x(s,a)\|$ can be bounded as follows

$$\|G_x(s,a)\| \leq 2C_r + C_r \sum_{s,a} |[p_1 Y_x]_s| \pi_x(a|s)$$

$$\leq 2C_r + C_r \sum_s |[p_1 Y_x]_s|$$

$$\leq 2C_r + \frac{C_r \rho \kappa}{1 - \rho} := C_G, \tag{155}$$

where the last inequality follows from condition (n) and

$$\sum_s |[p_1 Y_x]_s| \leq \sum_{t=1}^{\infty} \sum_s |\mathbf{Pr}_{\pi_x}(s_t = s|s_0, a_0) - \mu_{\pi_x}(s)|$$

$$= \sum_{t=1}^{\infty} \|\mathbb{P}_{\pi_x}(s_t \in \cdot|s_0, a_0) - \mu_{\pi_x}(\cdot)\|_{TV} \leq \frac{\rho \kappa}{1 - \rho}. \tag{156}$$

Then we have

$$\|G_x(s,a) - G_{x'}(s,a)\|$$

$$\leq \|\bar{r}_x - \bar{r}_{x'}\| + C_r \sum_{s,a} |[p_1 Y_x]_s| \|\pi_x(a|s) - \pi_{x'}(a|s)\| + C_r \sum_{s,a} |[p_1(Y_x - Y_{x'})]_s| \pi_{x'}(a|s)$$

$$\leq \|\bar{r}_x - \bar{r}_{x'}\| + \sum_s |[p_1 Y_x]_s| L_\pi |\mathcal{A}| \|x - x'\| + \|p_1(Y_x - Y_{x'})\|_1$$

$$\leq \|\bar{r}_x - \bar{r}_{x'}\| + \frac{\rho \kappa L_\pi |\mathcal{A}|}{1 - \rho} \|x - x'\| + \|Y_x - Y_{x'}\|_\infty \tag{157}$$

where the last inequality follows from (156). The first term in (157) can be bounded as

$$\|\bar{r}_x - \bar{r}_{x'}\| \leq \|\mu_x \cdot \pi_x - \mu_{x'} \cdot \pi_{x'}\|_{TV} \sup_{s,a,s'} \|r(s,a,s')\| \leq C_r L'_\mu, \tag{158}$$

where the last inequality follows from Lemma 6. By [40, Theorem 2.5], we have $Y_x + P_{\pi_x}^{\infty} = (I - P_{\pi_x} + P_{\pi_x}^{\infty})^{-1}$. First note that

$$\|(I - P_{\pi_x} + P_{\pi_x}^{\infty})^{-1}\|_\infty \leq \|Y_x\|_\infty + \|P_{\pi_x}^{\infty}\|_\infty$$

$$\leq \sum_{t=0}^{\infty} \|P_{\pi_x}^t - P_{\pi_x}^{\infty}\|_\infty + 1$$

$$= \sum_{t=0}^{\infty} \max_{s_0 \in \mathcal{S}} \sum_s |\mathbf{Pr}_{\pi_x}(s_t = s|s_0) - \mu_{\pi_x}(s)| + 1 \leq \frac{\kappa}{1 - \rho} + 1, \tag{159}$$

where the last inequality follows from condition (n).

We also have

$$\|(I - P_{\pi_x} + P_{\pi_x}^\infty)^{-1} - (I - P_{\pi_{x'}} + P_{\pi_{x'}}^\infty)^{-1}\|_\infty$$

$$\leq \|(I - P_{\pi_x} + P_{\pi_x}^\infty)^{-1}\|_\infty \|P_{\pi_x} - P_{\pi_{x'}} + P_{\pi_{x'}}^\infty - P_{\pi_x}^\infty\|_\infty \|(I - P_{\pi_{x'}} + P_{\pi_{x'}}^\infty)^{-1}\|_\infty$$

$$\overset{(159)}{\leq} \left(\frac{\kappa}{1-\rho} + 1\right)^2 \left(\|P_{\pi_x} - P_{\pi_{x'}}\|_\infty + \|P_{\pi_{x'}}^\infty - P_{\pi_x}^\infty\|_\infty\right)$$

$$\leq \left(\frac{\kappa}{1-\rho} + 1\right)^2 (L_\pi|\mathcal{A}| + L_\mu)\|x - x'\| \tag{160}$$

where in the last inequality we have used

$$\|P_{\pi_x} - P_{\pi_{x'}}\|_\infty = \max_s \sum_{s'} |\sum_a \pi_x(a|s)\mathcal{P}(s'|s,a) - \sum_a \pi_{x'}(a|s)\mathcal{P}(s'|s,a)|$$

$$= \max_s |\sum_a \pi_x(a|s) - \sum_a \pi_{x'}(a|s)| \sum_{s'} \mathcal{P}(s'|s,a)$$

$$\leq \max_s \sum_a |\pi_x(a|s) - \pi_{x'}(a|s)| \leq L_\pi|\mathcal{A}|\|x - x'\|,$$

$$\|P_{\pi_{x'}}^\infty - P_{\pi_{x'}}^\infty\|_\infty = \|\mu_{\pi_x} - \mu_{\pi_{x'}}\|_{TV} \leq L_\mu\|x - x'\| \quad \text{(Lemma 6)}. \tag{161}$$

With (160) and (161), we can write

$$\|Y_x - Y_{x'}\|_\infty \leq \|P_{\pi_x}^\infty - P_{\pi_{x'}}^\infty\|_\infty + \|(I - P_{\pi_x} + P_{\pi_x}^\infty)^{-1} - (I - P_{\pi_{x'}} + P_{\pi_{x'}}^\infty)^{-1}\|_\infty$$

$$\leq \left(\left(\frac{\kappa}{1-\rho} + 1\right)^2 (L_\pi|\mathcal{A}| + L_\mu) + L_\mu\right)\|x - x'\|. \tag{162}$$

Substituting (158) and (162) into (157) gives

$$\|G_x(s,a) - G_{x'}(s,a)\| \leq \left(C_r L'_\mu + \frac{\rho\kappa L_\pi|\mathcal{A}|}{1-\rho} + \left(\frac{\kappa}{1-\rho} + 1\right)^2 (L_\pi|\mathcal{A}| + L_\mu) + L_\mu\right)\|x - x'\|. \tag{163}$$

This completes the proof. ∎

# F   Proof of Lemma 3 and Corollary 2

Here we prove Lemma 3 which along with the generic Theorem 1 and 2 implies Corollary 2.

**Proof.** We will verify the assumptions by order.

**(1) Condition (g) $\Rightarrow$ Assumption 1.** Note that $y^{n,*}(y^{n-1}) = f^{n-1}(y^{n-1})$, then (g) directly implies Assumption 1 holds.

**(2) Condition (g) $\Rightarrow$ Assumption 2.** First note

$$v(x) = v\left(x, y^{1,*}(x), y^{2,*}(y^{1,*}(x)), \ldots, y^{N,*}(\ldots y^{2,*}(y^{1,*}(x))\ldots)\right)$$

$$= v\left(x, f^0(x), f^1(f^0(x)), \ldots, f^{N-1}(\ldots f^1(f^0(x))\ldots)\right)$$

$$= \nabla f^0(x)\nabla f^1(f^0(x))\cdots\nabla f^N(f^{N-1}(\ldots f^1(f^0(x))\ldots)). \tag{164}$$

By Lemma 14, in order for $v(x)$ to be Lipschitz continuous, it suffices to let $\nabla f^n(x)$ be bounded and Lipschitz continuous for every $n = 0, 1, \ldots, N$. This is satisfied under condition (g).

Now in order for $v(x, y^1, y^2, \ldots, y^N)$ be Lipschitz continuous w.r.t. $y^1, y^2, \ldots, y^N$, it again suffices to let $\nabla f^n(x)$ be bounded and Lipschitz continuous for every $n = 0, 1, \ldots, N$, which is satisfied under condition (g).

Finally, the Lipschitz continuity of $h^n(y^{n-1}, y^n)$ w.r.t. $y^n$ is directly implied by condition (g).

**(3) Condition (h)  and  (i) $\Rightarrow$ Assumption 3.** First we have

$$\mathbb{E}[\xi_k|\mathcal{F}_k^1] = -v(x_k, y_k^1, \ldots, y_k^N) + \mathbb{E}[\nabla f^0(x_k; \hat{\zeta}_k^0)\cdots\nabla f^N(y_k^N; \zeta_k^N)|\mathcal{F}_k^1]$$

$$= -v(x_k, y_k^1, \ldots, y_k^N) + \mathbb{E}[\nabla f^0(x_k; \hat{\zeta}_k^0)\cdots\nabla f^N(y_k^N; \zeta_k^N)|\mathcal{F}_k]$$

$$= -v(x_k, y_k^1, \ldots, y_k^N) + \nabla f^0(x_k)\cdots\nabla f^N(y_k^N) = 0, \tag{165}$$

where we have used the condition that $\hat{\zeta}_k^0, \zeta_k^0, \zeta_k^1, \ldots, \zeta_k^N$ are conditionally independent of each other given $\mathcal{F}_k$. The same goes for $\psi_k^n$ that

$$\mathbb{E}[\psi_k^n|\mathcal{F}_k^{n+1}] = -h^n(y_k^{n-1}, y_k^n) + \mathbb{E}[f^{n-1}(y_k^{n-1}, \zeta_k^{n-1})|\mathcal{F}_k] - y_k^n = 0. \tag{166}$$

The bounded variance condition directly implies that $\mathbb{E}[\|\psi_k^n\|^2|\mathcal{F}_k^{n+1}] < \infty$. Now for $\xi_k$ we have

$$\mathbb{E}[\|\xi_k\|^2|\mathcal{F}_k^1]$$
$$= \mathbb{E}[\|\xi_k\|^2|\mathcal{F}_k]$$
$$= \mathbb{E}_k\|\nabla f^0(x)\nabla f^1(y^1)\cdots\nabla f^N(y^N) - \nabla f^0(x_k;\hat{\zeta}_k^0)\cdots\nabla f^N(y_k^N;\zeta_k^N)\|^2$$
$$= \mathbb{E}_k\|\nabla f^0(x_k;\hat{\zeta}_k^0)\|^2...\mathbb{E}_k\|\nabla f^N(y_k^N;\zeta_k^N)\|^2 - \|\nabla f^0(x)\|^2\|\nabla f^1(y^1)\|^2...\|\nabla f^N(y^N)\|^2, \tag{167}$$

which is bounded by a constant since under contion (h), we have $\mathbb{E}_k\|\nabla f^n(x_k;\zeta_k^n)\|^2 < \infty$ for any $n$.

**(4) Condition (j) $\Rightarrow$ Assumption 5, (k) $\Rightarrow$ Assumption 6.** These assumptions are directly implied by the conditions.

**(5) Verifying Assumption 4.** By plugging in $y^{n,*}(y^{n-1}) = f^{n-1}(y^{n-1})$, it is immediate that Assumption 4 holds with $\lambda_n = 1$ for any $n \in [N]$. ∎

# G  Proof of Theorem 4

Before we prove the result, we first give a lemma that establishes the connection between Theorem 4 and the generic Theorem 2.

**Lemma 9** *In the context of the MAMPG update in (32) and (33). Consider the following conditions:*

- *(p) There exist constants $L_\pi, L'_\pi, L''_\pi$ and $C_\pi$ such that for any $(s,a) \in \mathcal{S} \times \mathcal{A}$ and $x, x' \in \mathbb{R}^{d_0}$, we have: i) $\|\pi_x(a|s) - \pi_{x'}(a|s)\| \leq L_\pi \|x - x'\|$; ii) $\|\nabla \log \pi_x(a|s) - \nabla \log \pi_{x'}(a|s)\| \leq L'_\pi \|x - x'\|$; iii) $\|\nabla^2 \log \pi_x(a|s) - \nabla^2 \log \pi_{x'}(a|s)\| \leq L''_\pi \|x - x'\|$ and iv) $\|\nabla \log \pi_x(a|s)\| \leq C_\pi$.*

- *(q) Given $\mathcal{F}_k$, we have for any $n \in \{1, \ldots, N\}$ and $i \in \{1, 2, ..., M\}$: $f_i^n(y_{k,i}^{n-1}; \zeta_{k,i}^n)$ and $\nabla f_i^n(y_{k,i}^n; \zeta_{k,i}^n)$ are respectively the unbiased estimators of $f_i^n(y_{k,i}^{n-1})$ and $\nabla f_i^n(y_{k,i}^n)$ with bounded variance. Likewise, $f_i^0(x_k; \zeta_{k,i}^0)$ and $\nabla f_i^0(x_k; \hat{\zeta}_{k,i}^0)$ are respectively unbiased estimators of $f_i^0(x_k)$ and $\nabla f_i^0(x_k)$ with bounded variance.*

- *(r) Given $\mathcal{F}_k$, $\hat{\zeta}_{k,i}^0, \zeta_{k,i}^0, \zeta_{k,i}^1, \ldots, \zeta_{k,i}^N$ are conditionally independent for $i = 1, 2, ..., M$.*

*We use $a \Rightarrow b$ to indicate that $a$ is a sufficient condition of $b$. Then we have*

$$(p) \Rightarrow \text{Assumption 1\&2}; \quad (q)\&(r) \Rightarrow \text{Assumption 3};$$
$$\textit{Assumption 4 holds naturally for (32)}; \quad \textit{Assumption 6 holds under bounded reward.} \tag{168}$$

Condition (p) is a standard assumption commonly adopted in the literature; see e.g., [14]. It is satisfied with certain popular policy parameterization such as the softmax policy. Conditions (q)&(r) can be satisfied with certain choice of the estimators and a simple sampling protocol.

**Proof.** We now check the assumptions by order.

**(1) (p)$\Rightarrow$ Assumption 1.** First we have $y^{n,*}(y^{n-1}) = f^{n-1}(y^{n-1})$. In order for the concatenation $f^{n-1}(y^{n-1})$ to be Lipschitz continuous and smooth, we only need each block $f_i^{n-1}(y_i^{n-1})$ to be Lipschitz continuous and smooth. Recall that $f_i^{n-1}(y_i^{n-1}) = y_i^{n-1} + \eta \nabla F_i(y_i^{n-1})$. The Lipschitz continuity of $f_i^{n-1}(y_i^{n-1})$ is guaranteed by the Lipschitz smoothness of $F_i(\cdot)$, which is well established in the literature [64]. Thus we only need to check the Lipschitz smoothness of $f_i^{n-1}(y_i^{n-1})$, that is, the Lipschitz continuity of $\nabla^2 F_i(y_i^{n-1})$. By [14], the policy hessian is given by

$$\nabla^2 F(x) = \mathbb{E}_{\zeta \sim p(\cdot|x)}\bigg[\underbrace{g(x;\zeta)\sum_{t=0}^{H}\nabla\log\pi_x(a_t|s_t)^\top + \nabla g(x;\zeta)}_{H(x;\zeta)}\bigg], \tag{169}$$

where $\zeta = (s_0, a_0, ..., s_H, a_H)$ and $p(\zeta|x) = \rho(s_0)\pi_x(a_0|s_0)\Pi_{t=0}^{H-1}\mathcal{P}(s_{t+1}|a_t, s_t)\pi_x(a_{t+1}|s_{t+1})$; $g(x;\zeta) := \sum_{h=0}^{H} \nabla \log \pi_x(a_h|s_h) \sum_{t=h}^{H} \gamma^t r(s_t, a_t)$, and we omit $i$ since the result holds for all $i$.

For any $x, x' \in \mathbb{R}^{d_0}$, we have

$$\|\nabla^2 F(x) - \nabla^2 F(x')\|$$
$$\leq \left\|\mathbb{E}_{\zeta\sim p(\cdot|x)}[H(x;\zeta)] - \mathbb{E}_{\zeta\sim p(\cdot|x')}[H(x;\zeta)]\right\| + \left\|\mathbb{E}_{\zeta\sim p(\cdot|x')}[H(x;\zeta)] - \mathbb{E}_{\zeta\sim p(\cdot|x')}[H(x';\zeta)]\right\|$$
$$\leq \left\|\mathbb{E}_{\zeta\sim p(\cdot|x)}[H(x;\zeta)] - \mathbb{E}_{\zeta\sim p(\cdot|x')}[H(x;\zeta)]\right\| + \mathbb{E}_{\zeta\sim p(\cdot|x')}\left\|H(x;\zeta) - H(x';\zeta)\right\|. \tag{170}$$

We consider the second term first. By Lemma 14, in order for $H(x;\zeta)$ to be Lipschitz continuous w.r.t. $x$, it suffices to prove: i) $\nabla \log \pi_x(a|s)$ can be bounded and Lipschitz continuous; ii) $g(x;\zeta)$ can be bounded and Lipschitz continuous; iii) $\nabla g(x;\zeta)$ is Lipschitz continuous. First, i) is directly implied by condition (p). We then prove ii) as follows

$$\|g(x;\zeta)\| \leq \sum_{h=0}^{H} \|\nabla \log \pi_x(a_h|s_h)\| |\sum_{t=h}^{H} \gamma^t r(s_t, a_t)| \leq \frac{C_\pi}{(1-\gamma)^2} \tag{171}$$

$$\|g(x;\zeta) - g(x';\zeta)\| = \sum_{h=0}^{H} \|\nabla \log \pi_x(a_h|s_h) - \nabla \log \pi_{x'}(a_h|s_h)\| \sum_{t=h}^{H} |\gamma^t r(s_t, a_t)|$$
$$\leq \frac{L'_\pi}{(1-\gamma)^2} \|x - x'\|. \tag{172}$$

Next we prove iii) as follows

$$\|\nabla g(x;\zeta) - \nabla g(x';\zeta)\| \leq \sum_{h=0}^{H} \|\nabla^2 \log \pi_x(a_h|s_h) - \nabla^2 \log \pi_{x'}(a_h|s_h)\| \sum_{t=h}^{H} |\gamma^t r(s_t, a_t)|$$
$$\leq \frac{L''_\pi}{(1-\gamma)^2} \|x - x'\|. \tag{173}$$

By Lemma 14, we know i), ii) and iii) imply the Lipschitz continuity of $H(x;\zeta)$, i.e. it holds that

$$\|H(x;\zeta) - H(x';\zeta)\| \leq \frac{L''_\pi + 2HC_\pi L'_\pi}{(1-\gamma)^2} \|x - x'\|. \tag{174}$$

The first term in (170) can be bounded as

$$\left\|\mathbb{E}_{\zeta\sim p(\cdot|x)}[H(x;\zeta)] - \mathbb{E}_{\zeta\sim p(\cdot|x')}[H(x;\zeta)]\right\| \leq \sup_\zeta \|H(x;\zeta)\| \sum_\zeta |p(\zeta|x) - p(\zeta|x')|$$
$$\overset{(176)}{\leq} \frac{HC_\pi^2 + L'_\pi}{(1-\gamma)^2} \sum_\zeta |p(\zeta|x) - p(\zeta|x')|$$
$$\leq \frac{HC_\pi^2 + L'_\pi}{(1-\gamma)^2}(H+1)|\mathcal{A}|L_\pi\|x - x'\| \tag{175}$$

where the second inequality follows from

$$\|H(x;\zeta)\| \leq \|g(x;\zeta)\| \sum_{t=0}^{H} \|\nabla \log \pi_x(a_t|s_t)\| + \|\nabla g(x;\zeta)\| \leq \frac{HC_\pi^2}{(1-\gamma)^2} + \frac{L'_\pi}{(1-\gamma)^2}. \tag{176}$$

Substituting (174) and (175) into (170) yields

$$\|\nabla^2 F(x) - \nabla^2 F(x')\| \leq \left(\frac{L''_\pi + 2HC_\pi L'_\pi}{(1-\gamma)^2} + \frac{HC_\pi^2 + L'_\pi}{(1-\gamma)^2}(H+1)|\mathcal{A}|L_\pi\right)\|x - x'\|. \tag{177}$$

This implies that $f_i^{n-1}(\cdot)$ is $\eta\left(\frac{L''_\pi + 2HC_\pi L'_\pi}{(1-\gamma)^2} + \frac{HC_\pi^2 + L'_\pi}{(1-\gamma)^2}(H+1)|\mathcal{A}|L_\pi\right)$-Lipschitz smooth for $n \in [N]$.

**(2) (q)&(r)$\Rightarrow$ Assumption 3**. It is clear that conditions (q)&(r) imply condition (h)&(i). Thus by Lemma 3, Assumption 3 is satisfied.

Now we only need to specify the estimators that satisfy condition (q) as follow. First, it is known that the policy gradient takes the following form [14]:

$$\nabla F_i(x) = \mathbb{E}_{\zeta \sim \pi_x} \Big[ \sum_{h=0}^{H} \nabla \log \pi_x(a_h|s_h) \sum_{t=h}^{H} \gamma^t r_i(s_t, a_t) | \rho_i, \mathcal{P}_i \Big]. \tag{178}$$

Then to estimate $f_i^n(y)$ $(n = 0, 1, ..., N-1)$, one can use:

$$f_i^n(y; \zeta_i^n) := y + \eta \sum_{h=0}^{H} \nabla \log \pi_y(a_h|s_h) \sum_{t=h}^{H} \gamma^t r_i(s_t, a_t), \ n = 0, 1, ..., N-1, \tag{179}$$

where $\zeta_i^n = (s_0, a_0, ..., s_H, a_H)$ is generated under policy $\pi_y$, transition distribution $\mathcal{P}_i$ and initial distribution $\rho_i$. The estimator satisfies condition (q):

$$\mathbb{E}_{\zeta_i^n}[f_i^n(y; \zeta_i^n)] = y + \eta \nabla F_i(y) = f_i^n(y),$$

$$\mathbb{E}_{\zeta_i^n}[\|f_i^n(y; \zeta_i^n) - f_i^n(x)\|^2] \le \mathbb{E}_{\zeta_i^n} \| \sum_{h=0}^{H} \nabla \log \pi_y(a_h|s_h) \sum_{t=h}^{H} \gamma^t r_i(s_t, a_t)\|^2 \le \frac{C_\pi^2}{(1-\gamma)^4}. \tag{180}$$

To estimate $\nabla f_i^n(y)$ $(n = 0, 1, ..., N-1)$, one can use:

$$\nabla f_i^n(y; \zeta_i^n) := I + \eta H(y; \zeta_i^n), \ n = 0, 1, ..., N-1, \tag{181}$$

where $\zeta_i^n = (s_0, a_0, ..., s_H, a_H)$ is generated under policy $\pi_y$, transition distribution $\mathcal{P}_i$ and initial distribution $\rho_i$. The estimator satisfies condition (q):

$$\mathbb{E}_{\zeta_i^n}[\nabla f_i^n(y; \zeta_i^n)] = I + \eta \nabla^2 F_i(y) = \nabla f_i^n(y),$$

$$\mathbb{E}_{\zeta_i^n}\|\nabla f_i^n(y; \zeta_i^n) - \nabla f_i^n(y)\|^2 \le \mathbb{E}_{\zeta_i^n}\|\nabla f_i^n(y; \zeta_i^n)\|^2 \overset{(176)}{\le} 2 + 2\eta^2 \frac{(HC_\pi^2 + L_\pi')^2}{(1-\gamma)^4}. \tag{182}$$

To estimate $\nabla f_i^N(x)$, one can use

$$\nabla f_i^N(x; \zeta_i^N) := \sum_{h=0}^{H} \nabla \log \pi_x(a_h|s_h) \sum_{t=h}^{H} \gamma^t r_i(s_t, a_t), \tag{183}$$

where $\zeta_i^n = (s_0, a_0, ..., s_H, a_H)$ is generated under policy $\pi_y$, transition kernel $\mathcal{P}_i$ and initial distribution $\rho_i$. This estimator satisfies the condition (q), following the similar lines in (180).

**(3) Verifying Assumption 4 and 6**. Assumption 4 is satisfied with $\lambda_n = 1$ by directly plugging in $y^{n,*}(y^{n-1}) = f^{n-1}(y^{n-1})$. Assumption 6 is satisfied by observing that

$$F(x) = \frac{1}{M} \sum_{i=1}^{M} F_i(\tilde{x}_i^N(x)) \le \frac{1}{1-\gamma}, \tag{184}$$

where we have used the fact that $F_i(x) \le \frac{1}{1-\gamma}$ for any $x$. ∎

Given the generic result in Theorem 2, Lemma 9 directly implies Theorem 4.

**Theorem 6 (Restatement of Theorem 4)** *Consider the sequences generated by the MAMPG update in* (32) *and* (33) *for* $k = [K]$. *Under conditions* (p)–(r), *we have Theorem 2 holds.*

## H  Technical Lemmas

**Lemma 10** *Suppose Assumption 1 & 2 hold. Recall that* $L_y(n) = \sum_{i=n}^{N} L_{y,i-1} L_{y,i-2} \dots L_{y,n}$ *with* $L_{y,n-1} L_{y,n-2} \dots L_{y,n} = 1$ *for any* $n \in [N]$. *Then it holds that*

$$\big\| v(x_k, y_k^{1:N}) - v(x_k) \big\| \le L_{v,y} \sum_{n=1}^{N} L_y(n) \| y_k^n - y^{n,*}(y_k^{n-1}) \|. \tag{185}$$

**Proof.** By the Lipschitz continuity of $v(x, y^1, \ldots, y^N)$ w.r.t. $y^1, \ldots, y^N$, we have

$$\|v(x_k, y_k^{1:N}) - v(x_k)\| \leq L_{v,y} \sum_{n=1}^{N} \|y_k^n - y^{n,*}(\ldots y^{2,*}(y^{1,*}(x_k)) \ldots)\|. \tag{186}$$

For any $n \geq 2$, we have

$$\begin{aligned}
&\|y_k^n - y^{n,*}(\ldots y^{2,*}(y^{1,*}(x_k)) \ldots)\| \\
&\leq \|y_k^n - y^{n,*}(y_k^{n-1})\| + \|y^{n,*}(y_k^{n-1}) - y^{n,*}(\ldots y^{2,*}(y^{1,*}(x_k)) \ldots)\| \\
&\leq \|y_k^n - y^{n,*}(y_k^{n-1})\| + L_{y,n-1}\|y_k^{n-1} - y^{n-1,*}(\ldots y^{2,*}(y^{1,*}(x_k)) \ldots)\|.
\end{aligned} \tag{187}$$

Unraveling yields

$$\|y_k^n - y^{n,*}(\ldots y^{2,*}(y^{1,*}(x_k)) \ldots)\| \leq \sum_{j=1}^{n} L_{y,n-1} L_{y,n-2} \ldots L_{y,j} \|y_k^j - y^{j,*}(y_k^{j-1})\|, \tag{188}$$

where $L_{y,n-1} L_{y,n-2} \ldots L_{y,n} := 1$. Substituting (188) into (186) completes the proof. ∎

**Lemma 11** *With any positive $\lambda_1$ and non-negative constants $\lambda_0$, $\lambda_2 < \lambda_1$ and $C_1, ..., C_4$, consider the following inequality about the step size $\beta_{k,n-1}$:*

$$(1 + C_1\beta_{k,n-1} + C_2\beta_{k,n-1}^2)(1 - \lambda_1\beta_{k,n}) - 1 + \lambda_2\beta_{k,n} + C_3\alpha_k + C_4\alpha_k^2 \leq -\lambda_0\alpha_k. \tag{189}$$

*Suppose all step sizes are in the same time-scale. Then given any $\beta_{k,n}$, if $\alpha_k \leq \beta_{k,n-1} \leq 1$, the above inequality always admits solutions for $\beta_{k,n-1}$.*

**Proof.** First we have

$$C_2\beta_{k,n-1}^2 \leq C_2\beta_{k,n-1}, \quad C_4\alpha_k^2 \leq C_4\alpha_k. \tag{190}$$

With the above inequality, we can simplify (189) to

$$(1 + (C_1 + C_2)\beta_{k,n-1})(1 - \lambda_1\beta_{k,n}) + \lambda_2\beta_{k,n} \leq 1 - (\lambda_0 + C_3 + C_4)\alpha_k. \tag{191}$$

By $\lambda_2\beta_{k,n} \leq (1 + (C_1 + C_2)\beta_{k,n-1})\lambda_2\beta_{k,n}$, the sufficient condition of (189) is

$$(1 + (C_1 + C_2)\beta_{k,n-1})(1 - \lambda'\beta_{k,n}) \leq 1 - (\lambda_0 + C_3 + C_4)\alpha_k. \tag{192}$$

where $\lambda' = \lambda_1 - \lambda_2 > 0$. Next we show that (192) holds. With $\alpha_k \leq \beta_{k,n-1}$, rearranging and simplifying (192) gives

$$\beta_{k,n-1} \leq \lambda' \frac{\beta_{k,n}}{\lambda_0 + C_1 + C_2 + C_3 + C_4}, \tag{193}$$

which can be satisfied if $\beta_{k,n-1}, \beta_{k,n}$ are in the same scale, and $\beta_{1,n-1}$ is small relative to $\beta_{1,n}$. ∎

**Lemma 12 (Robbins-Siegmund [18, Theorem 2.3.5])** *Consider a sequence of $\sigma$-algebras $\{\mathcal{F}_k\}_{k \geq 1}$ and four integrable non-negative sequences $\{U_k\}, \{V_k\}, \{\tau_k\}, \{\delta_k\}$ that satisfy*

  *i) $U_k, V_k, \tau_k, \delta_k$ are $\mathcal{F}_k$-measurable.*

  *ii) $\Pi_{k \geq 1}(1 + \tau_k) < \infty$ and $\sum_{k \geq 1} \mathbb{E}[\beta_k] < \infty$.*

  *iii) For $k \geq 1$, $\mathbb{E}[V_{k+1}|\mathcal{F}_k] \leq V_k(1 + \tau_k) + \delta_k - U_{k+1}$.*

*Then it holds that*

  *1) $V_k \xrightarrow{k \to \infty} V_\infty < \infty$ and $\sup_{k \geq 1} \mathbb{E}[V_k] < \infty$.*

  *2) $\sum_{k \geq 1} \mathbb{E}[U_k] < \infty$ and $\sum_{k \geq 1} U_k < \infty$ a.s.*

**Lemma 13** *Suppose Assumption 1 holds. Then there exists a positive constant $C_N$ such that*

$$\|x_k - x^*\|^2 + \sum_{n=1}^{N} \|y_k^n - y^{n,*}\|^2 \leq C_N \left( \|x_k - x^*\|^2 + \sum_{n=1}^{N} \|y_k^n - y^{n,*}(y_k^{n-1})\|^2 \right). \tag{194}$$

**Proof.** First note that under Assumption 1, we have

$$\sum_{n=1}^{N} \|y_k^n - y^{n,*}\| = \sum_{n=1}^{N} \|y_k^n - y^{n,*}(\ldots y^{2,*}(y^{1,*}(x^*)))\|. \tag{195}$$

To bound the RHS of the above inequality, we can directly follow the derivation of (186)–(188) with $x_k = x^*$ and obtain

$$\sum_{n=1}^{N} \|y_k^n - y^{n,*}\| = \sum_{n=1}^{N} \|y_k^n - y^{n,*}(\ldots y^{2,*}(y^{1,*}(x^*)))\|$$

$$\leq L_y(1)\|y_k^1 - y^{1,*}(x^*)\| + \sum_{n=2}^{N} L_y(n)\|y_k^n - y^{n,*}(y_k^{n-1})\|, \tag{196}$$

where $\{L_y(n)\}_{n=1}^{N}$ is a series of constants specified in Lemma 10.

Continuing from the last inequality, we have

$$\sum_{n=1}^{N} \|y_k^n - y^{n,*}\| \leq L_y(1)\|y^{1,*}(x_k) - y^{1,*}(x^*)\| + L_y(1)\|y_k^1 - y^{1,*}(x_k)\| + \sum_{n=2}^{N} L_y(n)\|y_k^n - y^{n,*}(y_k^{n-1})\|$$

$$\leq L_y(1)L_{y,1}\|x_k - x^*\| + \sum_{n=1}^{N} L_y(n)\|y_k^n - y^{n,*}(y_k^{n-1})\|. \tag{197}$$

Then we have

$$\|x_k - x^*\|^2 + \sum_{n=1}^{N} \|y_k^n - y^{n,*}\|^2$$

$$\leq \left( \|x_k - x^*\| + \sum_{n=1}^{N} \|y_k^n - y^{n,*}\| \right)^2$$

$$\overset{(197)}{\leq} 2(1 + L_y(1)L_{y,1})^2\|x_k - x^*\|^2 + 2N \sum_{n=1}^{N} L_y^2(n)\|y_k^n - y^{n,*}(y_k^{n-1})\|^2. \tag{198}$$

With the above inequality, choosing $C_N = 2\max\{(1 + L_y(1)L_{y,1})^2, NL_y^2(1), \ldots, NL_y^2(n)\}$ completes the proof. ∎

**Lemma 14 (Lipschitz continuity of a product.)** *Define* $f_i : \mathbb{R}^d \mapsto \mathbb{R}^{d_i \times d_{i+1}}$. *If there exist positive constants* $L_1, L_2, \ldots, L_n$ *and* $C_1, C_2, \ldots, C_n$ *such that for any* $x, x' \in \mathbb{R}^d$ *it holds that*

$$\|f_i(x) - f_i(x')\| \leq L_i\|x - x'\|, \quad \|f_i(x)\| \leq C_i, \quad \forall i \in [n]. \tag{199}$$

*Then it holds that*

$$\|f_1(x)f_2(x)\ldots f_n(x) - f_1(x')f_2(x')\ldots f_n(x')\| \leq \sum_{j=1}^{n} C_1 C_2 \ldots L_j \ldots C_n \|x - x'\|. \tag{200}$$

**Proof.** We can decompose the product as

$$\|f_1(x)f_2(x)\ldots f_n(x) - f_1(x')f_2(x')\ldots f_n(x')\|$$
$$= \|f_1(x)f_2(x)\ldots f_n(x) - f_1(x')f_2(x)\ldots f_n(x) + f_1(x')f_2(x)\ldots f_n(x) - f_1(x')f_2(x')\ldots f_n(x)$$
$$+ \cdots + f_1(x')f_2(x')\ldots f_n(x) - f_1(x')f_2(x')\ldots f_n(x')\|$$
$$\leq C_2\ldots C_n\|f_1(x) - f_1(x')\| + C_1 C_3 \ldots C_n\|f_2(x) - f_2(x')\| + \cdots + C_1 C_2 \ldots C_{n-1}\|f_n(x) - f_n(x')\|$$
$$\leq \sum_{j=1}^{n} C_1 C_2 \ldots L_j \ldots C_n \|x - x'\|. \tag{201}$$

This completes the proof. ∎