# OpenReview forum: "A Single-timescale Analysis for Stochastic Approximation with Multiple Coupled Sequences"
_NeurIPS.cc/2022/Conference — NeurIPS 2022 Accept_

### Official Review · Reviewer_xK38 · 2022-07-08

**Rating:** 8
**Confidence:** 3
**Soundness:** 3 good
**Presentation:** 4 excellent
**Contribution:** 4 excellent

**Summary:**

This paper fomulates the stochastic approximation of multiple coupled sequences and builds the non-asymptotic convergence rate for both strongly monotone and non-monotone case. Multiple applications are introduced after providing the theoretical convergence guarantee, including bilevel optimization and compositional optimization.

**Questions:**

* I agree that the monotonicity assumption (Assumption 4 and Assumption 5) is standard in many existing work. But I am not clear how this assumption is connected to concave or strongly concave function. Are they just different name?

**Limitations:**

Since this work is purely theoretical, there is no any negative impact.

**Strengths And Weaknesses:**

***Strengths***
* This work is sufficiently complete and well-written. It gives a clear definition of the multiple sequences stochastic approximation and builds the convergence rate for both of strongly monotone and non-monotone cases. Also, those derived bounds have matches the best known results.

* It is nice to see there is a unifying framework that can include stochastic bilevel optimization and stochastic compositional optimization. I am not sure if this paper is the first work doing so, but this paper reveals the significance of studying the stochastic approximation with multiple coupled sequences.

* The sngle-time-scale  learning rate scheduling can be better than the standard two-time-scale one. This result is really new.

***Weaknesses***
* I have not found any major weakness of this work.

---

> ### Author Response · Authors · 2022-08-02
> **Response to Reviewer xK38**
>
> Thank you for the supportive comments! Our response to your questions follows.
>
> **1. I agree that the monotonicity assumption (Assumption 4 and Assumption 5) is standard in many existing work. But I am not clear how this assumption is connected to concave or strongly concave function. Are they just different name?**
>
> Thanks for raising this question. Assumptions 4 and 5 are slightly weaker than the strong-concavity/convexity conditions. Let us explain.
>
> A function $h(x)$ is strongly-monotone on **a point $x'$** if there exists a constant $\mu>0$ such that $$
> \langle  h(x)-h(x'),x-x' \rangle \leq -\mu\|x-x'\|^2,~\forall x.
> $$
>
> On the other hand, as an implication of the strong-concavity of $g$, we have
> $$\langle \nabla g(x)-\nabla g(x'),x-x' \rangle \leq -\mu\|x-x'\|^2,~\forall x,x'$$ holds for some $\mu > 0$. Then if $h(x)=\nabla g(x)$, we see that $h(x)$ is strongly-monotone on **any point $x'$**. While in Assumption 4, given any $y^{n-1}$, we only assume $h^n(y^{n-1},y^n)$ (here $y^{n}$ is the variable) to be strongly-monotone on the optimal point $y^{n,*}(y^{n-1})$. This assumption is weaker than assuming $h^n(y^{n-1},y^n)$ to be strongly-monotone on any point $y^n$. Therefore, if $\nabla_{y^n} g^n (y^{n-1},y^n)=h^n (y^{n-1},y^n)$, Assumption 4 is weaker than the strong-concavity of $g^n (y^{n-1},\cdot)$.
>
> Thanks again for summarizing our paper and recognizing the merits of our paper!

---

### Official Review · Reviewer_QKd6 · 2022-07-10

**Rating:** 8
**Confidence:** 5
**Soundness:** 4 excellent
**Presentation:** 4 excellent
**Contribution:** 4 excellent

**Summary:**

The paper studies the finite-time convergence of nonlinear stochastic approximation (SA) with multiple coupled sequences. While there are few work on analyzing the performance of SA in this setting, existing analysis all adopt a multi-timescale analysis in the sense that the stepsizes for all the sequences decay at the different rates, which leads to a convergence rate that is slower than that of the single-sequence SA. The focus of this paper is on finding settings where it suffices to using single-timescale updates for multi-sequence SA, leading to the same convergence rate as that of the classic single-sequence SA. The implications of the new results on applications to bilevel, compositional and reinforcement learning have been discussed.

**Questions:**

1. If the smoothness assumption does not hold, can the existing complexity of SA be improved using the analysis in this paper?
2. In addition to stochastic bilevel and compositional optimization, what other new special class of stochastic optimization problems can benefit from this generic results?
3. It would be interesting to discuss and highlight in the main paper how the new proof improves the existing analysis. Now they are hidden in the supplementary document.


**Ethics Review Area:**

["I don’t know"]

**Strengths And Weaknesses:**

Strengths:
1.	This paper presents the ﬁrst $k^{-1}$ and $k^{-1/2}$ convergence rates for nonlinear SA under two settings: a) all sequences have strongly monotone increments; and b) all but the main sequence have strongly monotone increments. These results have been then extended to multiple coupled sequences with the same iteration complexity, which is also new.
2.	The paper presents a unified and new perspective of understanding the recent theoretical advances in stochastic bilevel, compositional optimization and reinforcement learning. Applying the new results of nonlinear SA with multiple coupled sequences to those special cases lead to either improved iteration/sample complexity or relaxed assumptions.
3.	The nice thing about this framework is that the iteration/sample complexity of new SA algorithms developed in these applications can be established by just verifying the assumptions in this paper. If those assumptions are verified, they automatically enjoy the same convergence rate as single-sequence stochastic approximation or SGD.
4.	The simulation, although very simple, gets the key point of this paper – there is a gap between the existing theory and the actual performance of nonlinear SA.

Weaknesses:
1.	The new iteration complexity results for nonlinear SA hold under the smoothness assumption on the fixed point. While the paper has justified it in several applications, it does not improve the existing complexity of SA without this smoothness assumption.
2.	The paper can do a better job on discussing and highlighting in the main paper how the new proof techniques improve the existing analysis.

---

> ### Author Response · Authors · 2022-08-02
> **Response to Reviewer QKd6**
>
> Thank you for appreciating our work! Our response to your questions follows.
>
> **1. If the smoothness assumption does not hold, can the existing complexity of SA be improved using the analysis in this paper?**
>
> In the current analysis, the smoothness of lower level optimal solution is essential in the decomposing of the lower level drift term in (51), which is the key enabler for the refined bounds thereafter. Without this condition, an improved rate may be not possible.
>
> **2. In addition to stochastic bilevel and compositional optimization, what other new special class of stochastic optimization problems can benefit from this generic results?**
>
> Another potential application is the min-max problem in the form of $\min_x\max_y f(x,y)$. Using stochastic gradient descent-ascent method will result in an update scheme in the form of STSA with two sequences, where $x_k$ is the main sequence and $y_k$ is the follower sequence. Under certain assumptions on $f(x,y)$ (e.g., smoothness of $f$, strong-concavity of $f$ with respect to $y$), we can cast the stochastic min-max method as a special case of the STSA.
>
> **3. It would be interesting to discuss and highlight in the main paper how the new proof improves the existing analysis. Now they are hidden in the supplementary document.**
>
> By exploiting the smoothness of $y^*$, we decompose the optimality drift term (e.g., (51) and (79)) into two terms. Through the thereafter refined analysis, we show that the two terms are decreasing without relying on the decay of $\alpha_k$. This allows $\alpha_k$ to be in the same time-scale as $\beta_{k,n}$ instead of being in a faster decaying time-scale, therefore improving the convergence rate. We agree that it is a good idea to highlight the crucial steps and will do so in the revision.
>
> Thanks again for the support and recognizing the contribution of our work!

---

### Official Review · Reviewer_gp8F · 2022-07-11

**Rating:** 6
**Confidence:** 1
**Soundness:** 2 fair
**Presentation:** 2 fair
**Contribution:** 2 fair

**Summary:**

This work proves the convergence guarantees for SA with double-sequence and also extends to multi-sequence and the results seem tight. Besides, this paper also applies their general results to SBO and SCO problems and show the improvement of their results.

**Questions:**

see Strengths And Weaknesses

**Limitations:**

see Strengths And Weaknesses

**Strengths And Weaknesses:**

As I'm not an expert in optimization area, I'm only able to give some common issues in this paper.

Questions:
1. I think Assumption 3 is a little bit weird because LHS is a random variable while the RHS is a non-random term and without a high probability guarantee. And I think it is not the generalized version of Assumption 2.1 in [30] because the second-order condition in Assumption 3 can induce moment assumption in [30].
2. I would be appreciated if the author could provide an example of a weak dependent sequence as Assumption 3 describes.
3. Is $y^n$ arbitrary in Assumption 4? If so, please use more precise language.
4. Line 169, it is not $\mathcal{O}(K^{-2})$ but $\mathcal{O}(\epsilon^{-2})$ or $\mathcal{O}(K^{-1/2})$

In conclusion, in my own opinion, I think this paper provides a solid result. But I will consider changing my score after reading other reviewers' comments and authors' responses.

---

> ### Author Response · Authors · 2022-08-02
> **Response to Reviewer gp8F**
>
> We thank the reviewer for the support and the careful review. Our response to your comments follows.
>
> **1. I think Assumption 3 is a little bit weird because LHS is a random variable while the RHS is a non-random term and without a high probability guarantee.**
>
> Indeed, we want to clarify that Assumption 3 needs to hold for any possible LHS.
>
> **And I think it is not the generalized version of Assumption 2.1 in [1] because the second-order condition in Assumption 3 can induce moment assumption in [1].**
>
> We are sorry for the confusion. We wish to make a correction here that Assumption 3 is not directly comparable with [Assumption 2.1, 1] since [Assumption 2.1, 1] assumes the independence between the noise variables across iterations. Assumption 3 is a generalization of the bias and variance assumption in stochastic programming [Assumption A1, 2] or the noise assumption in single-sequence SA [Assumption A4, 3] to multi-sequence case. In particular, it is easy to check that [Assumption A1, 2] implies Assumption 3 by setting $\xi_k=\nabla f(x_k)-G_k$ and $c_0=0$. When applying STSA to the stochastic optimization problems (see, e.g., Lemma 8), the conditional independence between the samples of different levels along with the standard small bias and bounded variance assumption in optimization will imply Assumption 3. We have rephrased the justification of Assumption 3 in the revision.
>
> ```
> [1] V. Konda and J. Tsitsiklis. Convergence rate of linear two-time-scale stochastic approximation.
> [2] S. Ghadimi and G. Lan. Stochastic first-and zeroth-order methods for nonconvex stochastic programming.
> [3] B. Karimi, B. Miasojedow, E. Moulines, and H. Wai. Non-asymptotic analysis of biased stochastic approximation scheme.
> ```
>
> **2. I would be appreciated if the author could provide an example of a weak dependent sequence as Assumption 3 describes.**
>
> Consider two-sequences SA. At each iteration, given $x_k$, let $z^\prime_k$ be a random variable with distribution $P(\cdot|x_k)$, and for $t\in \mathbb{N_0}$, let $z_{t,k}$ be a random variable with distribution $P_t(\cdot|x_k)$ . Suppose the sequences are generated with
> \begin{align}
> v(x_k,y_k)&= \mathbb{E}{\small z_{t,k}}\big[\sum_{t=0}^\infty \gamma^t f(x_k,y_k;z_{t,k})\big],~\xi_k=\sum_{t=0}^H \gamma^t f(x_k,y_k;z_{t,k})-\mathbb{E}{\small z_{t,k}}\big[\sum_{t=0}^\infty \gamma^t f(x_k,y_k;z_{t,k})\big],\nonumber\\\\
> h(x_k,y_k)&= \mathbb{E}{\small z_k^\prime}[g(x_k,y_k;z_k^\prime)], ~\psi_k = g(x_k,y_k;z_k^\prime)-\mathbb{E}{\small z_k^\prime}[g(x_k,y_k;z_k^\prime)]
> \end{align}
> where $f$ and $g$ are bounded functions ($C_f =\sup \Vert f \Vert$), $H \in \mathbb{N}$ and $\gamma \in (0,1)$. Suppose $z_{t,k}$ is conditionally indepenedent of  $z^\prime_k$ given $\mathcal{F_k}$, then we have
> \begin{align}
>  \Vert \mathbb{E}[\xi_k | \mathcal{F_k^1}] \Vert= \Vert \mathbb{E}[\xi_k | \mathcal{F_k}] \Vert=\big \Vert \mathbb{E}{\small z_{t,k}}\big[\sum_{t=H+1}^\infty \gamma^t f(x_k,y_k;z_{t,k})\big]\big \Vert \leq \sum_{t=H+1}^\infty \gamma^t C_f \leq \frac{\gamma^{H+1} C_f}{1-\gamma}.
> \end{align}
> Given $\alpha_k < 1$, setting $H=\lceil \log_{\gamma} \alpha_k \rceil$ leads to $c_0=\frac{\gamma C_f}{1-\gamma}$ in Assumption 3. It is easy to check that $c_1=0$. Lastly, $\sigma_0,\sigma_1$ exist by the boundedness of $f,g$.
>
> **3. Is $y^n$ arbitrary in Assumption 4? If so, please use more precise language.**
>
> Yes, $y^n$ is arbitrary in Assumption 4. We will make this clear in the revision.
>
> **4. Line 169, it is not $\mathcal{O}(K^{-2})$ but $\mathcal{O}(\epsilon^{-2})$ or $\mathcal{O}(K^{-1/2})$.**
>
> Yes, it should be $\mathcal{O}(\epsilon^{-2})$ in line 169. Thank you for the correction!
>
> We very much appreciate your time and efforts on reviewing our paper. After reading our rebuttal, we hope you can re-evaluate the merits of our paper.

---

> > ### Comment · Reviewer_gp8F · 2022-08-04
> > **Response**
> >
> > I find satisfactory by authors' response. I'd like to raise the score to 6.

---

### Official Review · Reviewer_qTJ6 · 2022-07-12

**Rating:** 6
**Confidence:** 4
**Soundness:** 3 good
**Presentation:** 2 fair
**Contribution:** 3 good

**Summary:**

In this paper, this paper studies the finite-time convergence of nonlinear SA with multiple coupled sequences. Different from existing multi-timescale analysis, the authors seek for scenarios where a fine-grained analysis can provide the tight performance guarantee for single-timescale multi-sequence SA (STSA). When all sequences have strongly monotone increments, they establish the iteration complexity of O(eps^{−1}) to achieve eps-accuracy, which improves the existing O(eps^{−1.5}) complexity for two coupled sequences. When the main sequence does not have strongly monotone increment, they establish the iteration complexity of O(eps^{−2}).

**Questions:**

1. Can we still get a single-timescale algorithm when \xi and \psi are not mutually independent?

**Limitations:**

According to the submission instructions, the authors need to complete the check list after the reference section. In this paper, the authors did not provide limitations of this work. I think the authors can further discuss the limitation of this theoretical analysis in their submission.

**Strengths And Weaknesses:**

Strengths:
1. This paper provides an improved rate for the case where all sequences have strongly monotone increments in single-timescale multi-sequence SA.
2. This paper further provides an analysis for a single-timescale multi-sequence SA when the main sequence does not have strongly monotone increment.
3. The theoretical analysis is novel, which makes significant contribution to the related area.

Weaknesses:

1. I think this paper lacks sufficient discussions and comparisons on the assumptions, which makes the technical part of this paper not sufficiently clear:

There is another work studying the single-timescale minimiax optimization [1], which is a special case of the single-timescale two-sequence SA when the main sequence does not have strongly monotone increment. In their Algorithm 1, they show that a momentum updating step is needed in order to obtain a single-timescale algorithm. However, this paper does not need the momentum updates. In my understanding, the main technique for the proof to get rid of momentum is in (57) and (59). And (57) and (59) depends on two important assumptions:

a)  \xi and \psi are mutually independent. Otherwise, the second equation in (57) does not hold. This means we need to sample twice independently in the application of the algorithm. In [1], they do not have this assumption. I think this mutual independence should be highlighted as a formal assumption and a comparison with the existing work is needed as well.

b) Existing works e.g., [1] [2], only need a Lipchitz assumption for y*, while this work makes a relatively stronger assumption that the gradient of y* exists. Such a comparison is also necessary in the paper to clarify the major difference in the assumptions and the reason why we can obtain a single-timescale algorithm.

[1] Single-Timescale Stochastic Nonconvex-Concave Optimization for Smooth Nonlinear TD Learning
[2] On Gradient Descent Ascent for Nonconvex-Concave Minimax Problems

2. According to the submission instructions, the authors need to complete the check list after the reference section. In this paper, the authors did not provide limitations of this work.

---

> ### Author Response · Authors · 2022-08-02
> **Response to Reviewer qTJ6**
>
> We thank the reviewer for the support and the careful review. Our response to your comments follows.
>
>
> **1. I think this paper lacks sufficient discussions and comparisons on the assumptions, which makes the technical part of this paper not sufficiently clear.**
>
> Thanks for your insightful comments and precise understanding on the technical details! Yes, we agree that more discussion on the assumptions will improve the clarity further. In the final version (with one additional page), we will add more discussion on the assumptions, and the detailed comparison with [1][2]. For now, we will respond to your questions below.
>
> **1a). $\xi$ and $\psi$ are mutually independent, otherwise, the second equation in (57) does not hold. In [1], they do not have this assumption. I think this mutual independence should be highlighted as a formal assumption and a comparison with the existing work is needed as well.**
>
> The second equation in (57) requires Assumption 3 to hold. And in general, the mutual independence between $\xi_k$ and $\psi_k$ is a sufficient but not necessary condition for Assumption 3. We agree that highlighting the mutual independence of $\xi_k$ and $\psi_k$, although it makes Assumption 3 slightly more restrictive, could be clear or easier to understand for audience. We can make this explicit in the revision if all the reviewers agree on this.
>
> **Can we still get a single-timescale algorithm when $\xi$ and $\psi$ are not mutually independent?**
>
> Under the current framework, the small bias condition, that is, the upper bound condition on $\|\mathbb{E}[\xi_k|\mathcal{F}_k^1]\|$ and $\|\mathbb{E}[\psi_k^n|\mathcal{F}_k^{n+1}]\|$ in Assumption 3, enables fast decreasing of the LHS of (59). When bounding LHS of (59), this condition allows us to treat $\xi_k$ as a bias upper bounded by a decreasing sequence by Assumption 3, as shown in the proof. If the small bias assumption does not hold, that is, we do not have the bound for $\mathbb{E}[\xi_k|\mathcal{F}_k^1]$, $\xi_k$ needs to be bounded as a variance instead. In latter case, a faster decreasing step size $\alpha_k$ is then needed to compensate for the non-decreasing of variance, which results in slower convergence. Nevertheless, it is very interesting to think about how the independence condition can be relaxed.
>
>
> **1b). Existing works e.g., [1] [2], only need a Lipchitz assumption for $y^\ast$, while this work makes a relatively stronger assumption that the gradient of $y^\ast$ exists. Such a comparison is also necessary in the paper to clarify the major difference in the assumptions and the reason why we can obtain a single-timescale algorithm.**
>
>
> From our understanding, as is shown in Lemma B.4 of [1], the momentum update reduces the noise of the stochastic gradient, thus enabling a single time-scale algorithm. In [2], a large batch size of $\Theta(\epsilon^{-2})$ is used to ensure a small varaince, which enables the constant step size. While in this work, we do not assume a variance reduction effect. Our analysis relies on the additional smoothness assumption on $y^*$. In particular, the smoothness condition enables the decomposition of the optimality drift terms (e.g., (51) and (79)) into two terms. As shown in the fine-grained analysis thereafter, as iterates converge, the two terms are fast decreasing without relying on the decay of $\alpha_k$, allowing $\alpha_k$ to be in the same time-scale as $\beta_{k,n}$ rather than decreasing in a faster time-scale. In the final revision, we will make a formal comparison with [1,2], while highlighting crucial steps in the analysis that lead to the theoretical merits.
>
> ```
> [1] S. Qiu, Z. Yang, X. Wei, J. Ye, and Z. Wang. Single-Timescale Stochastic Nonconvex-Concave Optimization for Smooth Nonlinear TD Learning.
> [2] T. Lin, C. Jin, and M. Jordan. On Gradient Descent Ascent for Nonconvex-Concave Minimax Problems.
> ```
>
> **2. In this paper, the authors did not provide limitations of this work.**
>
> We have added the limitation in the checklist and copied below.
>
> > This work only considers the strongly-monotone increments for the follower sequences, but it will be interesting to also consider the monotone increments with non-unique fixed point $y^*$. Right now the generic result in this work can only be applied to the unconstrained stochastic optimization, while it will be also interesting to consider whether it is possible to establish similar result that is applicable to the constrained stochastic optimization problems. We will add discussion on the limitations in the revision.
>
> Thank you again for the careful review. Hope these resolve your remaining concerns.

---

> > ### Comment · Reviewer_qTJ6 · 2022-08-08
> > **Response**
> >
> > Thanks for the responses. Overall, I think the writing of the current submission is not sufficiently clear due to the lack of the above important discussions. In my understanding, I still feel that Lipchitz assumption for $y*$ is a kind of stronger assumption, which gives a chance to directly get a single-timescale algorithm without modifying the algorithm. But obtaining a single-timescale algorithm by exploring such a stronger assumption is still an interesting contribution. Thus, I would like to keep my score unchanged.

---

### Meta-Review · Area_Chair_yT38 · 2022-08-22

**Recommendation:** Accept
**Confidence:** Certain

**Metareview:**

This paper provides convergence analysis for nonlinear stochastic approximation with a "multi-sequence" update structure motivated by applications in reinforcement learning and bilevel learning. When all sequences have strongly monotone increments, the authors provide iteration complexity of O(\epsilon^{−1}) to achieve accuracy, which improves the existing O(\epsilon^{−1.5}) complexity for two coupled sequences. When the main sequence does not have strongly monotone increments, they establish iteration complexity O(\epsilon^{−2}).

The reviewers agreed that the techniques in this paper are novel, and that it is well-written. In addition, the paper improves upon existing results when applied to problems in reinforcement learning and bilevel optimization, and hence is likely to have broader impact. However, the reviewers felt that the for the final version, the discussion of the smoothness assumption needs to be expanded, and the comparison with prior work needs to be improved.

**Award:**

No

---

### Decision · Program_Chairs · 2022-09-14

Accept